# Decapping factor Dcp2 controls mRNA abundance and translation to adjust metabolism and filamentation to nutrient availability

Anil Kumar Vijjamarri[1], Xiao Niu[2], Matthew D Vandermeulen[3], Chisom Onu[4], Fan Zhang[1], Hongfang Qiu[1], Neha Gupta[1], Swati Gaikwad[1], Miriam L Greenberg[4], Paul J Cullen[3], Zhenguo Lin[2], Alan G Hinnebusch[1]*

[1]Division of Molecular and Cellular Biology, Eunice Kennedy Shriver National Institute of Child Health and Human Development, Bethesda, United States; [2]Department of Biology, Saint Louis University, St. Louis, United States; [3]Department of Biological Sciences, State University of New York, Buffalo, United States; [4]Department of Biological Sciences, Wayne State University, Detroit, United States

*For correspondence:
ahinnebusch@nih.gov

**Abstract** Degradation of most yeast mRNAs involves decapping by Dcp1/Dcp2. DEAD-box protein Dhh1 has been implicated as an activator of decapping, in coupling codon non-optimality to enhanced degradation, and as a translational repressor, but its functions in cells are incompletely understood. RNA-Seq analyses coupled with CAGE sequencing of all capped mRNAs revealed increased abundance of hundreds of mRNAs in *dcp2Δ* cells that appears to result directly from impaired decapping rather than elevated transcription. Interestingly, only a subset of mRNAs requires Dhh1 for targeting by Dcp2, and also generally requires the other decapping activators Pat1, Edc3, or Scd6; whereas most of the remaining transcripts utilize nonsense-mediated mRNA decay factors for Dcp2-mediated turnover. Neither inefficient translation initiation nor stalled elongation appears to be a major driver of Dhh1-enhanced mRNA degradation. Surprisingly, ribosome profiling revealed that *dcp2Δ* confers widespread changes in relative translational efficiencies (TEs) that generally favor well-translated mRNAs. Because ribosome biogenesis is reduced while capped mRNA abundance is increased by *dcp2Δ*, we propose that an increased ratio of mRNA to ribosomes increases competition among mRNAs for limiting ribosomes to favor efficiently translated mRNAs in *dcp2Δ* cells. Interestingly, genes involved in respiration or utilization of alternative carbon or nitrogen sources are upregulated, and both mitochondrial function and cell filamentation are elevated in *dcp2Δ* cells, suggesting that decapping sculpts gene expression post-transcriptionally to fine-tune metabolic pathways and morphological transitions according to nutrient availability.

## Editor's evaluation

This fundamental study represents a real tour de force, demonstrating the impact of mutation on the mRNA decapping machinery. Accumulation of mRNAs in dcp2 mutants, is dependent both on the classical 5' to 3' pathway of mRNA decay and on the NMD pathway- highlighting the 'non-nonsense' roles of the NMD pathway and how little we really know about the complete set of pathways of mRNA degradation.

## Introduction

The translation and degradation of mRNAs are intertwined, as the 5′ $m^7G$ cap and 3′ poly(A) tail are involved in both mechanisms. In general, translation is initiated by association of initiation factors eIF4E and eIF4G with the capped 5′ end of mRNA followed by recruitment of the small (40S) subunit of the ribosome pre-loaded with other initiation factors and initiator methionyl tRNA. Interaction between eIF4G and poly(A)-binding protein (PABP) stabilizes a 'closed-loop' mRNP conformation that both protects the mRNA from degradation and enhances translation (*Ghosh and Jacobson, 2010*; *Hinnebusch, 2014*). In yeast, degradation is generally initiated by shortening of the poly(A) tail (deadenylation), catalyzed by the Pan2/Pan3 and Ccr4–Not deadenylase complexes. Deadenylation is followed by either 3′ to 5′ exonucleolytic degradation by the cytoplasmic exosome or decapping by the Dcp1/Dcp2 holoenzyme with attendant 5′ to 3′ degradation by the exoribonuclease Xrn1 (*Parker, 2012*). Dcp2-mediated decapping is critical for multiple mRNA decay pathways including bulk 5′ to 3′ decay (*Decker and Parker, 1993*), nonsense-mediated mRNA decay (NMD) triggered by premature termination codons (*He and Jacobson, 2001*), AU-rich mRNA decay (*Barreau et al., 2005*), microRNA-mediated turnover (*Jonas and Izaurralde, 2015*), and transcript-specific degradation (*Badis et al., 2004*).

Dcp2 consists of an N-terminal regulatory domain (NRD) followed by a Nudix superfamily hydrolase domain, attached to an intrinsically disordered C-terminal region (IDR) that contains short leucine-rich helical motifs (HLMs). Dcp1 enhances Dcp2 catalytic activity by interacting with the Dcp2 NRD and is essential for mRNA decapping in vivo (*Beelman et al., 1996*; *Steiger et al., 2003*). Although Dcp1 promotes the closed conformation of Dcp2, the active site is not fully formed and the RNA-binding site is blocked by the NRD, thus forming a catalytically incompetent enzyme complex (*Wurm et al., 2017*). Additional factors stimulate the catalytic activity of the Dcp1/Dcp2 complex, including enhancer of decapping (Edc) proteins. Interaction of Dcp1 with yeast Edc2 (*Borja et al., 2011*) stabilizes the active conformation of the Dcp1/Dcp2 complex (*Wurm et al., 2017*). Autoinhibitory motifs in the IDR interact with the core domain of Dcp2 and stabilize the inactive conformation of Dcp2 (*He and Jacobson, 2015*; *Paquette et al., 2018*). Edc1 alone cannot overcome the inhibitory effect of the IDR, requiring additional stimulation by Edc3 (*Paquette et al., 2018*). The LSm domain of Edc3 binds to HLMs located in the IDR and activates decapping by alleviating autoinhibition and promoting RNA binding by Dcp2 in yeast, and deleting the autoinhibitory region bypasses activation by Edc3 (*He and Jacobson, 2015*; *Paquette et al., 2018*).

Dhh1, Pat1, Scd6, and Lsm1–7 also activate Dcp1/Dcp2 (*Parker, 2012*). Edc3 and Scd6 may act interchangeably in recruiting Dhh1 to the same segment of the Dcp2 IDR (*He et al., 2022*). HLMs in Dcp2 additionally mediate interaction with Pat1 (*Charenton et al., 2017*), although the Pat1/Lsm1–7 complex can also bind to the 3′ ends of deadenylated mRNAs to stimulate decapping (*Tharun et al., 2000*; *Tharun and Parker, 2001*). Numerous contacts were identified among the decapping activators, as Dhh1 interacts with Pat1, Scd6, and Edc3, and Pat1 interacts with Scd6 in addition to Lsm1–7, as well as with Xrn1 (*Nissan et al., 2010*; *Sharif et al., 2013*; *He and Jacobson, 2015*). Interestingly, Dhh1 and the Pat1/Lsm1–7 complex appear to target distinct subsets of mRNAs with overlapping substrate specificities (*He et al., 2018*). From these results and others (*He et al., 2022*), it can be proposed that the decapping activators assemble distinct decapping complexes that target different subsets of mRNAs for degradation. Dhh1, Pat1, Scd6, and Lsm1–7, along with Dcp1/Dcp2, are found concentrated in processing bodies (PBs) with translationally repressed mRNAs (*Parker and Sheth, 2007*), consistent with a role in translational repression in addition to mRNA turnover. Recently, Edc3 and Scd6 were shown to retain Dcp2 in the cytoplasm, preventing import of Dcp1/Dcp2 into the nucleus where it cannot function in cytoplasmic mRNA decay (*Tishinov and Spang, 2021*). Thus, decapping activators can stimulate decapping by multiple, distinct mechanisms.

Dhh1 is a conserved ATP-dependent DEAD-box helicase involved in translation repression and mRNA decay in yeast. Dhh1 and its orthologs in *Schizosaccharomyces pombe* (Ste13), *Caenorhabditis elegans* (CGH-1), *Xenopus laevis* (Xp54), *Drosophila melanogaster* (Me31b), and mammals (RCK/p54) interact with factors involved in deadenylation, decapping and translational repression (*Coller et al., 2001*; *Fischer and Weis, 2002*; *Maillet and Collart, 2002*; *Weston and Sommerville, 2006*). Dhh1 promotes decapping-mediated mRNA turnover, as deadenylated capped mRNAs are stabilized in *dhh1Δ* cells. Moreover, recombinant Dhh1 stimulates the decapping activity of the purified decapping

enzyme (*Coller et al., 2001*; *Fischer and Weis, 2002*). The mRNAs preferentially targeted for decay by Dhh1 or Pat1/Lsm1–7 appear to be inefficiently translated at the elongation stage (*He et al., 2018*).

Although most yeast mRNAs are targeted by a common degradation pathway, the half-lives of individual mRNAs vary greatly (*Coller and Parker, 2004*). Specific sequence and structural elements present in 5′ or 3′ UTRs, and poly(A) tail lengths, can modulate the degradation rate; however, these features alone do not explain the variation in half-lives among all transcripts (*Muhlrad and Parker, 1992*; *Lee et al., 2013*; *Geisberg et al., 2014*). Recently, codon optimality—the balance between the supply of charged tRNA molecules in the cytoplasmic pool and the demand of tRNA usage by translating ribosomes—was identified as a determinant of mRNA decay. Evidence indicates that non-optimal codons, which are decoded by ribosomes more slowly than optimal codons, accelerate mRNA decay co-translationally (*Hu et al., 2009*; *Presnyak et al., 2015*). Multiple lines of evidence support the model that Dhh1 binds to ribosomes elongating slowly through non-optimum codons to trigger decapping and degradation (*Radhakrishnan et al., 2016*). Recently, cryo-EM analysis revealed that the N-terminal region of Not5 interacts with the E-site of ribosomes stalled at suboptimal codons with an empty A-site, leading to the model that ribosomes stalled at suboptimal codons are detected by Not5/Caf1 and targeted for degradation via Dhh1 (*Buschauer et al., 2020*). Much of the evidence indicating that codon non-optimality is a major determinant of mRNA instability derives from reporter transcripts with relatively long strings of non-optimal or optimal codons, which likely does not pertain to many native mRNAs.

A different model proposes that translation initiation and decay are inversely related, with functional initiation complexes impeding both decapping and deadenylation. Thus, mutations impairing the cap-binding eIF4F complex or initiation factor eIF3 that reduce translation initiation accelerate degradation of particular yeast mRNAs via elevated deadenylation and decapping (*Schwartz and Parker, 1999*). Strong secondary structures in the 5′ UTR (*Muhlrad et al., 1995*) and poor start codon context (*LaGrandeur and Parker, 1999*) increased decapping. Moreover, binding of eIF4E to the cap structure was sufficient to inhibit decapping by Dcp1/Dcp2 (*Schwartz and Parker, 2000*). From these findings it can be predicted that translation initiation, directly or indirectly, competes with the decay machinery in a manner that influences mRNA turnover rates. Indeed, non-invasive measurements of mRNA half-lives suggested that competition between translation initiation and mRNA decay factors is a major determinant of yeast mRNA turnover, whereas global inhibition of elongation generally led to stabilization vs. degradation of transcripts (*Chan et al., 2018*).

Starving yeast for glucose engenders a rapid loss of protein synthesis, accompanied by a shift of mRNAs from polysomes to free mRNPs and an increase in both size and number of PBs (*Ashe et al., 2000*; *Teixeira et al., 2005*; *Arribere et al., 2011*). Deletion of *DHH1* or *PAT1* partially impaired the loss of polysomes evoked by glucose depletion, whereas deletion of both genes simultaneously abrogated translational repression, indicating that both Dhh1 and Pat1 are required for repression of translation initiation during glucose starvation (*Holmes et al., 2004*; *Coller and Parker, 2005*). Furthermore, overexpression of Dhh1 or Pat1 in wild-type (WT) cells conferred a general repression of translation with attendant PB formation (*Coller and Parker, 2005*). These findings suggested that Dhh1 and Pat1 act as general repressors of translation in glucose-starved yeast. Consistent with this, Dhh1 can inhibit translation initiation in vitro by blocking assembly of 48S preinitiation complexes (PICs) (*Coller and Parker, 2005*). Alternatively, the recovery of polysomes during stress in *dhh1Δ* and *pat1Δ* mutants could result indirectly from a broad stabilization of mRNAs that helps to restore 48 S PIC assembly by mass action and overcome inhibition of PIC formation during stress produced by other means (*Holmes et al., 2004*). Indeed, impairing the functions of RNA helicases eIF4A and Ded1, which normally stimulate 48S PIC assembly, is involved in suppressing translation during stress (*Castelli et al., 2011*; *Bresson et al., 2020*; *Iserman et al., 2020*; *Sen et al., 2021*). Other evidence indicts that Dhh1 can directly impede translation elongation by associating with slowly moving ribosomes (*Coller and Parker, 2005*; *Sweet et al., 2012*). Consistent with this, the Not4 subunit of Ccr4–Not complex, as well as Dhh1 and Dcp1, were implicated in translational repression of mRNAs that exhibit transient ribosome stalling in a manner that limits protein misfolding during nutrient limitation (*Preissler et al., 2015*). Together, these findings suggest that multiple factors that stimulate mRNA decay can also repress translation at the initiation or elongation steps during nutrient starvation. Interestingly, Dhh1 helicase activity was reported to enhance, not repress, translation of *ATG1* and *ATG13* mRNAs required for autophagy during nitrogen starvation (*Liu et al., 2019*). Ribosome profiling of

*dhh1Δ* cells grown in rich medium uncovered hundreds of mRNAs displaying either increased or decreased relative TEs, suggesting that Dhh1 can function as a translational repressor or activator for distinct sets of mRNAs, even in nonstarvation conditions (*Radhakrishnan et al., 2016*; *Jungfleisch et al., 2017*; *Zeidan et al., 2018*).

Based on its central role in mRNA decapping, most studies on Dcp2 have focused primarily on mRNA decay, and the participation of Dcp2 in regulating translation is poorly understood. Here, we demonstrate that in nutrient-replete cells Dcp2 modulates translation in addition to mRNA turnover by pathways both dependent and independent of decapping activator Dhh1. Surprisingly, a large fraction of mRNAs appear to be targeted for degradation by Dcp2 independently of Dhh1, Pat1, Edc3, or Scd6, which are enriched for NMD substrates; whereas the remaining fraction is controlled concurrently by all four decapping activators. Codon non-optimality does not appear to be a major driver of mRNA degradation for either set of Dcp2-targeted mRNAs. Unexpectedly, ribosome profiling revealed that *dcp2Δ* confers widespread TE changes that generally favor mRNAs well translated in WT cells at the expense of more poorly translated mRNAs. This competition can be attributed to increased mRNA abundance, resulting from impaired decapping/decay, coupled with reduced ribosome production. Superimposed on this general reprogramming is a Dhh1-dependent translational repression of certain poorly translated mRNAs by Dcp2. Finally, we provide evidence that Dcp2 helps to repress the abundance or translation of many mRNAs encoding proteins whose functions are dispensable during growth on glucose-replete rich medium, involved in catabolism of non-preferred carbon or nitrogen sources, mitochondrial respiration, or cell filamentation and invasive growth. These findings support the emerging model that regulators of mRNA turnover add a layer of post-transcriptional control to well-established transcriptional repression mechanisms that control various responses to nutrient limitation.

## Results

## Dcp2 controls mRNA abundance via both Dhh1-dependent and -independent pathways

To determine the role of Dcp2 and Dhh1 in regulating mRNA abundance and translation in nutrient-replete cells, we interrogated our previous ribosome profiling and RNA-Seq datasets obtained from isogenic WT, *dcp2Δ*, *dhh1Δ* and *dcp2Δdhh1Δ* strains cultured in nutrient-rich YPD medium at 30°C (*Zeidan et al., 2018*). RNA-Seq analysis identified numerous mRNAs differentially expressed in *dcp2Δ* vs. WT cells, including 1376 up-regulated mRNAs (*Figure 1A*, blue) exhibiting a median increase of 1.96-fold (*Figure 1B*, mRNA_up_*dcp2Δ*), and 1281 down-regulated transcripts (*Figure 1A*, red) showing a median decrease of 0.57-fold in mRNA abundance conferred by *dcp2Δ* (*Figure 1B*, mRNA_dn_*dcp2Δ*). (Note in *Figure 1B* that the median change for all mRNAs is unity ($\log_2 = 0$) owing to normalization of RNA reads for library depth for both mutant and WT strains. In this and all subsequent box-plots, if the notches in two boxes do not overlap, their median values differ with 95% confidence.) We performed quantitative Reverse Transcription Polymerase Chain Reaction (qRT-PCR) analysis of selected up- or down-regulated transcripts in total mRNA from *dcp2Δ* and WT cells, normalizing their abundance to a luciferase mRNA spiked into all samples, and found a strong positive correlation between the mRNA changes identified in RNA-Seq vs. qRT-PCR (Spearman's correlation coefficient $\rho$ = 0.86) (*Figure 1C* and *Figure 1—figure supplement 1A*).

To identify the importance of Dcp2 catalytic activity in the mRNA changes conferred by *dcp2Δ*, we conducted RNA-Seq on an isogenic *dcp2-E149Q,E153Q* mutant (*dcp2-EE*) with Gln substitutions of two conserved Glu residues in the catalytic domain expected to abolish decapping (*Aglietti et al., 2013*; *He et al., 2018*). The results for the WT, *dcp2Δ*, and *dcp2-EE* strains showed high reproducibility among biological replicates ($\rho$ = 0.99) (*Figure 1—figure supplement 2A–C*) and the mRNA densities for all expressed genes were also highly correlated between the *dcp2Δ* vs. *dcp2-EE* strains ($\rho$ = 0.99, *Figure 1—figure supplement 2G*), with only one mRNA (*YOR178C*) showing a significant difference (*Figure 1D*). Consistent with these results, the mRNA changes conferred by the catalytically defective *dcp2-E153Q-N245* and *dcp2-E198Q-N245* alleles (encoding only the N-terminal 245 residues of Dcp2) were found previously to be highly correlated with those conferred by *dcp2Δ* (coefficients of 0.7–0.8) (*He et al., 2018*). Together these results indicate that the decapping activity of Dcp2 mediates its effects on the majority of yeast mRNAs whose abundance is controlled by Dcp2 in vivo.

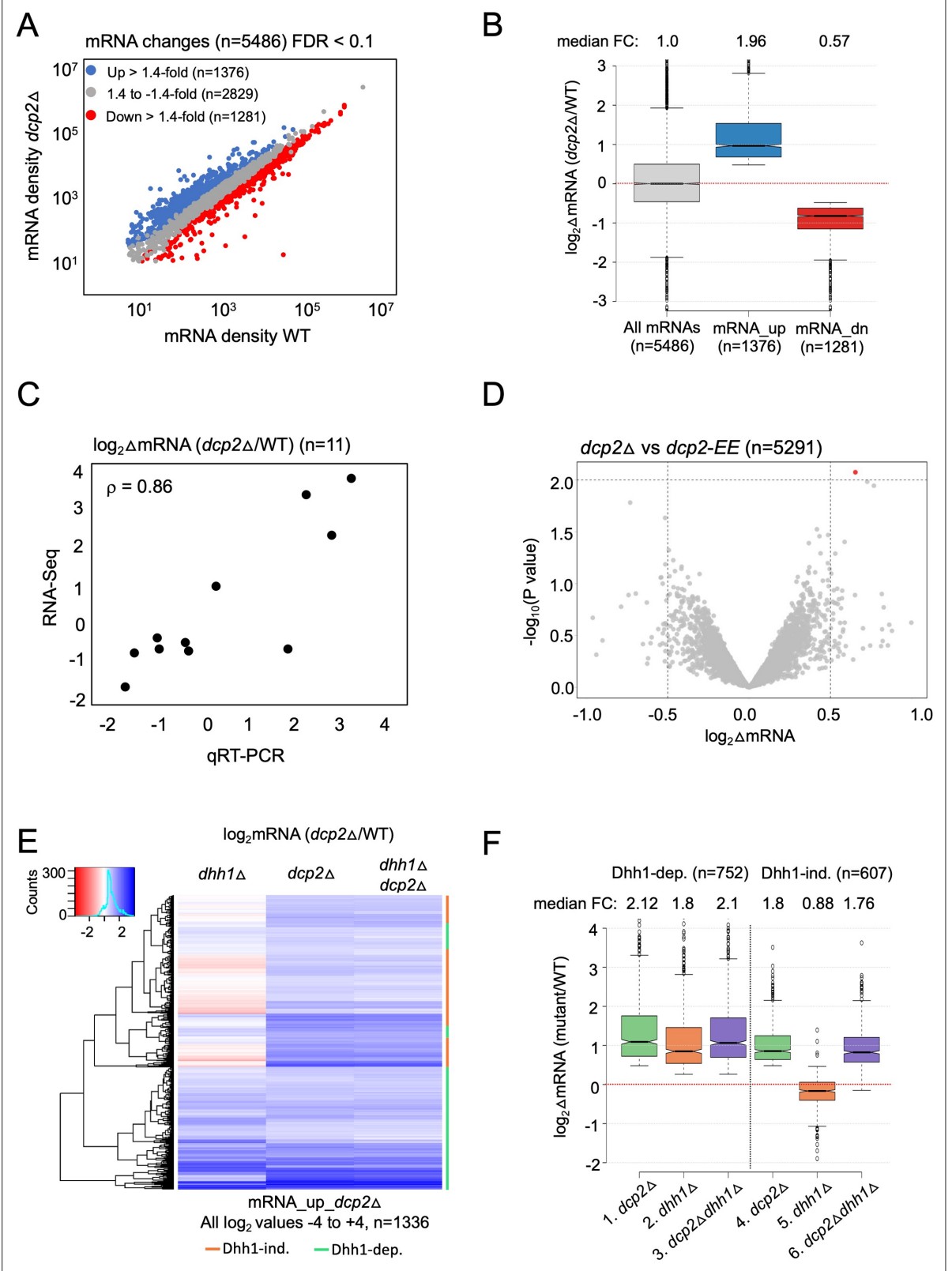

**Figure 1.** Dhh1-dependent and -independent regulation of mRNA abundance by decapping factor Dcp2. (**A**) Scatterplot of normalized mRNA densities in RPKM (Reads Per Kilobase of transcript, per Million mapped reads) for 5486 genes in *dcp2Δ* vs. WT cells determined by RNA-Seq. Genes showing significant changes in mRNA abundance of >1.4-fold in *dcp2Δ* vs. WT cells at false discovery rate (FDR) <0.1 as determined by DESeq2 analysis are shown as blue or red dots. (**B**) Notched box-plot of the log₂ fold-changes in mRNA abundance (from RNA-Seq) in *dcp2Δ* vs. WT cells for all mRNAs (*n* =

*Figure 1 continued on next page*

*Figure 1 continued*

5486, median FC = 1) or subsets of transcripts either increased (*n* = 1376, median FC = 1.96) or decreased (*n* = 1281, median FC = 0.57). Median values for each group are indicated at the top, and the numbers of mRNAs for which data were obtained for each group is indicated at the bottom. A few outliers (*n* = 34 for mRNA_up and *n* = 44 for mRNA_dn groups) were omitted to expand the *y*-axis scale. (**C**) Scatterplot showing correlation between the log$_2$ fold-changes in mRNA abundance determined by RNA-Seq vs. qRT-PCR for 11 different genes in *dcp2Δ* vs. WT cells, with the Spearman correlation coefficient (*ρ*) indicated. (**D**) Evidence that mRNA changes conferred by *dcp2Δ* result from loss of Dcp2 catalytic activity. A Volcano plot showing log$_2$ fold-changes in mRNA abundance (from RNA-Seq) in *dcp2Δ* vs. *dcp2-E149Q,E153Q* (*dcp2-EE*) (x-axis) vs. −log$_{10}$ p values for the mRNA changes (y-axis) cells for all mRNAs (*n* = 5291). (**E**) Hierarchical clustering analysis of the log$_2$ fold-changes in mRNA abundance conferred by *dhh1Δ*, *dcp2Δ*, or *dhh1Δdcp2Δ* vs. WT for 1336 of the mRNA_up_*dcp2Δ* transcripts that are increased in abundance in *dcp2Δ* vs. WT cells. Transcripts annotated on the right with green or red bars require Dhh1 or are independent of Dhh1, respectively, for their repression by Dcp2 in WT cells. The color scale indicating log$_2$ΔmRNA values ranges from 4 (strong derepression, dark blue) to −4 (repression, dark red). A few outlier mRNAs (*n* = 40) with log$_2$ΔmRNA values of >4.0 or <−4.0 were excluded to enhance the color differences among the remaining mRNAs. (**F**) Notched box-plot of log$_2$ΔmRNA values in the mutants indicated at the bottom vs. WT for the two sets of mRNAs decreased in abundance by Dcp2 in a manner dependent (cols. 1–3) or independent of Dhh1 (cols. 4–6). Note that 17 of the 1376 mRNA_up_*dcp2Δ* transcripts were not detected in the *dhh1Δ* vs. WT experiment and, hence, were excluded from consideration. A few outliers (*n* = 8 from columns 1 and 3, and *n* = 14 from column 2) were omitted from the plots to expand the *y*-axis scale. Horizontal red dotted lines indicate log$_2$ΔmRNA values of zero, for a median fold-change of 1.

The online version of this article includes the following source data and figure supplement(s) for figure 1:

**Source data 1.** log$_2$ fold-changes in mRNA abundance observed by RNA-Seq analysis in *dcp2Δ* vs. WT cells for all transcripts, mRNA_up_*dcp2Δ*, and mRNA_dn_*dcp2Δ* groups (***Figure 1A, B***).

**Source data 2.** Comparison of qRT-PCR and RNA-Seq analyses of mRNA changes for 11 selected genes observed in *dcp2Δ* vs. WT cells (***Figure 1C***).

**Source data 3.** log$_2$ fold-changes in mRNA abundance observed by RNA-Seq analysis in *dcp2Δ* vs. *dcp2Δ-EE* cells for all transcripts (***Figure 1D***).

**Source data 4.** log$_2$ fold-changes in mRNA abundance observed by RNA-Seq analysis in *dcp2Δ*, *dhh1Δ*, and *dcp2Δdhh1Δ* mutants vs. WT for all transcripts, Dhh1-dependent mRNA_up_*dcp2Δ* and Dhh1-independent mRNA_up_*dcp2Δ* groups of transcripts (***Figure 1E, F***).

**Figure supplement 1.** Supporting information that Dcp2 regulates mRNA abundance in a manner dependent or independent of Dhh1.

**Figure supplement 1—source data 1.** log$_2$ fold-changes in mRNA observed by RNA-Seq analysis in *dhh1Δ* vs. WT cells for all transcripts from three published datasets (***Figure 1—figure supplement 1B***).

**Figure supplement 2.** Reproducibility between biological replicates of ribosome footprint profiling and RNA-Seq analyses for WT, *dcp2Δ*, and *dcp2-EE* strains.

**Figure supplement 2—source data 1.** RPKM-normalized reads from RNA-Seq and Ribo-Seq for biological replicates of WT, *dcp2Δ*, and *dcp2-EE* cells (***Figure 1—figure supplement 2A–F***); normalized mRNA densities for all transcripts in *dcp2Δ* vs. *dcp2-EE* cells (***Figure 1—figure supplement 2G***).

To evaluate the contribution of decapping activator Dhh1 in controlling mRNA abundance by Dcp2, we compared the mRNA expression changes conferred by *dcp2Δ*, *dhh1Δ*, and the *dcp2Δdhh1Δ* double mutation for the aforementioned 1376 transcripts increased by the *dcp2Δ* single mutation. A subset of 752 transcripts (dubbed Dhh1-dependent) show similar derepression in all three mutants (***Figure 1E***, green bars), with median increases in abundance compared to WT of 1.8- to 2.1-fold in the three mutants (***Figure 1F***, cols. 1–3). That combining *dcp2Δ* and *dhh1Δ* confers similar derepression ratios as those given by each single mutation is to be expected if Dhh1 and Dcp2 function in the same pathway to control the abundance of these mRNAs, with Dhh1 activating decapping by Dcp2 and attendant 5′ to 3′ degradation. In contrast, the remaining 607 transcripts in the mRNA_up_*dcp2Δ* group showed little change in abundance in the *dhh1Δ* single mutant, despite derepression in the *dcp2Δ* mutant *on par* with that observed for the Dhh1-dependent group (***Figure 1E***, red bars). Whereas *dcp2Δ* conferred a median increase of 1.8-fold, *dhh1Δ* produced a small repression of 0.88-fold for these last mRNAs compared to WT (***Figure 1F***, cols. 4–5). As expected if these mRNAs are decapped independently of Dhh1, they display essentially the same increases in response to the *dcp2Δ* and *dhh1Δdcp2Δ* mutations vs. WT (***Figure 1E***, cols. 2–3; ***Figure 1F***, cols. 4 and 6). Supporting our analysis above, we interrogated RNA-Seq data for *dhh1Δ* vs. WT comparisons from three published studies, two of which involved a different strain background from ours (***Radhakrishnan et al., 2016***; ***Jungfleisch et al., 2017***; ***He et al., 2018***). Importantly, the transcripts increased by *dcp2Δ* in a Dhh1-dependent manner in our study showed significant increases in median mRNA abundance in *dhh1Δ* vs. WT comparisons in all three datasets (***Figure 1—figure supplement 1B***, cols. 5–8); whereas our Dhh1-independent group of transcripts showed little change or reduced median abundance in the published datasets (***Figure 1—figure supplement 1B***, cols. 9–12). Together, the results suggest that only about one-half of the mRNAs repressed in abundance by Dcp2 require Dhh1 for this repression.

# Multiple decapping activators target a common subset of transcripts for Dcp2-mediated repression of mRNA abundance

To evaluate whether Dhh1-independent degradation by Dcp2 involves other decapping activators that function in place of Dhh1, we evaluated RNA-Seq data we obtained recently comparing isogenic *pat1Δ*, *pat1Δdhh1Δ*, or *edc3Δscd6Δ* mutants to WT, as well as published RNA-Seq data on an isogenic *upf1Δ* mutant lacking a key NMD factor (*Celik et al., 2017*). Interestingly, *pat1Δ*, *pat1Δdhh1Δ*, and *edc3Δscd6Δ* mutants exhibit increased median abundance for the Dhh1-dependent mRNA_up_*dcp2Δ* transcripts, as observed for the *dhh1Δ* mutant (*Figure 2A*, cols. 2–5). Consistent with this, cluster analysis of individual transcripts reveals increases of similar magnitude in the four decapping activator mutants and the *dcp2Δ* strain for many of the Dhh1-dependent transcripts (*Figure 2B*); although, there are numerous transcripts increased to a greater or lesser extent in the *pat1Δ*, *pat1Δdhh1Δ*, or *edc3Δscd6Δ* mutants compared to the *dhh1Δ* single mutant. (An in depth analysis of this transcript specificity is being presented elsewhere.) In contrast, *upf1Δ* did not increase the median mRNA abundance of this group of transcripts (*Figure 2A*, col. 6) and cluster analysis revealed that *upf1Δ* increased only a small subset of the Dhh1-dependent transcripts elevated in the other mutants (*Figure 2B*). Thus, the mRNAs dependent on Dhh1 for repression by Dcp2 generally also require Pat1 and/or Edc3/Scd6, but not Upf1, for full repression in WT cells.

A different outcome was observed for the Dhh1-independent group of mRNA_up_*dcp2Δ* transcripts, which showed no increase in median abundance in the *pat1Δ*, *pat1Δdhh1Δ*, or *edc3Δscd6Δ* mutants, as observed for *dhh1Δ* (*Figure 2A*, cols. 8–11). Cluster analysis confirmed that most transcripts in this group showed little change in abundance in all four of these decapping activator mutants; although, a subset of exceptional mRNAs were increased in the single and double mutants lacking *PAT1* (*Figure 2C*). Thus, the majority of Dhh1-independent mRNAs also appear to be independent of Pat1, Scd6, or Edc3 for their repression by Dcp2. The fact that expression of these mRNAs is generally unaffected even in the *pat1Δdhh1Δ* and *edc3Δscd6Δ* double mutants eliminates the possibility that they exhibit redundant requirements for either Edc3 or Scd6, or either Pat1 or Dhh1. Interestingly, *upf1Δ* increased the median abundance of this group of transcripts comparably to *dcp2Δ* (*Figure 2A*, cols. 12 vs. 7) and up-regulated a much larger fraction vs. any of the other decapping activator mutations (*Figure 2C*). All three mutants lacking NMD factors Upf1, Upf2, or Upf3 exhibited similar marked derepression of the Dhh1-independent transcripts with little effect on the Dhh1-dependent group (*Figure 2—figure supplement 1*). These results are significant in revealing two groups of mRNAs whose abundance is repressed by Dcp2 in a manner that (1) generally requires the concerted action of Dhh1, Pat1, and Edc3/Scd6 (Dhh1-dependent group) or (2) frequently requires the NMD factors but none of the other four decapping activators, nor even one of the Edc3/Scd6 and Dhh1/Pat1 pairs of activators, for repression by Dcp2 (Dhh1-independent group).

As noted above, previous findings have suggested that a key determinant of decapping and mRNA decay is the rate of translation at either the initiation or elongation stages (*Chan et al., 2018*; *Hanson et al., 2018*). In particular, Dhh1 has been implicated in targeting mRNAs enriched for slowly decoded non-optimal codons for decapping and decay (*Radhakrishnan et al., 2016*). Consistent with previous findings (*He et al., 2018*), we observed that the entire group of transcripts increased in abundance by *dcp2Δ* (mRNA_up_*dcp2Δ*) exhibit lower than average TEs in WT cells, as determined by ribosome profiling of our *dcp2Δ* and WT strains conducted in parallel with RNA-Seq (*Zeidan et al., 2018*; *Figure 2D*, col. 2). However, these mRNAs have only a slightly lower species-specific tRNA adaptation index (stAI) (*Figure 2E*, col. 2), a measure of codon optimality that quantifies the relative cellular supply of cognate and near-cognate tRNAs for a given codon (*Radhakrishnan et al., 2016*), suggesting that their lower TEs generally reflect slower rates of initiation vs. elongation. Importantly, the Dhh1-dependent subset of Dcp2-repressed mRNAs exhibits average TEs and stAI values (*Figure 2D, E*, col. 3). We found that the mRNAs elevated by *dhh1Δ* in two other studies (*Radhakrishnan et al., 2016*; *Jungfleisch et al., 2017*) similarly exhibit average median TEs and stAIs. Previously, we and others observed a moderate inverse correlation between stAI values and changes in mRNA abundance in *dhh1Δ* vs. WT cells for all expressed mRNAs (*Radhakrishnan et al., 2016*; *Zeidan et al., 2018*), consistent with the model that Dhh1 targets mRNAs occupied by elongating ribosomes paused at slowly decoded codons. Our findings here that the mRNAs repressed most strongly by Dcp2 exhibit average codon optimality might be explained if these mRNAs contain short runs of suboptimal codons that trigger Dhh1-dependent decapping despite an average frequency of poor codons; alternatively, other

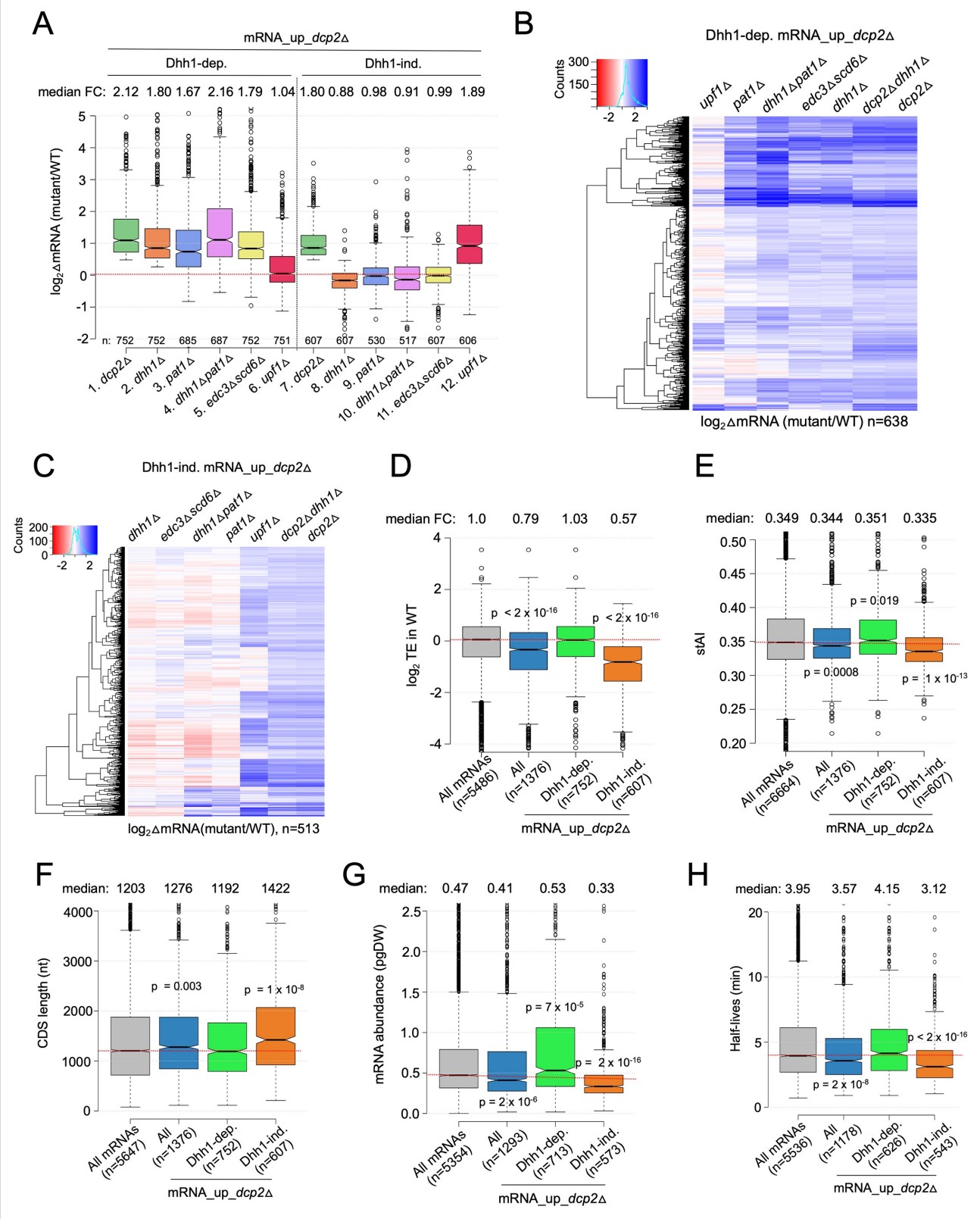

**Figure 2.** Multiple decapping activators function in unison to down-regulate a subset of Dcp2-repressed mRNAs. (**A**) Notched box-plot as in *Figure 1B* showing log₂fold-changes in mRNA abundance in the mutants indicated at the bottom vs. WT for the Dhh1-dependent (cols. 1–6) or Dhh1-independent (cols. 7–12) subsets of the mRNA_up_*dcp2*Δ set of transcripts. (**B, C**) Hierarchical clustering analyses as in *Figure 1E* of the log₂ fold-changes in mRNA abundance conferred by the mutations listed across the top vs. WT for the Dhh1-dependent (panel B, *n* = 638) or Dhh1-independent (panel C, *n* =

*Figure 2 continued on next page*

*Figure 2 continued*

513) subset of the mRNA_up_dcp2Δ transcripts (excluding a few outliers [n = 36] with log₂ΔmRNA values >4 or <−4). Notched box-plot as in **Figure 1B** showing the log₂TE in WT cells (D), species-specific tRNA adaptation index (stAI) values (E), coding sequence (CDS) length (F), mRNA abundance expressed as pg per dry cellular weight (G) and half-lives (H) for All mRNAs, all mRNA_up_dcp2Δ transcripts, or the Dhh1-dependent or -independent subsets of mRNA_up_dcp2Δ transcripts. p-values calculated using the Mann–Whitney *U* test for the differences between all mRNAs and the indicated groups are shown. For (D), WT translational efficiency (TE) values were calculated as the ratio of mRNA reads protected from RNAse digestion by association with translating 80S ribosomes (RPFs) to the total mRNA reads measured by RNA-Seq for the same transcript in WT cells.

The online version of this article includes the following source data and figure supplement(s) for figure 2:

**Source data 1.** log₂ fold-changes in mRNA abundance observed by RNA-Seq analysis in *pat1Δ*, *dhh1Δpat1Δ*, *edc3scd6Δ*, and *upf1Δ* cells relative to WT for all transcripts (**Figure 2A–C**).

**Source data 2.** mRNA properties for all transcripts including TE in WT cells, species-specific tRNA adaptation index (stAI), CDS lengths, half-lives, and mRNA abundance (**Figure 2D–H**).

**Figure supplement 1.** All three deletion strains lacking a Upf factor exhibit similar changes in mRNA abundances for the two groups of transcripts regulated by Dcp2.

**Figure supplement 1—source data 1.** log₂ fold-changes in mRNA abundance observed by RNA-Seq analysis in *upf1Δ*, *upf2Δ*, and *upf3Δ* cells vs. WT for all transcripts (**Figure 2—figure supplement 1**).

properties besides codon optimality could be more important in dictating preferential degradation by Dcp2/Dhh1.

Consistent with previous results on NMD substrates (**Celik et al., 2017**), the Dhh1-independent transcripts exhibit lower median TE and stAI values compared to all Dcp2-repressed mRNAs (**Figure 2D, E**, cols. 4 vs. 2). They also display other features of poorly translated mRNAs (**Pelechano et al., 2013**; **Radhakrishnan et al., 2016**; **Lahtvee et al., 2017**; **Chan et al., 2018**), including a greater median length of coding sequences (CDSs), lower median transcript abundance in WT cells, and lower median mRNA half-life, compared to all mRNAs and to all Dcp2-repressed mRNAs, none of which are characteristic of the Dhh1-dependent group of transcripts (**Figure 2F–H**, cols. 3–4 vs. 1–2). It is unclear whether any of these attributes of Dhh1-independent mRNAs are instrumental in enhancing decapping/degradation, or in targeting NMD factors to the Upf1-dependent members of the group; however, it has been suggested that low codon optimality may increase frameshifting errors that lead to premature termination events recognized by the Upf proteins (**Celik et al., 2017**).

## Both Dhh1-dependent and -independent repression of mRNA abundance by Dcp2 appears to involve decapping and degradation rather than reduced transcription

It was shown previously that *dcp2Δ* and many other slow-growing yeast mutants exhibit changes in gene expression (**O'Duibhir et al., 2014**) that correspond to the Environmental Stress Response (ESR), wherein hundreds of mRNAs are either repressed (rESR, 545 genes) or induced (iESR, 283 genes) stereotypically in response to diverse stress or starvation conditions (**Gasch et al., 2000**). Induction of iESR genes involves transcription factors Msn2 and Msn4 that bind stress response elements (STREs) in promoters (**Görner et al., 1998**). Most rESR gene products are involved in ribosome biogenesis or translation. Consistent with the previous findings, gene ontology (GO) analysis of the mRNAs dysregulated by *dcp2Δ* identified stress response genes for the mRNA_up_dcp2Δ group, and genes involved in ribosome production and translation for the mRNA_dn_dcp2Δ group of genes (**Figure 7—source data 1A, B**). Accordingly, it was possible that a large fraction of the mRNAs increased by *dcp2Δ* are induced at the transcriptional level as a manifestation of the iESR. At odds with this idea however, the Dhh1-independent mRNA_up_dcp2Δ transcripts are ~twofold underrepresented in iESR mRNAs (p = 5.3 × 10⁻³, **Figure 3B**); and although the Dhh1-dependent group is significantly overrepresented for iESR mRNAs (p = 1.9 × 10⁻⁹⁰, **Figure 3A**), 76% are not iESR transcripts. Thus, the bulk of mRNA increases conferred by *dcp2Δ* does not involve the ESR.

Because eliminating Dcp2 might provoke other transcriptional responses besides the ESR, we sought additional evidence that reduced mRNA degradation owing to diminished decapping is a major driver of increased transcript levels in *dcp2Δ* cells. Recent findings indicate that many transcripts decapped by Dcp1/Dcp2 undergo 5′ to 3′ degradation while associated with translating ribosomes (**Hu et al., 2009**), with Xrn1 following the last translating ribosomes loaded on mRNA prior

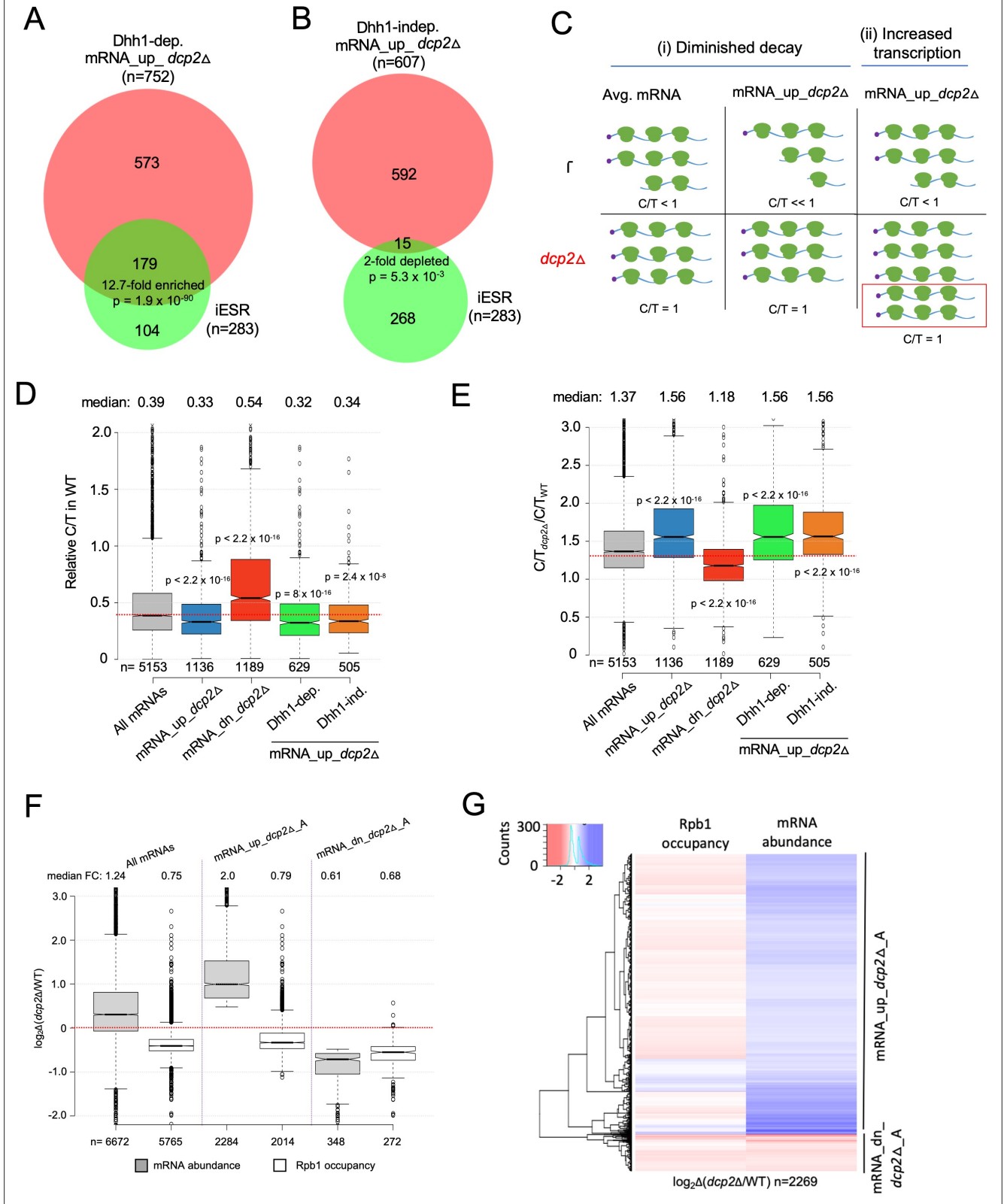

**Figure 3.** Evidence that Dhh1-independent changes in mRNA abundance conferred by *dcp2Δ* do not result from altered transcription. (**A, B**) The Dhh1-independent subset of Dcp2-repressed mRNAs are not enriched for iESR transcripts. Proportional Venn diagrams showing overlap between the Dhh1-dependent (**A**) and -independent (**B**) subsets of mRNA_up_*dcp2Δ* transcripts with induced ESR (iESR) mRNAs. Hypergeometric distribution p-values are displayed for significant changes. (**C**) Schematics depicting the predicted effects of *dcp2Δ* on levels of the capped proportion of mRNAs for

*Figure 3 continued on next page*

*Figure 3 continued*

the mRNAs preferentially repressed by Dcp2 compared to the average mRNA, according to the two different derepression mechanisms of (**i**) diminished decapping/degradation, and (**ii**) increased transcription. *Mechanism* (**i**). In WT cells (upper panel), mRNAs preferentially targeted by Dcp2 for decapping and degradation (mRNA_up_*dcp2Δ*) have a smaller proportion of capped transcripts (C/T<<1) compared to mRNAs with average susceptibility to Dcp2 (Avg. mRNA, C/T<1). In *dcp2Δ* cells (lower panel, red), the C/T ratios for both groups of mRNAs increase to unity, which confers a relatively larger increase in C/T ratio in *dcp2Δ* vs. WT cells for the mRNA_up_*dcp2Δ* group. *Mechanism* (**ii**). The mRNA_up_*dcp2Δ* group is preferentially induced at the transcriptional level and thus resembles the average mRNA both in C/T ratio in WT cells and the increase in C/T ratio in *dcp2Δ* vs. WT cells. The red box depicts the increase in number of transcripts in *dcp2Δ* by this mechanism. (**D, E**) Dcp2-repressed mRNAs exhibit greater than average decapping in WT cells that is reversed by *dcp2Δ*. Notched box-plots of ratios of capped mRNA transcripts per million (TPMs) to total mRNA TPMs (C/T) in WT cells (**D**) or in *dcp2Δ* relative to WT cells (**E**) for all mRNAs (gray) or the following sets of mRNAs: mRNA_up_*dcp2Δ* (blue), mRNA_dn_*dcp2Δ* (red), Dhh1-dependent mRNA_up_*dcp2Δ* (green), and Dhh1-independent mRNA_up_*dcp2Δ* (orange). (**F, G**) Quantification of absolute Rpb1 occupancies and mRNA abundance by spike-in normalization reveals reduced Pol II occupancies of most genes showing mRNA derepression in *dcp2Δ* cells. (**F**) Notched box-plot analysis of changes in Rpb1 occupancies or mRNA abundance in *dcp2Δ* vs. WT cells for the mRNA_up_*dcp2Δ*_A and mRNA_dn_*dcp2Δ*_A groups identified by ERCC-normalized RNA-Seq for all mRNAs (cols. 1–2), mRNA_up_*dcp2Δ*_A (cols. 3–4), and mRNA_dn_*dcp2Δ*_A (cols. 5–6). (**G**) Hierarchical clustering analysis of the same data from (**F**) (excluding a few outliers [*n* = 16] with log₂Δ values >+4 or <−4).

The online version of this article includes the following source data and figure supplement(s) for figure 3:

**Source data 1.** List of iESR and rESR transcripts (*Figure 3A, B*).

**Source data 2.** Relative capped to total mRNA ratios (C/T) for all mRNAs in WT, *dcp2Δ*, *xrn1Δ*, *dcp2Δ* cells vs. WT and in *xrn1Δ* vs. WT (*Figure 3D, E*).

**Source data 3.** Calculation of size factors from *S. pombe* Rpb1 reads, and spike-in normalized *S. cerevisiae* Rpb1 reads obtained from Rpb1 ChIP-Seq.

**Figure supplement 1.** Reproducibility between biological replicates of CAGE-Seq and parallel RNA-Seq analyses for WT, *dcp2Δ*, and *xrn1Δ* strains.

**Figure supplement 1—source data 1.** Statistics for CAGE-Seq and parallel RNA-Seq data analysis.

**Figure supplement 2.** mRNAs increased in abundance by *dcp2Δ* or *xrn1Δ* differ in accumulating as capped (*dcp2Δ*) or uncapped (*xrn1Δ*) species.

**Figure supplement 2—source data 1.** log₂ fold-changes in capped mRNAs (CAGE-Seq) and total mRNA (RNA-Seq) for all transcripts in *dcp2Δ* vs. WT and *xrn1Δ* vs. WT cells (*Figure 3—figure supplement 2A, B*).

**Figure supplement 3.** Supporting evidence that mRNAs increased in abundance by *dcp2Δ* exhibit greater than average levels of both decapping and degradation of uncapped intermediates by Xrn1.

**Figure supplement 4.** Analyses of codon-protection index (CPI) values and Rpb1 occupancies determined by ChIP-Seq indicate that mRNA derepression by *dcp2Δ* results primarily from loss of decapping-mediated cotranslational mRNA decay not increased transcription.

**Figure supplement 4—source data 1.** CPI values for all mRNAs (*Figure 3—figure supplement 4A*).

**Figure supplement 4—source data 2.** List of transcripts showing ratios of absolute mRNA abundance to absolute Rpb1 occupancy of 0.8–1.2 (*Figure 3—figure supplement 4C*).

to decapping (*Pelechano et al., 2015*). Such decapped degradation intermediates may account for ~12% of the polyadenylated mRNA population in WT yeast (*Pelechano et al., 2013*). We reasoned that mRNAs targeted preferentially for decapping and degradation by Dcp2 should be enriched for such decapped degradation intermediates and hence exhibit a greater than average proportion of uncapped transcripts in WT cells, which will be eliminated in the *dcp2Δ* mutant. In contrast, mRNAs that are up-regulated indirectly as the result of increased transcription in *dcp2Δ* cells should exhibit an average proportion of uncapped isoforms in WT. These predictions are illustrated in *Figure 3C* for an idealized scenario of enhanced mRNA decapping/degradation (model (i)) vs. elevated transcription (model (ii)). To evaluate these predictions, we measured the abundance of capped isoforms for all expressed transcripts by CAGE (cap analysis of gene expression), using a revised no-amplification-nontagging technique (nAnT-iCAGE) (*Murata et al., 2014*), and subjected the same RNA samples from WT and *dcp2Δ* cells to standard RNA-Seq. We calculated the ratios of capped transcripts per million (TPM) from nAnT-iCAGE to the total TPMs calculated from RNA-Seq (dubbed C/T) as a proxy for the fraction of capped transcripts for each gene in the two strains. (Because the nAnT-iCAGE and RNA-Seq data were normalized individually, the C/T ratios are relative rather than actual proportions of capped transcripts). If increased mRNA levels in *dcp2Δ* cells result from impaired decapping/decay vs. increased transcription, we expect to observe lower than average C/T ratios in WT cells for the mRNA_up_*dcp2Δ* transcripts up-regulated by *dcp2Δ*. To validate this approach, we similarly compared WT and isogenic *xrn1Δ* cells, reasoning that eliminating Xrn1-mediated 5′ to 3′ degradation should lead to accumulation of decapped, rather than capped, intermediates for the set of transcripts preferentially targeted for degradation by Dcp1/Dcp2 and Xrn1.

We observed strong correlations ($\rho$ = 0.99) between biological replicates for WT, $dcp2\Delta$, and $xrn1\Delta$ cells, for both nAnT-iCAGE-Seq and parallel RNA-Seq analysis (*Figure 3—figure supplement 1A–F*). There is also a strong positive correlation ($\rho$ = 0.83) between changes in TPMs for total RNA from RNA-Seq ($\Delta$mRNA_T) with TPMs of capped RNA from nAnT-iCAGE-Seq ($\Delta$mRNA_C) in $dcp2\Delta$ vs. WT (*Figure 3—figure supplement 2A*), and strong overlap between genes showing significant increases in total mRNA vs. capped mRNA in $dcp2\Delta$ vs. WT cells as determined by DESeq2 analysis (*Figure 3—figure supplement 2C*)—all as expected if transcripts increased by $dcp2\Delta$ accumulate as capped isoforms in the mutant cells. In contrast, we found a marked negative correlation ($\rho$ = −0.64) between the changes in total vs. capped mRNAs conferred by $xrn1\Delta$ (*Figure 3—figure supplement 2B*), and a corresponding under-enrichment of mRNAs significantly increased in total vs. capped transcripts (*Figure 3—figure supplement 2D*), as expected if the mRNAs increased by $xrn1\Delta$ accumulate as decapped isoforms. The opposite effects of $dcp2\Delta$ and $xrn1\Delta$ in these comparisons validate this approach to evaluating relative proportions of capped transcripts for each gene in different strains.

Consistent with the 'Diminished decay' model, transcripts elevated in total RNA by $dcp2\Delta$ exhibit a lower C/T ratio in WT cells compared to all mRNAs (*Figure 3D*, cols. 1–2) and their C/T ratios are increased by $dcp2\Delta$ to a greater extent than for all mRNAs (*Figure 3E*, cols. 1–2), as expected if their low C/T ratios in WT result from enhanced decapping (*Figure 3C* (i)). The transcripts decreased in abundance by $dcp2\Delta$ exhibit the opposite features of greater than average C/T ratios in WT (*Figure 3D*, col. 3 vs. 1) and lower than average increases in C/T ratios in $dcp2\Delta$ vs. WT cells (*Figure 3E*, col. 3 vs. 1). (Results nearly identical to those in *Figure 3D, E* were obtained after excluding all ESR mRNAs from consideration.) The $xrn1\Delta$ mutation has the opposite effects from $dcp2\Delta$ on the C/T ratios of the mRNA_up_$dcp2\Delta$ and mRNA_dn_$dcp2\Delta$ groups in comparison to all mRNAs (*Figure 3—figure supplement 3A*). Extending the analysis to include all mRNAs, which were binned according to their changes in total mRNA abundance between $dcp2\Delta$ and WT cells, revealed that greater increases in total mRNA levels are generally associated with greater increases in C/T ratios conferred by $dcp2\Delta$ vs. WT (*Figure 3—figure supplement 3B*). The inverse trend was observed between changes in C/T ratios and changes in total mRNA abundance conferred by $xrn1\Delta$ (*Figure 3—figure supplement 3C*). These findings support the model that loss of decapping and attendant increased mRNA stability is a major driver of increased mRNA abundance in $dcp2\Delta$ vs. WT cells. Moreover, both the Dhh1-independent and -dependent subsets of mRNAs up-regulated by $dcp2\Delta$ (described in *Figure 1E, F*) resemble the entire group of mRNA_up_$dcp2\Delta$ transcripts in showing lower than average C/T ratios in WT (*Figure 3D*, cols. 4–5 vs. 2) and greater than average increases in C/T ratios in response to $dcp2\Delta$ (*Figure 3E*, cols. 4–5 vs. 2). Thus, the Dhh1-independent mRNAs, which are generally repressed by NMD factors, appear to be targeted for decapping to the same extent as the Dhh1-dependent transcripts whose repression depends on Dhh1/Pat1/Edc3 or Scd6.

We came to the same overall conclusions after analyzing the data differently by comparing the ratio of CAGE TPMs between mutant and WT (TPM_CAGE$_{dcp2\Delta/WT}$) to the ratio of total RNA TPMs between the two strains (TPM_RNA$_{dcp2\Delta/WT}$) for individual genes. If changes in mRNA abundance result from altered transcription, we expect similar changes in both ratios for the transcripts up-regulated by $dcp2\Delta$. If instead impaired decapping is responsible, then the TPM_CAGE$_{dcp2\Delta/WT}$ ratio should exceed the TPM_RNA$_{dcp2\Delta/WT}$ ratio for the up-regulated transcripts. The latter result was observed for the majority of the 811 mRNA_up_$dcp2\Delta$ transcripts with available CAGE data (*Figure 3—figure supplement 3D*).

As an independent assessment of whether the increased mRNA levels in $dcp2\Delta$ cells results from impaired decapping, we determined the codon protection index (CPI) of the mRNA_up_$dcp2\Delta$ transcripts, a measure of co-translational decay. Decapped mRNA degradation intermediates exhibit three-nucleotide periodicity generated by Xrn1 exonucleolytic cleavage behind the last translating ribosomes, and the CPI metric captures the prevalence of such intermediates for each mRNA (*Pelechano et al., 2015*). Importantly, the mRNA_up_$dcp2\Delta$ group exhibits higher than average CPIs, indicating a greater than average involvement of decapping and co-translational degradation by Xrn1 in their decay, whereas the mRNA_dn_$dcp2\Delta$ transcripts exhibit lower than average CPI values (*Figure 3—figure supplement 4A*), consistent with the involvement of an alternative degradation pathway controlling their abundance.

Finally, to investigate the possible contribution of increased transcription to increased mRNA abundance conferred by $dcp2\Delta$, we performed ChIP-Seq on RNA Polymerase II subunit Rpb1 to measure

Pol II occupancies across the CDSs of each gene. Adding a spike-in of *S. pombe* chromatin to each *S. cerevisiae* chromatin sample prior to immunoprecipitation of *S.cerevisiae* and *S. pombe* Rpb1 with the same antibodies allowed us to compare absolute Pol II occupancies between WT and *dcp2Δ* cells. We observed excellent correlations among the Rpb1 occupancies determined in three biological replicates (*r* = 0.99, ***Figure 3—source data 3***). To measure absolute changes in mRNA abundance, we repeated the RNA-Seq experiments by adding a fixed amount of External RNA Controls Consortium (ERCC) transcripts to equal amounts of total RNA from WT and *dcp2Δ* cells prior to preparation of cDNA libraries. Importantly, the spike-in normalized RNA-Seq data revealed a median increase of 24% in bulk mRNA in *dcp2Δ* vs. WT cells (***Figure 3F***, col. 1), indicating elevated total mRNA levels in cells lacking Dcp2. We identified groups of transcripts with >1.4-fold absolute changes in *dcp2Δ* vs. WT cells (FDR <0.01), dubbed mRNA_up_*dcp2Δ*_A and mRNA_dn_*dcp2Δ*_A, respectively (***Figure 3F***, cols. 3 and 5). (The ERCC spike-in shrank the group of down-regulated mRNAs while expanding the group of up-regulated mRNAs because many mRNAs that show reduced relative abundance compared to the average gene exhibit increased absolute abundance after spike-in normalization.) Interestingly, *dcp2Δ* confers a 1.3-fold reduced median Rpb1 occupancy for all expressed genes, indicating a global reduction in transcription rate (***Figure 3F***, col. 2). This is consistent with previous results indicating that decreased mRNA turnover in mutants lacking mRNA degradation enzymes is buffered by decreased rates of transcription (***Sun et al., 2013***). Importantly, the increased mRNA_up_*dcp2Δ*_A transcripts show reduced absolute Rpb1 occupancies in *dcp2Δ* vs. WT cells (***Figure 3F***, cols. 3–4), which applies broadly to individual transcripts within the mRNA_up_*dcp2Δ*_A group (***Figure 3G***) and is exemplified by *ATG8*, *FLO5*, and *CAT8* (***Figure 3—figure supplement 4B***). Supporting this, the mRNA_up_*dcp2Δ* group is 9.4-fold depleted for a group of 930 transcripts showing similar changes in Rpb1 occupancies and mRNA levels conferred by *dcp2Δ* (***Figure 3—figure supplement 4C***). These results confirm our conclusion that most mRNAs are increased in abundance by *dcp2Δ* owing primarily to decreased mRNA turnover vs. increased transcription.

A departure from this last generalization occurred for a subset of the iESR genes known to be activated by transcription factors Msn2/Msn4 (***Elfving et al., 2014***), which exhibit increased median Rpb1 occupancies in *dcp2Δ* vs. WT cells, whereas the remaining iESR genes show decreased Rpb1 levels (***Figure 3—figure supplement 4D*** cols. 3-4). A representative Msn2-activated iESR gene, *TPS2*, displays elevated Rpb1 occupancy across the CDS that parallels the increased mRNA abundance in *dcp2Δ* vs. WT cells (***Figure 3—figure supplement 4E***). Thus, it appears that activation of certain iESR genes by Msn2/Msn4 partially overrides the effect of transcriptional buffering to yield a net increase in transcription in *dcp2Δ* cells. The mRNA_dn_*dcp2Δ*_A group shows a lower median Rpb1 occupancy that is similar in magnitude to its decreased median mRNA abundance (0.68- vs. 0.61-fold, ***Figure 3F***, cols. 5–6), suggesting that decreased transcription contributes to the decreased abundance of these transcripts down-regulated by *dcp2Δ*.

## Dcp2 modulates the TEs of many mRNAs

Because the possible role of Dcp2 in translational control was largely unexplored, we determined the changes in translation for all mRNAs conferred by *dcp2Δ* using our previous ribosome profiling analysis (Ribo-Seq) of the same WT and *dcp2Δ* strains subjected to RNA-Seq (***Zeidan et al., 2018***). Ribo-Seq entails deep sequencing of mRNA fragments protected from RNase cleavage by translating 80S ribosomes (RPFs), which when normalized to total mRNA levels, yields the relative ribosome density on each mRNA, a measure of relative TE. Interestingly, DESeq2 analysis revealed that Dcp2 differentially controls the TEs of hundreds of mRNAs (***Figure 4A***), with 541 mRNAs showing higher TEs and a similar number (*n* = 659) displaying reduced TEs in *dcp2Δ* cells (TE_dn_*dcp2Δ*) (***Figure 4A, B***). These results suggest that Dcp2 broadly controls gene expression directly or indirectly at the translational level in addition to regulating mRNA stability.

To evaluate whether the changes in TE are generally associated with changes in the synthesis and abundance of the encoded proteins, we measured the changes in steady-state levels of individual proteins by Tandem Mass Tag Mass spectroscopy (TMT-MS/MS) of total proteins extracted from the *dcp2Δ* and WT strains. All peptides in the mutant and WT extracts were covalently labeled with different isobaric tags, which have the same mass but yield distinguishable reporter ions in the tandem MS mode. Analyzing a mixture of differentially labeled mutant and WT samples yields the ratios of peptide abundance in the two strains. We determined changes in abundance for ~4600 different

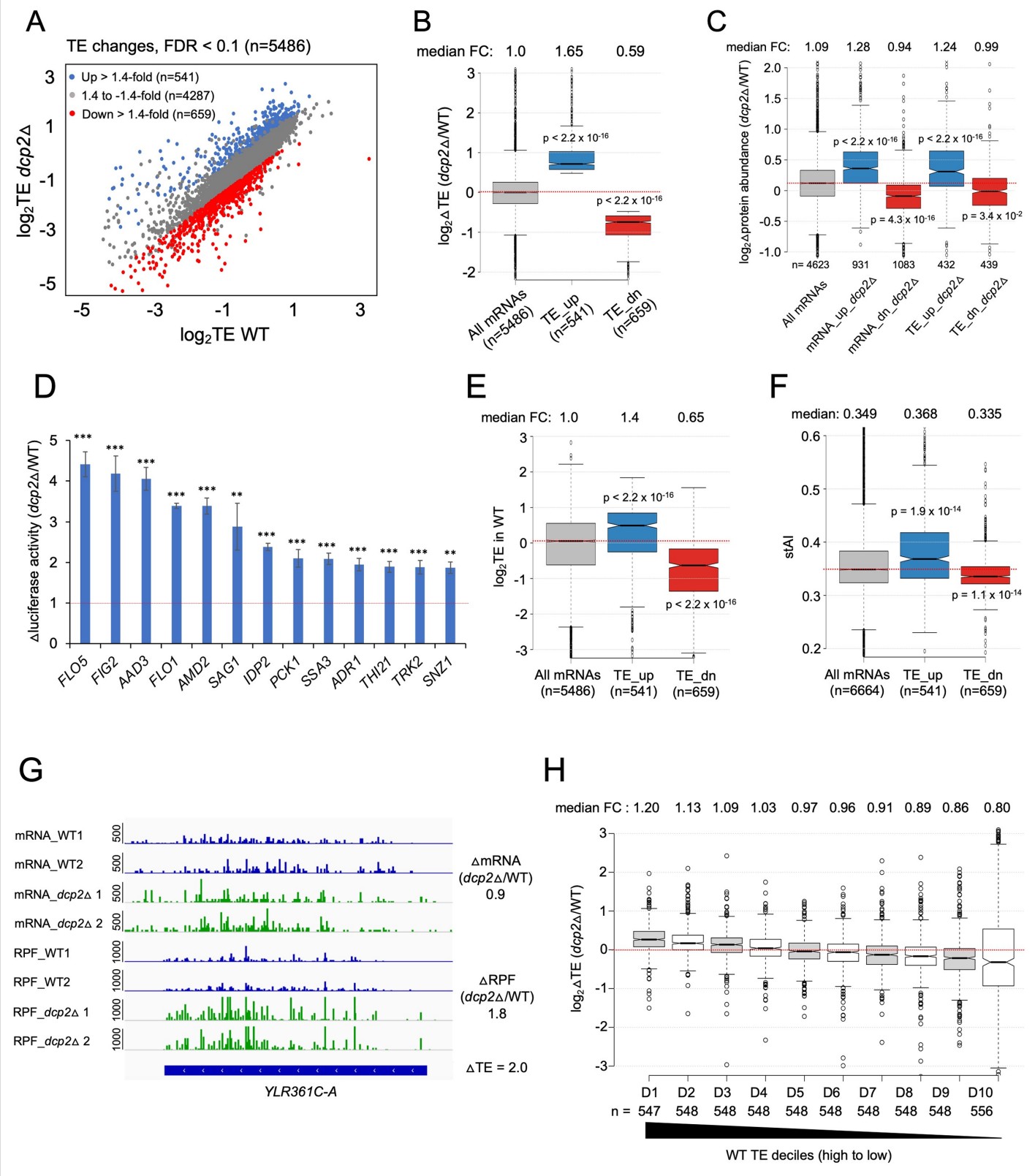

**Figure 4.** Dcp2 regulates the TEs of hundreds of transcripts. (**A**) Scatterplot as in *Figure 1A* except displaying significant log₂ fold-changes in TE in *dcp2Δ* vs. WT cells, determined by ribosome profiling. (**B**) Notched box-plot of log₂ fold-changes in TE (from ribosome profiling) in *dcp2Δ* vs. WT cells for all mRNAs (with median FC of 1.0) and for the two sets of TE_up_*dcp2Δ* or TE_dn_*dcp2Δ* transcripts showing translational repression or stimulation, respectively, by Dcp2. (**C**) Notched box-plot of log₂ fold-changes in protein abundance determined by TMT-MS/MS in *dcp2Δ* vs. WT cells for all mRNAs

*Figure 4 continued on next page*

*Figure 4 continued*

or for the four indicated groups showing Dcp2-mediated repression or stimulation of mRNA abundance or TE. (**D**) Changes in Nano-luciferase activity in *dcp2Δ* vs. WT cells expressed from the indicated 13 *nLUC* reporters. Average values (± standard error of the mean [SEM]) from at least three biological replicates are shown. Results of *t*-tests are indicated as: ***p < 0.001, **p < 0.01. Notched box-plot of $\log_2$TE in WT (**E**) and species-specific tRNA adaptation index (stAI) values (**F**) for all mRNAs or for the two groups showing repression or stimulation of translation by Dcp2. (**G**) Representative gene exhibiting increased TE in *dcp2Δ* vs. WT cells. Integrated Genomics Viewer (IGV, Broad Institute) display of mRNA and RPF reads across the *YLR361C-A* gene from two biological replicates each for WT, and *dcp2Δ* strains, shown in units of RPKM (reads per 1000 million mapped reads). Position of the CDS (blue) is at the bottom with the scale in bp; scales of RPKM for each track are on the left, and calculated ΔmRNA, ΔRPF and ΔTE values between each mutant and WT are on the right. (**F**) Notched box-plots of $\log_2$ fold-changes in TE in *dcp2Δ* vs. WT across ten deciles of transcripts binned according to TE in WT cells, progressing left to right from highest to lowest TEs.

The online version of this article includes the following source data and figure supplement(s) for figure 4:

**Source data 1.** $\log_2$ fold-changes in TE measured by Ribo-Seq and parallel RNA-Seq analyses of *dcp2Δ* vs. WT cells for all transcripts and for the TE_up_*dcp2Δ* and TE_dn_*dcp2Δ* groups of mRNAs (***Figure 4A, B***).

**Source data 2.** $\log_2$ fold-changes in protein abundance measured by TMT-MS/MS analysis of *dcp2Δ* vs. WT cells for all genes (***Figure 4C***).

**Source data 3.** Specific activity of luciferase expressed from *nLUC* reporters in WT and *dcp2Δ* cells for three biological replicates, and changes in luciferase activity calculated for *dcp2Δ* vs. WT with SEM values (***Figure 4D***).

**Figure supplement 1.** Marked correlation between RPF changes (from ribosome profiling) and protein abundance changes (from TMT-MS/MS) conferred by *dcp2Δ*.

**Figure supplement 1—source data 1.** Normalized protein abundances determined by TMT-MS/MS for biological replicates in WT and *dcp2Δ* cells (***Figure 4—figure supplement 1A–C***).

**Figure supplement 1—source data 2.** $\log_2$ fold-changes in protein abundance measured by TMT-MS/MS and RPFs measured by ribosome profiling for *dcp2Δ* vs. WT cells for all genes/transcripts (***Figure 4—figure supplement 1D***).

**Figure supplement 2.** Dcp2 translationally repressed transcripts have properties associated with well-translated mRNAs.

proteins, with highly reproducible results for three biological replicates of each strain (***Figure 4—figure supplement 1A–C***). Importantly, a positive correlation exists between the relative changes in protein abundance from TMT-MS/MS and changes in RPFs from Ribo-Seq in *dcp2Δ* vs. WT cells ($\rho = 0.6$) (***Figure 4—figure supplement 1D***). Furthermore, the groups of mRNAs defined above showing significant changes in mRNA abundance or TEs conferred by *dcp2Δ* displayed changes in protein levels in the same directions (***Figure 4C***). Considering that protein abundance is controlled by rates of degradation in addition to rates of synthesis, the substantial correspondence between Ribo-Seq and TMT/MS-MS data indicates that the changes in RPFs generally signify corresponding changes in translation rates between *dcp2Δ* and WT cells. To provide additional support for this last conclusion, we analyzed the expression of Nano-luciferase (*nLUC*) reporters constructed for particular genes by inserting *nLUC* CDSs immediately preceding the stop codon of each gene, preserving the native 5′UTR and 3′UTR sequences. We observed increased luciferase expression in cell extracts of *dcp2Δ* vs. WT transformants harboring reporter plasmids for 13 different genes that showed increased RPFs in *dcp2Δ* vs. WT cells (***Figure 4D***).

The mRNAs displaying increased TEs in *dcp2Δ* cells tend to have shorter CDS lengths, longer half-lives, and greater mRNA abundance compared to all mRNAs (***Figure 4—figure supplement 2A–C***). These features are associated with efficiently translated mRNAs and, indeed, the TE_up_*dcp2Δ* mRNAs have greater than average TEs in WT cells (***Figure 4E***) and are enriched for optimal codons (***Figure 4F***). The TE_dn_*dcp2Δ* mRNAs have properties opposite of those exhibited by the TE_up_*dcp2Δ* group, including longer CDS, shorter half-lives and lower transcript abundance compared to all mRNAs (***Figure 4—figure supplement 2A–C***), and lower than average TE in WT (***Figure 4E***) and frequency of non-optimal codons (***Figure 4F***). Extending this analysis to include all expressed mRNAs, sorted into 10 bins on the basis of their TEs in WT cells, revealed a direct correlation between TE changes conferred by *dcp2Δ* and TE in WT (***Figure 4H***), supporting the notion that Dcp2 translationally repressed mRNAs tend to be well translated, whereas Dcp2 translationally activated transcripts are generally poorly translated, in WT cells.

## Dcp2 translationally repressed transcripts generally do not accumulate as decapped low-TE species in WT cells

The mRNAs encoded by *YLR361C-A* and *YLR297A* are representative transcripts exhibiting TE increases conferred by *dcp2Δ* of 2.0- and 2.7-fold, respectively, but displaying no significant change or a considerably smaller increase in mRNA abundance in *dcp2Δ* vs. WT cells (*Figure 4G*; *Figure 4—figure supplement 2D*). This suggests that Dcp2 represses their translation without preferentially targeting these mRNAs for degradation. In fact, *dcp2Δ* generally confers the opposite effects on mRNA abundance and TE for the cohort of mRNAs it regulates translationally, as the TE_up_*dcp2Δ* group shows a decreased median mRNA level, while the TE_dn_*dcp2Δ* group of mRNAs shows an increased median mRNA level in *dcp2Δ* vs. WT cells (*Figure 5A*).

We wondered whether the mRNAs translationally repressed by Dcp2 might be preferentially decapped but not rapidly degraded by Xrn1, such that their relative mRNA abundance is not down-regulated by Dcp2. Such decapped mRNAs would have a low TE owing to the inability to bind eIF4F and to be activated for translation initiation, and their TEs would increase in *dcp2Δ* cells because they would remain capped and capable of binding eIF4F (see model in *Figure 5B*). This model predicts that the TE_up_*dcp2Δ* mRNAs should exhibit lower than average relative proportions of capped mRNAs (C/T ratios) in WT cells, and a greater than average increase in C/T ratios conferred by *dcp2Δ* owing to loss of decapping (*Figure 5B*). Instead, the TE_up_*dcp2Δ* mRNAs have somewhat higher than average C/T ratios in WT cells (*Figure 5C*), and show an increase in C/T ratios in *dcp2Δ* vs. WT cells indistinguishable from that seen for all mRNAs (*Figure 5D*), inconsistent with the decapping model for translational repression by Dcp2.

## Evidence that competition for limiting ribosomes reprograms translation in *dcp2Δ* cells

We considered an alternative possibility that *dcp2Δ* confers TE changes as an indirect consequence of elevated mRNA levels resulting from loss of the major pathway for mRNA degradation. This idea was prompted by the above finding that mRNAs showing TE increases in *dcp2Δ* cells tend to be efficiently translated in WT cells, whereas mRNAs poorly translated in WT tend to show TE reductions in the mutant (*Figure 4E, H*). Previously, we observed this pattern of translational reprogramming in yeast cells impaired in different ways for assembly of 43S PICs, including (1) increased phosphorylation of eIF2α in WT cells induced by isoleucine/valine starvation using the drug sulfometuron methyl (SM), which decreases formation of the eIF2-GTP-Met-tRNA$_i$ ternary complex required to assemble 43S PIC; and (2) deletion of genes *TMA64* and *TMA20* encoding factors that recycle 40S subunits from termination complexes at stop codons to provide free 40S subunits for PIC assembly. The translational reprogramming was explained as resulting from increased competition for limiting PICs that allows 'strong' well-translated mRNAs, highly efficient in recruiting PICs, to outcompete 'weak' poorly translated mRNAs that recruit PICs less efficiently (*Gaikwad et al., 2021*). Supporting that a similar competition exists among mRNAs translationally altered by *dcp2Δ*, we found that the TE_up_*dcp2Δ* group of mRNAs shows increased median TE in response to both SM treatment of WT cells and deletion of *TMA64/TMA20* (*tmaΔΔ*), whereas the TE_dn_*dcp2Δ* mRNAs exhibit the opposite changes in median TE in response to these conditions (*Figure 5—figure supplement 1A*). Hierarchical clustering analysis reveals that the majority of mRNAs showing significant TE changes in *dcp2Δ* vs. WT cells exhibit changes in the same direction in response to the *tmaΔΔ* mutations or SM treatment (*Figure 5—figure supplement 1B*). There are numerous exceptions to this trend, however, suggesting that perturbations specific to each condition can differentially affect the translation of particular mRNAs and alter their responses to increased competition for limiting PICs.

We considered that increased competition among mRNAs for limiting PICs might exist in *dcp2Δ* cells for two main reasons: (1) impairing decapping-mediated mRNA decay will stabilize most transcripts and elevate bulk cellular mRNA abundance; and (2) reduced ribosome content arising from the repression of rESR mRNAs, which encode ribosomal proteins and ribosome biogenesis factors (*Gasch et al., 2000*). Indeed, a recent study demonstrated reduced ribosome content in mutant cells undergoing the ESR (*Terhorst et al., 2020*). We hypothesized that the combination of increased mRNA abundance and reduced ribosomal content will increase competition among mRNAs for limiting PICs and confer the observed translational reprogramming that favors strong over weak mRNAs in *dcp2Δ* cells (*Figure 5E*).

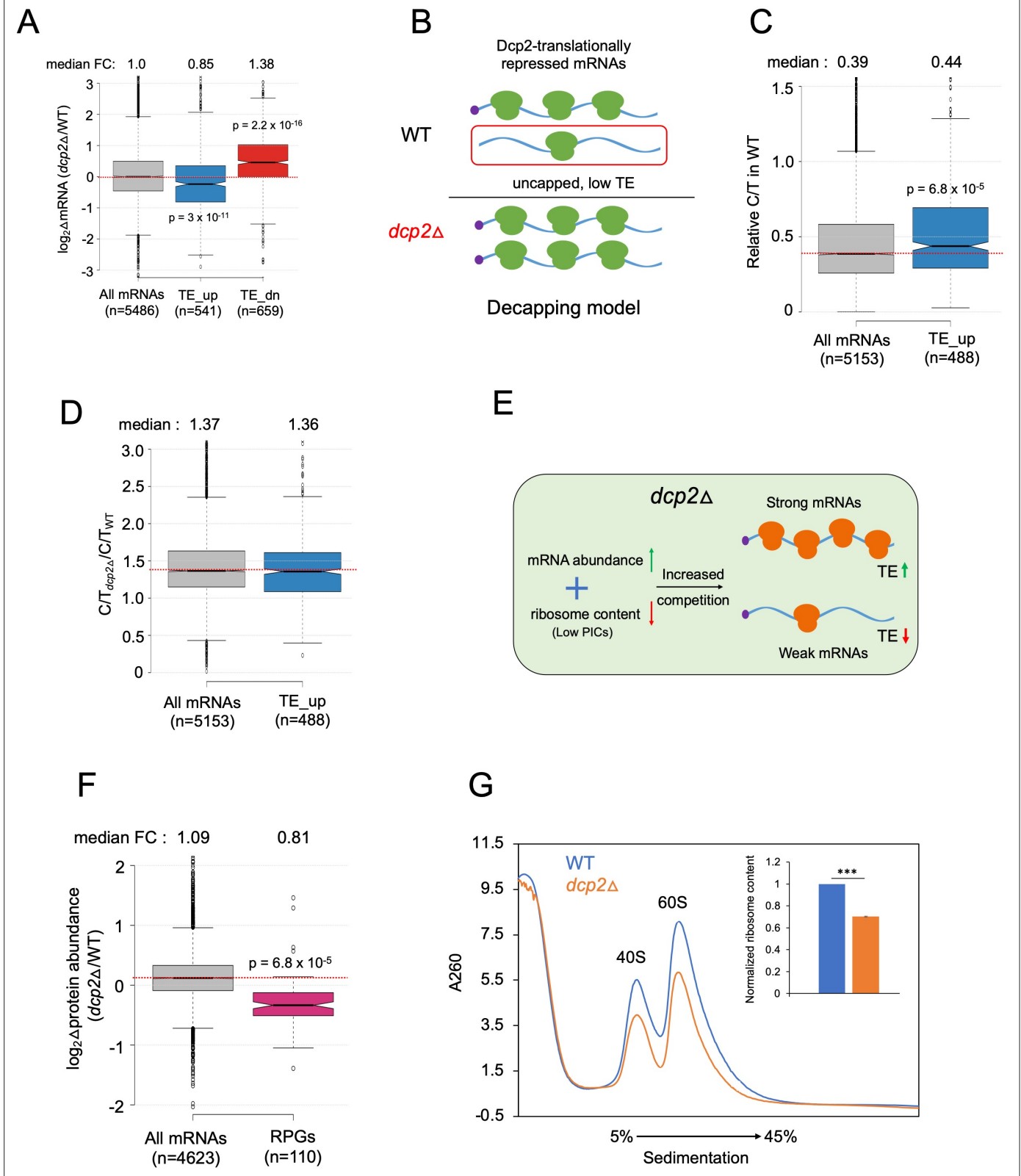

**Figure 5.** Evidence that the majority of TE changes conferred by *dcp2Δ* result from increased competition for limiting PICs owing to diminished ribosome production and elevated mRNA levels. (**A**) Notched box-plot showing log₂ fold-changes in mRNA abundance in *dcp2Δ* relative to WT cells for all mRNAs, or the mRNAs translationally repressed (blue) or stimulated (red) by Dcp2 in WT cells. (**B**) Hypothetical schematic model to explain TE increases conferred by *dcp2Δ* resulting from the persistence of translationally inert, decapped intermediates in WT cells. In WT (upper), the

*Figure 5 continued*

TE_up_*dcp2Δ* group of mRNAs is preferentially targeted by Dcp2 for decapping and these uncapped species cannot bind eIF4F and thus exhibit low TEs. In *dcp2Δ* cells (lower), decapping is eliminated and the low-TE, decapped fraction no longer exists, which increases the overall TE of the transcript pool. Notched box-plots of ratios of capped mRNA TPMs to total mRNA TPMs (C/T) (**C**) and C/T ratios in *dcp2Δ* vs. WT cells (**D**) for all mRNAs and the group translationally repressed by Dcp2. (**E**) Schematic of the preinitiation complex (PIC) competition model proposed to explain the broad reprogramming of TEs conferred by *dcp2Δ*. A combination of diminished ribosome production resulting from down-regulation of rESR transcripts and elevated bulk capped mRNAs resulting from loss of decapping-mediated mRNA turnover evokes increased competition among all mRNAs for limiting PICs, producing relatively greater translation of efficiently translated mRNAs in WT cells (strong mRNAs) at the expense of poorly translated mRNAs in WT (weak mRNAs). (**F**) Notched box-plot showing $\log_2$ fold-changes in protein abundance determined by TMT-MS/MS in *dcp2Δ* vs. WT cells for all mRNAs or those from 110 genes encoding ribosomal proteins (RPGs). (**G**) Quantification of total 40S and 60S ribosomal subunits in *dcp2Δ* vs. WT cells. Representative $A_{260}$ profiles of equal proportions of cell extracts obtained from WT (blue) and *dcp2Δ* (orange) cultures are shown. The inset summarizes the combined areas under the 40S and 60S peaks normalized to the $OD_{600}$ of the cell cultures calculated from three biological replicates of *dcp2Δ* and WT cells, setting the mean WT value to unity. An unpaired Student's *t*-test indicates a highly significant difference in the means calculated from the biological replicates of the two different strains (***$p < 0.001$).

The online version of this article includes the following source data and figure supplement(s) for figure 5:

**Source data 1.** Quantification of ribosome content relative to $OD_{600}$ in *dcp2Δ* vs. WT cells (***Figure 5G***).

**Figure supplement 1.** Supporting evidence that *dcp2Δ* evokes translational reprogramming by increasing competition for limiting preinitiation complexes (PICs).

**Figure supplement 1—source data 1.** $\log_2$ fold-changes in TE for all transcripts conferred by SM treatment of WT cells, or the *tma64Δ/tma20Δ* double mutation (***Figure 5—figure supplement 1A, B***).

**Figure supplement 1—source data 2.** $\log_2$ fold-changes in TE in *dcp2Δ* vs. *dcp2-EE* cells for all mRNAs measured by ribosome profiling (***Figure 5—figure supplement 1D***).

As mentioned above, our spike-in normalized RNA-Seq data revealed a median increase of 24% in bulk mRNA in *dcp2Δ* vs. WT cells (***Figure 3F***), indicating elevated mRNA abundance in cells lacking Dcp2. Considering that all mRNAs should be capped in *dcp2Δ* cells, whereas a fraction of mRNAs are uncapped in WT, the increase in capped mRNA abundance conferred by *dcp2Δ* should be even greater than 24%. Evidence that *dcp2Δ* reduces ribosome content came from our findings that the group of 139 ribosomal proteins exhibits ~twofold lower RPFs (***Figure 5—figure supplement 1C***) and a ~25% reduction in median abundance determined by TMT/MS (***Figure 5F***) in *dcp2Δ* vs. WT cells. We then measured the abundance of assembled ribosomal subunits by resolving whole cell extracts of mutant and WT cells by sedimentation through sucrose density gradients, using an extraction buffer lacking $Mg^{+2}$ to dissociate 80S ribosomes into free 40S and 60S subunits. Normalizing the $A_{260}$ absorbance of ribosomal subunits to the number of $OD_{600}$ units (cellular volume) of extracted cells, revealed ~30% lower levels of 40S and 60S subunits in *dcp2Δ* vs. WT cells (***Figure 5G***). The significant reduction in ribosome levels coupled with the increased capped mRNA content in *dcp2Δ* cells should increase competition among mRNAs for limiting PICs to favor well-translated mRNAs at the expense of poorly translated ones, as we observed (***Figure 4H***).

## Dhh1 is required for translational repression of a subset of transcripts by Dcp2

To examine whether Dhh1 contributes to this indirect mechanism of translational reprogramming, we asked whether mRNAs translationally dysregulated by *dcp2Δ* are dependent on Dhh1 for their TE changes. Cluster analysis revealed that most transcripts showing increased TEs in the *dcp2Δ* single mutant show similar TE increases in the *dhh1Δdcp2Δ* double mutant but exhibit little increase, or even decreases, in TE in the *dhh1Δ* single mutant vs. WT (***Figure 6A***, orange bars), indicating that they are controlled by Dcp2 independently of Dhh1. Indeed, this subset of Dcp2-translationally repressed mRNAs shows nearly identical increases in median TE in the *dhh1Δdcp2Δ* double and *dcp2Δ* single mutant of ~1.6-fold, but only a slight increase of ~1.1-fold in the *dhh1Δ* single mutant (***Figure 6B***, cols. 4–6). Interestingly, the remaining one-fourth of TE_up_*dcp2Δ* transcripts exhibits similar TE increases in all three deletion mutants (***Figure 6A***, green bars), with nearly identical ~2.3-fold increases in median TE relative to WT (***Figure 6B***, cols. 1–3), indicating dependence on Dhh1 for translational repression by Dcp2. The Dhh1-dependent subset shows considerably greater translational repression by Dcp2 compared to the Dhh1-independent group (***Figure 6B***, compare cols. 1 and 4).

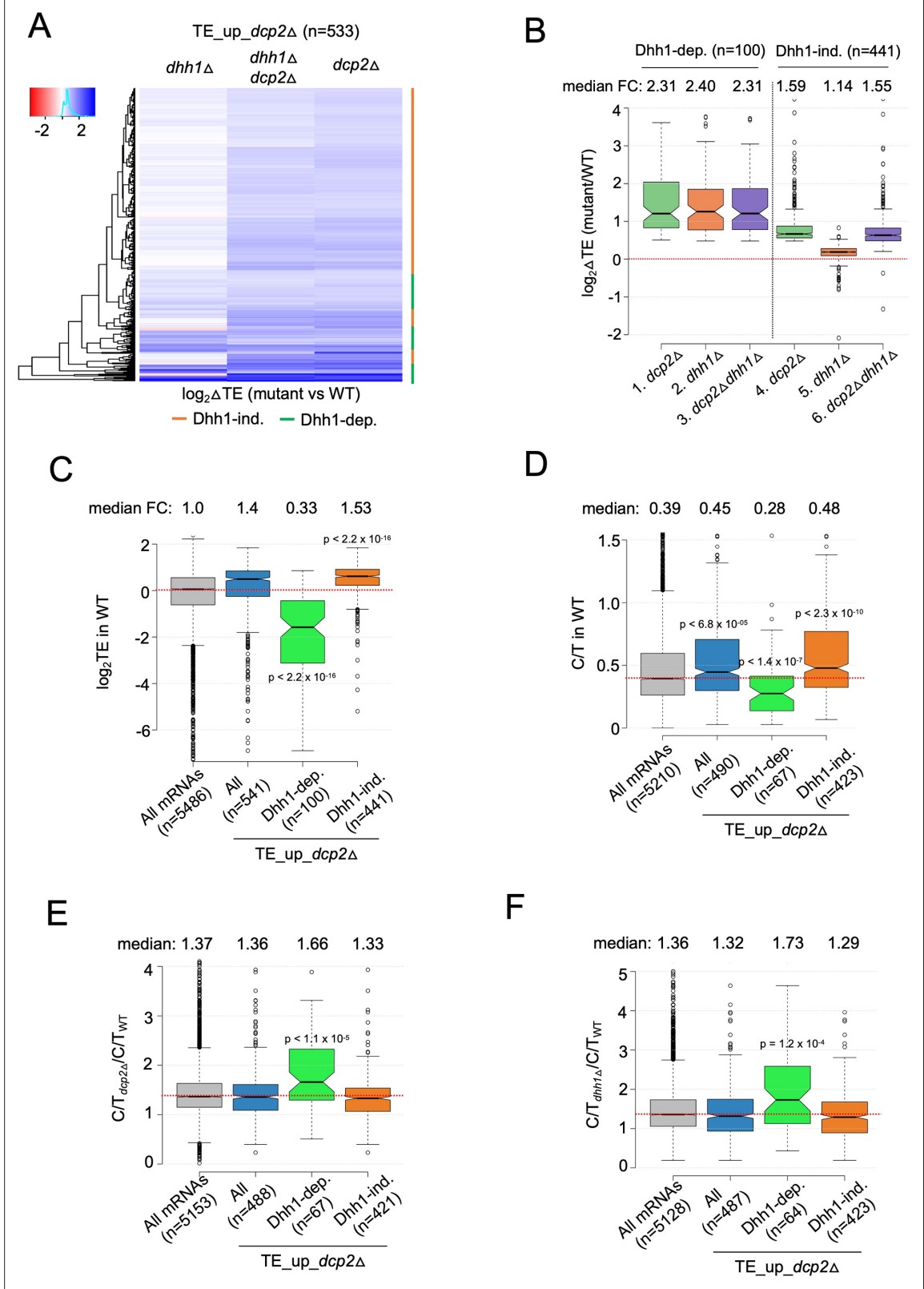

**Figure 6.** mRNAs exhibiting Dhh1-dependent translational repression by Dcp2 are poorly translated mRNAs preferentially targeted for decapping by Dcp2 and Dhh1. (**A**) Hierarchical clustering analysis of log₂ fold-changes in TE conferred by the mutations listed across the top vs. WT for the TE_up_*dcp2*Δ mRNAs (excluding a few outliers [*n* = 7] with log₂ΔTE values >+4 or <−4). Dhh1 dependence or Dhh1 independence for the TE changes is indicated on the right with green or red bars, respectively. (**B**) Notched box-lot of log₂ fold-changes in TE conferred by the mutations listed across the

*Figure 6 continued*

bottom vs. WT for the Dhh1-dependent TE_up_*dcp2*Δ (col. 1–3) or Dhh1-independent TE_up_*dcp2*Δ (col. 4–6) groups of mRNAs. Notched box-plot of log$_2$ TE in WT (**C**), C/T ratios in WT (**D**), C/T ratios in *dcp2*Δ vs. WT (**E**), and C/T ratios in *dhh1*Δ vs. WT (**F**) for all mRNAs, all TE_up_*dcp2*Δ mRNAs, or the Dhh1-dependent or -independent subsets of the TE_up_*dcp2*Δ mRNAs.

The online version of this article includes the following source data and figure supplement(s) for figure 6:

**Source data 1.** log$_2$ fold-changes in TE observed in *dcp2*Δ, *dhh1*Δ, and *dcp2*Δ*dhh1*Δ cells vs. WT cells for all transcripts, Dhh1-dependent TE_up_*dcp2*Δ, and Dhh1-independent TE_up_*dcp2*Δ transcript groups (**Figure 6A, B**).

**Figure supplement 1.** Additional evidence that mRNAs exhibiting Dhh1-dependent translational repression by Dcp2 are preferentially targeted for decapping by Dcp2 and Dhh1.

The Dhh1-independent subset exhibits higher than average median TEs in WT cells (**Figure 6C**, col. 4 vs. 1), suggesting that their TE increases conform to the PIC competition model (**Figure 5E**). The Dhh1-dependent subset, by contrast, have a much lower than average median TE in WT (**Figure 6C**, col. 3 vs. 1), suggesting that their translation is repressed by a different mechanism involving Dhh1. We considered that the latter might involve Dhh1-stimulated decapping and persistence of the decapped mRNAs in the low-TE state envisioned in the decapping model for translational repression (**Figure 5B**). Supporting this, the Dhh1-dependent subset of TE_up_*dcp2*Δ transcripts exhibit a lower relative proportion of capped transcripts (C/T ratio) in WT cells compared to all mRNAs and to the Dhh1-independent group of Dcp2 translationally repressed mRNAs (**Figure 6D**, col. 3 vs. 1 and 4). They also show greater than average increases in C/T ratios in both *dcp2*Δ and *dhh1*Δ vs. WT cells (**Figure 6E–F**, col. 3 vs. 1 and 4) as expected if Dhh1-stimulated decapping by Dcp2 produces the decapped isoforms in WT cells. The levels of total mRNA for this group are also increased by *dcp2*Δ and *dhh1*Δ (**Figure 6—figure supplement 1B, C**, col. 3 vs. 1–2), as expected if they are preferentially targeted for mRNA degradation by decapping/decay in addition to being translationally repressed by Dcp2/Dhh1. The Dhh1-independent group of translationally repressed transcripts, by contrast, exhibit average or lower than average repression of their abundance by Dcp2 and Dhh1 (**Figure 6—figure supplement 1B, C**, cols. 4 vs. 1). The 100 Dhh1-dependent mRNAs are not enriched for non-optimal codons (**Figure 6—figure supplement 1A**, col. 3 vs. 1), suggesting that targeting of stalled elongating ribosomes by Dhh1 (**Radhakrishnan et al., 2016**) does not underlie their translational repression by Dcp2.

## Dcp2 represses the abundance or translation of mRNAs encoding proteins involved in catabolism of alternative carbon sources or respiration

The mRNA_up_*dcp2*Δ transcripts are functionally enriched for stress response genes, reflecting mobilization of the iESR in this slow-growing mutant (**O'Duibhir et al., 2014**). They are also enriched for genes involved in metabolism of energy reserves glycogen and trehalose (**Figure 7—source data 1A**), the tricarboxylic acid (TCA) cycle (involved in respiration), or meiotic recombination (**Figure 7—source data 1A**), of which only two genes belong to the iESR (**Figure 7—figure supplement 2A-B**). The subset showing Dhh1-dependent repression of mRNA abundance by Dcp2 show enrichment for the same functional categories except for meiotic recombination, a category enriched among the Dhh1-independent subset of mRNA_up_*dcp2*Δ transcripts instead, along with genes involved in DNA repair and cell–cell adhesion (**Figure 7—source data 1C, D**). This suggests a functional bifurcation among Dcp2-repressed mRNAs based on involvement of Dhh1 in the degradation mechanism. Interestingly, the transcripts showing increased TEs in *dcp2*Δ cells (TE_up_*dcp2*Δ) are also enriched for genes involved in respiration, and for ribosomal protein genes (RPGs) (**Figure 7—source data 1E**). As most of these mRNAs are well translated in WT cells, their TE increases can be attributed to the competition mechanism of translational reprogramming in *dcp2*Δ cells (**Figure 5E**).

To integrate the outcome of altered mRNA abundance and TEs and obtain a measure of altered protein synthesis rates, we identified the groups of mRNAs showing significantly changed ribosome occupancies in *dcp2*Δ vs. WT cells and subjected them to GO analysis. The mRNAs exhibiting increased RPFs (Ribo_up_*dcp2*Δ) are enriched for genes involved in the same categories mentioned above, including stress response factors or proteins involved in metabolism of energy reserves or respiration, but also for metabolism of vitamins or cofactors or for glutamate biosynthesis (**Figure 7—source data**

1G). Derepression by *dcp2Δ* of the stress response gene *SSA3* and two genes involved in vitamin biosynthesis, *THI21* (thiamine) and *SNZ1* (pyridoxine), was recapitulated with the corresponding *nLUC* reporters (*Figure 4D*). Examining 37 genes involved in metabolism of energy reserves reveals derepression by *dcp2Δ* at the level of mRNA abundance (*Figure 7A*), epitomized by the trehalose biosynthetic gene *TPS2* (*Figure 7—figure supplement 1A*). In contrast, 52 genes whose products function directly in respiration as components of the electron transport chain (ETC), TCA cycle or mitochondrial ATPase, are increased primarily at the level of TE (*Figure 7B*), as exemplified by the TCA cycle gene *LSC1*(*Figure 7—figure supplement 1B*). Supporting the latter, western blot analysis revealed increased steady-state levels in *dcp2Δ* cells of five mitochondrial proteins involved in respiration, Aco1, Atp20, Cox14, Idh1, and Pet10, relative to the glycolytic enzyme GAPDH (*Figure 7C*).

To obtain independent evidence that *dcp2Δ* increases respiratory function, we measured the mitochondrial membrane potential ($\Delta\Psi_m$), produced by the ETC, using the probe tetramethylrhodamine (TMRM)—a cationic fluorescent dye that accumulates in mitochondria as a function of $\Delta\Psi_m$. Quantifying dye fluorescence by flow cytometry revealed a marked increase in TMRM fluorescence in the *dcp2Δ* mutant at levels substantially greater than the background signals observed when $\Delta\Psi_m$ was dissipated by addition of the uncoupler FCCP (carbonylcyanide *p*-trifluoromethoxyphenylhydrazone). The increased TMRM fluorescence was complemented by WT *DCP2* (*Figure 7D*), providing evidence that *dcp2Δ* increases mitochondrial $\Delta\Psi_m$.

During growth on glucose-replete medium, such as YPD, respiration is suppressed and energy is produced by fermentation. Proteins involved in catabolism of alternative carbon sources are also repressed on YPD medium. Interestingly, *dcp2Δ* confers increased median translation (RPFs) of a group of 102 mRNAs shown to be glucose repressed in WT cells and/or activated by the transcriptional activators Adr1 or Cat8 that function in catabolism of non-glucose carbon sources (*Young et al., 2003*; *Tachibana et al., 2005*), which is achieved primarily via increased transcript levels in *dcp2Δ* cells (*Figure 7E*). In fact, *ADR1* and *CAT8* (*Figure 7—figure supplement 1C, D*) are up-regulated by *dcp2Δ*, which might contribute to induction of their target gene transcripts in high-glucose medium observed here. The derepression by *dcp2Δ* of *ADR1*, as well as of *IDP2* and *PCK1*, encoding glucose-repressed enzymes of the glyoxylate cycle and gluconeogenesis, respectively, was recapitulated with the corresponding *nLUC* reporters (*Figure 4D*).

## Dcp2 represses the expression of genes involved in catabolism of alternative nitrogen sources, autophagy, and invasive growth on rich medium

In addition to derepressing genes involved in catabolism of non-glucose carbon sources, we observed increased mRNA levels and translation in *dcp2Δ* cells for a group of 36 nitrogen-catabolite repressed (NCR) genes, which are transcriptionally down-regulated by the presence of the preferred nitrogen sources present in YPD medium (*Godard et al., 2007*; *Figure 7F*). Related to this finding, a group of 24 *ATG* genes directly involved in autophagy show elevated median mRNA levels and translation in *dcp2Δ* cells (as exemplified by *ATG8*; *Figure 7—figure supplement 1F-G*). This fits with previous observations indicating a role for mRNA decapping and degradation in suppressing autophagy in non-starvation conditions (*Hu et al., 2015*), where salvaging amino acids from extraneous proteins is not adaptive.

Interestingly, *dcp2Δ* cells exhibit a flocculation phenotype, wherein cells stick together and settle to the bottom of a liquid culture. Flocculation typically results from increased cell adhesion due to the up-regulation of *FLO* genes, whose products promote cell adhesion. Indeed, *dcp2Δ* confers increased mRNA levels and translation for a group of 16 genes encoding cell wall proteins, which include multiple *FLO* gene products and other agglutinins (exemplified by *FLO5*; *Figure 7—figure supplement 1E, H*). Four of these genes displayed increased expression of the corresponding *nLUC* reporters in *dcp2Δ* cells (*FLO5, FLO1, FIG2,* and *SAG1, Figure 4D*). Cell adhesion is critical for filamentous growth wherein cells switch from separated round cells to adhesion-linked 'chains' of elongated cells, which allows them to penetrate substrates (invasive growth). The *dcp2Δ* mutant showed elevated invasive growth compared to WT in a plate-washing assay, which was complemented by WT *DCP2* on a plasmid (*Figure 7G*). Quantitation of invasive growth across separate trials showed that *dcp2Δ* conferred a sixfold increase compared to WT, which was largely diminished by plasmid-borne *DCP2* (*Figure 7I*). Microscopic examination revealed that the *dcp2Δ* mutant formed clumps of cells

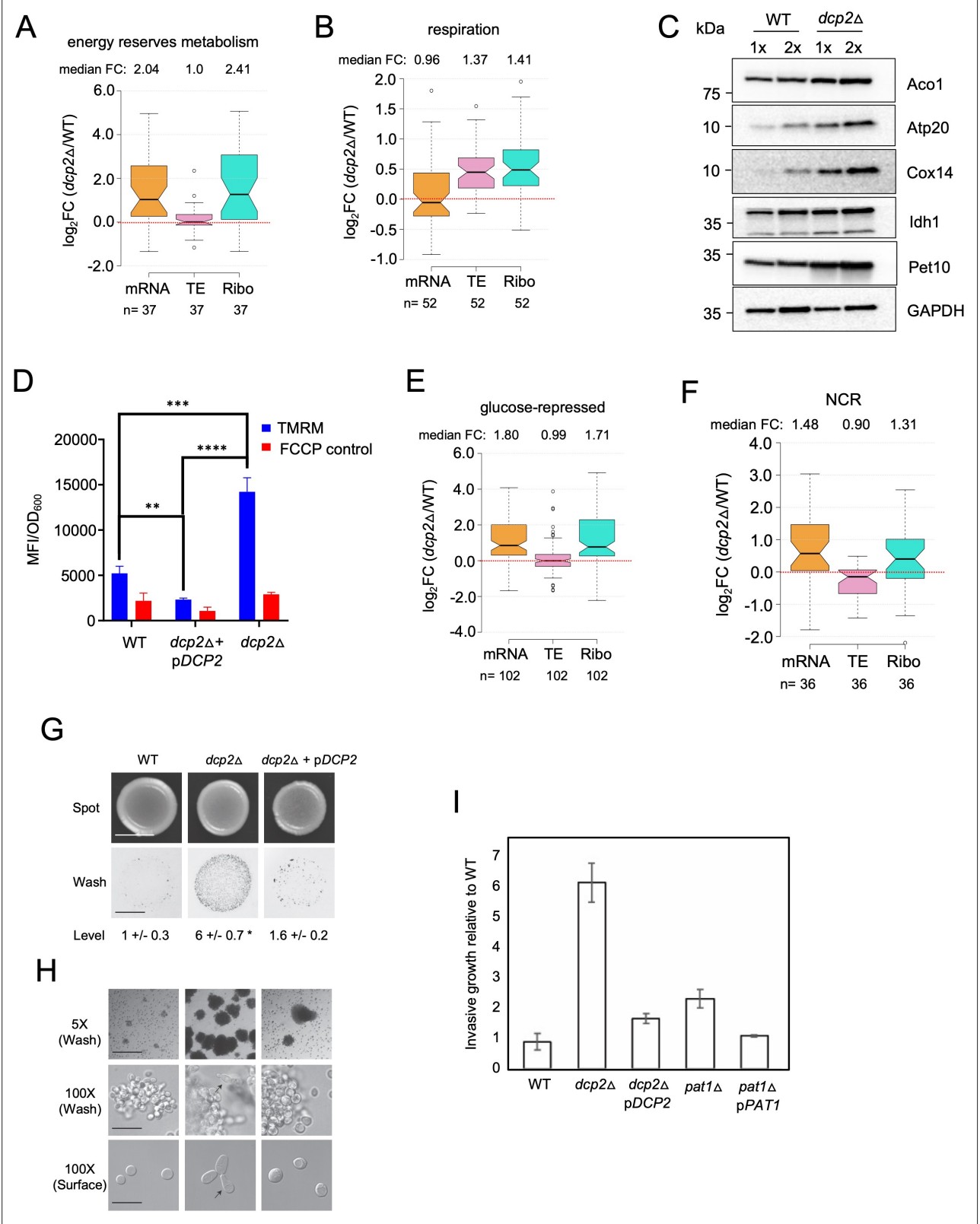

**Figure 7.** Dcp2 represses genes involved in respiration, catabolism of non-preferred carbon or nitrogen sources, or invasive growth, on rich medium (**A, B,** and **E–H**). Notched box-plot showing log₂ fold-changes in mRNA abundance, TE, or RPFs (Ribo) in *dcp2Δ* vs. WT cells for 37 genes involved in metabolism of energy reserves (identified by Kyto Encyclopedia of Genes and Genomes (KEGG) pathway analysis) (**A**), 52 genes encoding mitochondrial proteins with direct roles in OXPHOS (**B**), 102 genes subject to glucose repression or induced by Adr1 or Cat8 in low glucose (*Young et al., 2003*;

*Figure 7 continued on next page*

*Figure 7 continued*

*Tachibana et al., 2005*) (**E**), and 36 genes subject to nitrogen-catabolite repression (*Godard et al., 2007*) (**F**). (**C**) Western blot analyses of five mitochondrial proteins indicated on the right involved in respiration (and GAPDH examined as a loading control) in whole-cell extracts (WCEs) from WT or *dcp2Δ* cells, with adjacent lanes differing twofold in amount of extract. The postions of molecular weight markers are indicated to the left with masses in kilodaltons (kDa). Immune complexes were visualized using enhanced chemiluminescence. The results shown here are representative of three biological replicates that gave highly similar results, presented in the *Figure 7—source data 3*. (**D**) *dcp2Δ* confers increased mitochondrial $\Delta \Psi_m$. Cells were cultured to mid-log phase. Tetramethylrhodamine (TMRM) (500 nM) was added and incubated for 30 min before samples were collected and washed once with deionized water. $\Delta \Psi_m$ was determined by measuring TMRM fluorescence intensity using flow cytometry. Data are presented in arbitrary fluorescence intensity units per $OD_{600}$. Two-way analysis of variance (ANOVA) was used for statistical analysis and data are given as mean values ± standard deviation (SD; $n = 3$; **$p < 0.01$, ***$p < 0.001$, ****$p < 0.0001$). (**G, H**) *dcp2Δ* confers increased invasive growth. (**E**) The top and bottom panels show cells spotted on YPD agar medium and grown to confluence before or after washing under water, respectively, revealing increased invasive growth in the agar for the *dcp2Δ* strain compared to WT or the *dcp2Δ* strain complemented with WT *DCP2* on a plasmid (p*DCP2*). The levels of invasive growth were quantified and indicated below the images. (**F**) The three panels show colony or cell morphology at 5× (after wash; Bar, 100 microns), 100× (after wash; Bar, 30 microns), and 100× (surface; Bar, 20 microns) magnification, respectively, for the strains analyzed in (**E**). (**I**) Fold-change in invasive growth in *dcp2Δ*, *pat1Δ* and respective complemented strains with WT copy of gene relative to WT cells. Error represents the SD. Significance was determined by Student's *t*-test, p-value <0.05, n = 3.

The online version of this article includes the following source data and figure supplement(s) for figure 7:

**Source data 1.** Gene ontology analysis conducted on the indicated sets of mRNAs showing increased or decreased mRNA abundance (A - D), TE (E - F), or RPF abundance (G - H) in dcp2Δ vs. WT cells.

**Source data 2.** List of genes in each functional group analyzed.

**Source data 3.** Source data of western blot analyses of the expression of mitochondrial proteins, with GAPDH analyzed as loading control, for three biological replicates of *dcp2Δ* and WT strains (*Figure 7C*).

**Source data 4.** Mitochondria membrane potential in WT vs. *dcp2Δ* cells measured for three biological replicates using flow cytometry of cells stained with tetramethylrhodamine (*Figure 7D*).

**Figure supplement 1.** Supporting evidence that Dcp2 represses mRNA abundance, TE, or both properties of genes involved in different pathways not required for growth on rich medium.

**Figure supplement 2.** Overlap between iESR genes and genes in various pathways exhibiting up-regulation of mRNA abundance or TE in *dcp2Δ* vs. WT cells on YPD medium.

**Figure supplement 3.** Evidence that *dcp2Δ* reduces mRNA abundance or TE of genes involved in protein glycosylation, sulfur assimilation, or the unfolded protein response on rich medium (**A, C, D**).

---

that were larger and more abundant than WT; and that *dcp2Δ* cells exhibit an elongated morphology, which generally results from enhanced apical growth due to a delay in the cell cycle during filamentous growth (*Kron et al., 1994*; *Loeb et al., 1999*). This last phenotype was observed both in cells scraped from colonies before washing (*Figure 7H* and 100×, Surface) and in cells excised from the invasive scar after washing (*Figure 7H* and 100×, Wash). These cellular phenotypes, all of which were diminished by plasmid-borne *DCP2*, suggest that Dcp2 controls other aspects of filamentous/invasive growth besides cell adhesion. Importantly, we observed similar, less pronounced invasive growth in the mutant lacking Pat1 (*Figure 7I* and *Figure 7—figure supplement 1I-J*). Thus, mRNA decapping by Dcp1/Dcp2 and its activation by Pat1 contributes to the control of filamentous growth by the regulation of cell adhesion genes and other associated mechanisms. Only small fractions of the sets of agglutinin genes, autophagy genes, and the NCR or glucose-repressed genes that are up-regulated by *dcp2Δ* belong to the iESR (*Figure 7—figure supplement 2C-F*).

## Dcp2 stimulates expression of genes involved in protein synthesis, glycosylation, and the unfolded protein response on rich medium

In addition to reducing production of ribosomes, *dcp2Δ* also diminishes mRNA and RPF levels of genes encoding various translation initiation and elongation factors, and of proteins involved in sulfate assimilation into methionine and cysteine (*Figure 7—source data 1B, H*), which together should reduce rates of protein synthesis in the mutant cells. The lower expression of sulfur assimilation genes involves reductions in both transcript abundance and TE (*Figure 7—figure supplement 3A*), as exemplified by *MET5* (*Figure 7—figure supplement 3B*). The abundance of mRNAs and RPFs for numerous genes involved in protein glycosylation also is reduced by *dcp2Δ* (*Figure 7—source data 1B, H*; *Figure 7—figure supplement 3C*). The suppression of both synthesis and glycosylation of trans-membrane and secreted proteins in the endoplasmic reticulum predicted by these reductions

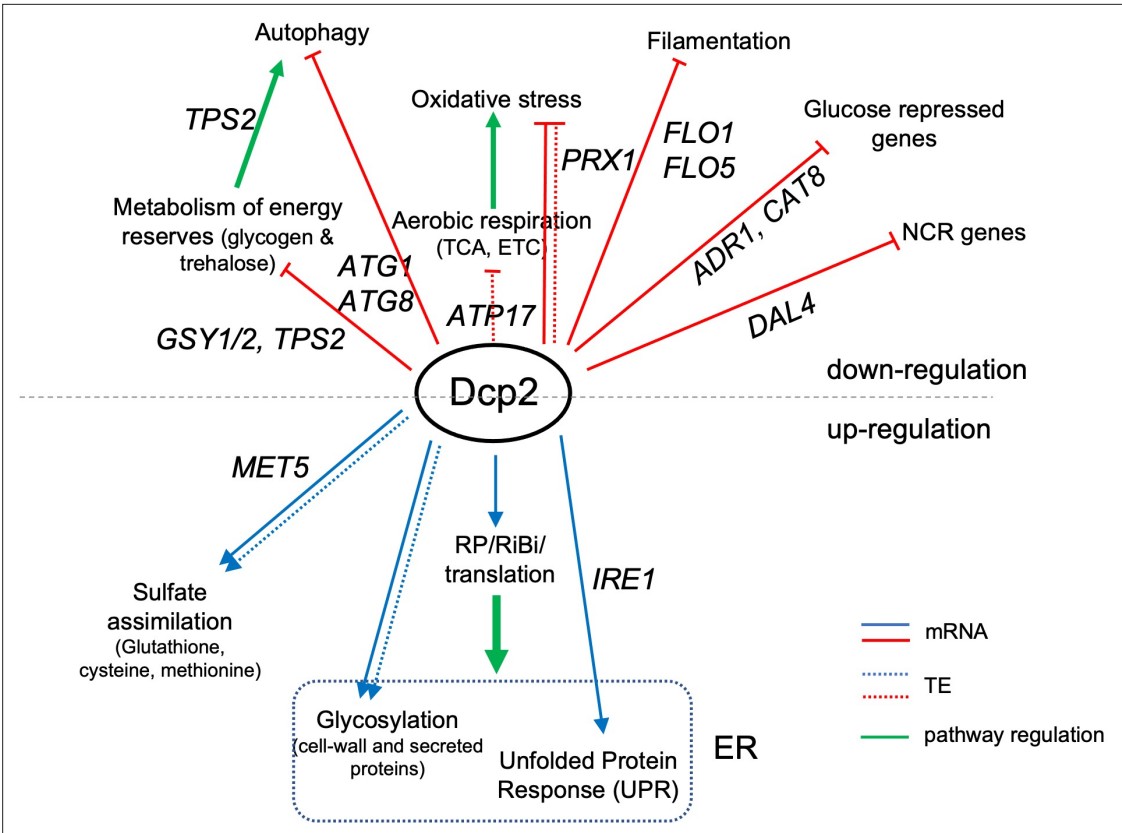

**Figure 8.** Summary of genes and pathways regulated directly or indirectly by Dcp2. Categories above the black dotted line are genes or pathways down-regulated by Dcp2 while those at the bottom are up-regulated in WT cells grown on rich medium. Straight and dotted lines represent changes in mRNA abundance and TE, respectively. Green arrows indicate pathway regulation. Important genes in each functional category are shown. ER, endoplasmic reticulum.

The online version of this article includes the following source data and figure supplement(s) for figure 8:

**Figure supplement 1.** Model depicting different decapping complexes targeting the Dhh1-dependent and -independent subsets of mRNAs repressed in abundance by Dcp1/Dcp2.

**Figure supplement 2.** Evidence that decapping by Dcp2 contributes to the iESR.

**Figure supplement 2—source data 1.** Lists of transcripts for iESR, Msn2-iESR, non-Msn2-iESR, and Msn2-target genes (*Figure 8—figure supplement 1A, B*).

**Figure supplement 3.** Dcp2 primarily represses the translation but not abundance of a group of high-confidence Puf3-repressed mRNAs highly enriched for mitochondrial proteins.

**Figure supplement 3—source data 1.** Mitochondrial protein mRNAs that bind to Puf3 and also exhibit increased expression of the encoded proteins in *puf3Δ* cells (Puf3 'cis' targets) (*Figure 8—figure supplement 3*).

might explain the observed down-regulation by *dcp2Δ* of genes involved in the unfolded protein response (UPR) (*Figure 7—source data 1B, H*, *Figure 7—figure supplement 3D*), including protein kinase *IRE1* that promotes splicing and activation of UPR transcription factor Hac1 (*Figure 7—figure supplement 3E*). *Figure 8* summarizes the major cellular pathways that are positively or negatively regulated by Dcp2.

## Discussion

In this study, we determined the genome-wide changes in mRNA abundance and translation conferred by eliminating or impairing the activity of the catalytic subunit of the mRNA decapping enzyme, Dcp2, or eliminating the decapping activator Dhh1. Roughly one-half of the mRNAs increased in abundance by *dcp2Δ* were similarly increased by *dhh1Δ*, with no further derepression seen in the *dcp2Δdhh1Δ* double mutant (*Figure 1E, F*), as expected if these mRNAs are dependent on Dhh1 to stimulate their

decapping by Dcp1/Dcp2 and attendant 5′–3′ degradation by Xrn1. Generally, these Dhh1-dependent mRNAs were also elevated by elimination of the decapping activators Pat1 or both Edc3 and Scd6 (*Figure 2A, B*), suggesting that these four factors frequently act together to stimulate decapping, such that eliminating any one stabilizes most of the entire group of transcripts. One way to account for this finding is to propose that the autoinhibition of Dcp2 activity exerted by its C-terminal IDR must be overcome by simultaneous, independent interactions of these four decapping activators with the IDR. In this view, an active decapping complex comprised of the Dcp1/Dcp2 holoenzyme and Dhh1, Pat1, and Edc3 or Scd6 would be assembled on such mRNAs to accelerate decapping and degradation in WT cells (*Figure 8—figure supplement 1A, B*). This interpretation is consistent with recent findings indicating that repression of certain Dhh1-target mRNAs is mediated by Edc3 and Scd6 acting interchangeably to recruit Dhh1 to the same segment of the Dcp2 IDR (*He et al., 2022*). Close inspection of the mRNA changes in *Figure 2B* suggests that certain of the Dhh1-dependent transcripts are targeted differentially by Pat1 vs. Dhh1, Edc3, or Scd6, which is also consistent with evidence for distinct complexes of Dcp2 bound to particular subsets of these decapping activators together with Xrn1 (*He et al., 2022*). The remaining one-half of the mRNAs up-regulated by *dcp2Δ* appear to be degraded independently of Dhh1, Pat1, and Edc3/Scd6, whereas most are elevated in mutants lacking Upf proteins to the same extent as in *dcp2Δ* cells. These latter transcripts thus appear to be among the natural NMD substrates defined previously (*Celik et al., 2017*; *Figure 8—figure supplement 1C*). Our data seem at odds with the model proposed by *He et al., 2022* that the Dcp1/Dcp2 complex containing Upf1 also contains Edc3; however, it is possible that Edc3 is present in the complex but dispensable for Upf1-mediated targeting of most Dhh1-independent mRNAs, as observed here (*Figure 2C*).

As would be expected if their abundance is controlled by decapping, Dcp2-repressed transcripts exhibit highly similar increases in response to *dcp2* point mutations that inactivate Dcp2 catalytic activity compared to the *dcp2Δ* mutation (*Figure 1D*). Moreover, combining CAGE sequencing with RNA-Seq revealed that these mRNAs exhibit greater than average proportions of uncapped molecules in WT cells, regardless of whether they are dependent or independent of Dhh1 for degradation (*Figure 3D*). Accumulation of decapped intermediates presumably reflects the fact that 5′–3′ degradation by Xrn1 is frequently delayed, for example by the presence of elongating ribosomes that initiated translation prior to decapping (*Pelechano et al., 2015*). Hence, mRNAs preferentially targeted for degradation by decapping should exhibit a greater than average proportion of uncapped transcripts in WT cells that is eliminated by *dcp2Δ*, which we observed for both Dhh1-dependent and -independent mRNAs repressed by Dcp2 (*Figure 3E*). Interestingly, the mRNA_up_*dcp2Δ* transcripts are significantly depleted for exosome substrates defined previously based on association with exosome subunits Rrp44, Rrp41, and Rrp6 (*Schneider et al., 2012*), supporting the presumption that these transcripts are primarily degraded 5′ to 3′ by Xrn1 following decapping rather than 3′ to 5′ by the exosome.

Our parallel analysis of spike-in normalized Pol II occupancies in chromatin by Rpb1 ChIP-Seq and of mRNA abundance by ERCC-normalized RNA-Seq revealed that the great majority of transcripts elevated in *dcp2Δ* cells exhibit reduced, rather than increased Pol II occupancies in the CDSs (*Figure 3G*), confirming that decreased mRNA turnover rather than increased transcription underlies the elevated mRNA levels. Even for iESR genes activated by transcription factor Msn2, which show slightly greater Rpb1 occupancies in *dcp2Δ* cells, the median increases in mRNA levels greatly exceed the increased Rpb1 occupancies (3.71- vs. 1.16-fold; cf. col. 3 in *Figure 8—figure supplement 2A* vs. *Figure 3—figure supplement 4D*), implying a substantial contribution from reduced decapping/degradation in elevating transcript levels. Supporting this, iESR transcripts, whether Msn2 activated or otherwise, exhibit greater than average proportions of uncapped mRNAs (lower C/T ratios) in WT cells that are reversed by *dcp2Δ* (*Figure 8—figure supplement 2B*, cols. 3–4 vs. 1), indicating preferential targeting by Dcp2 for decapping/decay. The entire cohort of 199 Msn2 targets, most of which are not iESR genes, does not show such evidence for enhanced decapping (*Figure 8—figure supplement 2B*, col. 5 vs. 1). Thus, the relatively greater transcript derepression of iESR- vs. non-iESR Msn2 targets in *dcp2Δ* cells (*Figure 8—figure supplement 2A* and *Figure 3—figure supplement 4D*, cols. 3–4) likely reflects a combination of Msn2-dependent transcriptional activation (in response to slow growth) and loss of Dcp2-mediated mRNA decay.

The mRNAs dependent on Dhh1 for Dcp2 repression of abundance generally exhibit average TEs in WT cells, which seems inconsistent with the possibility that enhanced decapping and degradation results from low rates of translation initiation and attendant greater access of the cap structure to Dcp1/Dcp2. These mRNAs also exhibit an average proportion of non-preferred codons (*Figure 2E*), suggesting that pausing during translation elongation is not a key driver of their enhanced decapping and degradation. As discussed further below, a third possibility is that Dcp1/Dcp2 are recruited to these mRNAs by one or more RNA-binding proteins that interacts with specific sequences in their 3'UTRs. The Dhh1-independent group of mRNAs, by contrast, tend to have lower than average TEs and codon optimality (*Figure 2D, E*), as noted previously for natural NMD substrates (*Celik et al., 2017*). Presumably, aberrant splicing or transcription initiation, or non-canonical translation initiation or elongation, leads to out-of-frame translation and premature termination that triggers activation of Dcp1/Dcp2 by the Upf proteins on most Dhh1-independent mRNAs. Interestingly, this group is enriched for transcripts involved in DNA repair or recombination (*Figure 7—source data 1D*), distinct from the cellular functions involving Dhh1-dependent mRNAs, which include stress responses, metabolism of energy reserves, and the TCA cycle (*Figure 7—source data 1C*).

Our ribosome profiling analysis revealed widespread changes in translation of many mRNAs in *dcp2Δ* cells (*Figure 4A, B*). The increased ribosome density observed for the Dcp2 translationally repressed mRNAs in *dcp2Δ* vs. WT cells was generally associated with increased steady-state protein abundance measured by TMT-MS/MS (*Figure 4C*), which was supported for particular mRNAs by increased expression of *nLUC* reporters for the corresponding genes (*Figure 4D*). As a group, the Dcp2 translationally repressed mRNAs do not appear to be preferentially targeted by Dcp2 for mRNA turnover, as they exhibit decreased rather than increased relative abundance in *dcp2Δ* vs. WT cells (*Figure 5A*). At odds with the possibility of translational repression via decapping (*Figure 5B*), Dcp2 translationally repressed mRNAs exhibit higher, not lower, than average proportions of capped mRNAs in WT cells (*Figure 5C*). Searching for a different mechanism, we noted that these mRNAs tend to be efficiently translated in WT cells, and also that *dcp2Δ* cells have two key attributes that could confer increased translation of such mRNAs owing to increased competition for limiting ribosomes. First, *dcp2Δ* cells have reduced ribosome content (*Figure 5F, G*), which probably reflects reduced expression of rESR transcripts—highly enriched for mRNAs encoding ribosomal proteins or biogenesis factors (*Gasch et al., 2000 Figure 5—figure supplement 1C, E*). Although RPG transcripts show increased TEs in *dcp2Δ* cells, this is outweighed by their reduced transcript abundance, to yield a net decrease in translation (RPF occupancies) in *dcp2Δ* vs. WT cells (*Figure 5—figure supplement 1E*) and ~30% reduced ribosome content (*Figure 5G*). Second, *dcp2Δ* cells have ~24% higher than WT levels of total mRNA, as determined by spike-in normalized RNA-Seq (*Figure 3F*), as might be expected from decreased decapping and mRNA turnover. It has been reported that reduced mRNA turnover in mutants lacking mRNA decay factors is buffered by decreased transcription (*Sun et al., 2013*), which likely limits the increase in mRNA levels we observed in *dcp2Δ* cells. The decreased ribosome content coupled with increased capped mRNA abundance should enhance competition among mRNAs for limiting 43S PICs and favor mRNAs most efficiently translated in WT cells (*Figure 5E*). Supporting this idea, first predicted by mathematical modeling (*Lodish, 1976*), the TE changes conferred by *dcp2Δ* are mimicked by other conditions that limit 43S PIC assembly: phosphorylation of eIF2α and impairment of 40S recycling at stop codons by eliminating Tma64/Tma20 (*Figure 5—figure supplement 1A, B*; *Gaikwad et al., 2021*). While other explanations are possible, this indirect mechanism is a plausible way to account for the bulk of translational programming conferred by eliminating Dcp2.

There is precedent for our proposal that elevated mRNA levels in *dcp2Δ* cells promote translational reprogramming. *Holmes et al., 2004* showed that suppression of bulk translation initiation under different stress conditions is diminished in various mutants that either lack decapping factors, accumulate free 40S subunits, or overexpress eIF4G, and concluded that increased amounts of mRNA, free 40S subunits or eIF4G can drive assembly of 48S PICs and counteract regulatory responses (e.g., eIF2α phosphorylation) that diminish PIC assembly during stress. It is possible therefore that the increased mRNA levels conferred by *dcp2Δ* mitigates the effects of decreased ribosome levels conferred by the ESR on 48S PIC formation; however, our results suggest that the increased mRNA:40S ratio still favors translation of the mRNAs best able to compete for limiting PICs in *dcp2Δ* cells.

Just as observed for changes in mRNA abundance, alterations in TE conferred by *dcp2Δ* can occur dependently or independently of Dhh1, with the minority, Dhh1-dependent group showing relatively

greater translational repression by Dcp2 (*Figure 6A, B*). The Dhh1-independent group exhibits the key features associated with the indirect mechanism of translational repression involving increased competition for limiting PICs (*Figure 6C, D*). The ~100 mRNAs belonging to the Dhh1-dependent group, by contrast, exhibit much lower than average TEs in WT (*Figure 6C*), which seems incompatible with the competition mechanism. They are however preferentially targeted by Dcp2 and Dhh1 for degradation (*Figure 6—figure supplement 1B, C*) and show higher than average proportions of uncapped degradation intermediates in WT, which are reversed by *dcp2Δ* and *dhh1Δ* (*Figure 6D–F*)—features compatible with the model of translational repression via decapping (*Figure 5B*).

An alternative mechanism to account for the ~100 Dhh1-dependent Dcp2 translationally repressed mRNAs would be to propose that Dcp2 and Dhh1 sequester them in PBs, where translationally silent mRNAs can be enriched (*Luo et al., 2018*), without stimulating their decapping. In this model, association with Dcp2/Dhh1 would be sufficient for translational repression without a requirement for Dcp2 catalytic activity. This mechanism seems unlikely because PBs are rare in non-stressed WT cells growing exponentially in rich medium (*Teixeira and Parker, 2007*). Furthermore, our Ribo-Seq analysis of the *dcp2-EE* catalytic mutant indicates highly similar TE changes conferred by this point mutation and the *dcp2Δ* deletion (*Figure 5—figure supplement 1D*). However, we cannot rule out the scenario that *dcp2-EE* impairs Dcp2 association with mRNAs to reduce PB formation, nor that the increased mRNA abundance conferred by *dcp2-EE* interferes with the ability of Dcp2 to sequester this specific subset of translationally repressed mRNAs in PBs. Additional work is required to elucidate the mechanism of Dhh1-dependent translational repression by Dcp2.

Accumulation of mRNA degradation intermediates in *dcp1Δ* or *xrn1Δ* cells greatly elevates PB assembly during exponential growth in glucose medium (*Sheth and Parker, 2003*; *Teixeira and Parker, 2007*). Our RNA-Seq and Ribo-Seq experiments involved centrifugation of cell extracts to remove cellular debris, which might have depleted mRNAs associated with PBs preferentially in *dcp2Δ* vs. WT cells (*Kershaw et al., 2021*). This may not be an important limitation of our study however, because *dcp2Δ* leads to much smaller PBs compared to *dcp1Δ* or *xrn1Δ* in glucose medium (*Teixeira and Parker, 2007*). Moreover, *dhh1Δ* does not increase PBs in glucose-grown cells (*Teixeira and Parker, 2007*), yet we observed highly similar up-regulation of Dhh1-dependent mRNAs between *dcp2Δ* and *dhh1Δ* cells (*Figure 2B*).

*S. cerevisiae* preferentially derives energy from glucose via fermentation, producing ethanol, and represses aerobic respiration in glucose-rich medium such as YPD. Importantly, we found that Dcp2-translationally repressed mRNAs are enriched for transcripts encoding mitochondrial proteins involved in respiration, including a majority of those encoding components of the ETC and mitochondrial ATPase; and that eight mRNAs encoding TCA cycle enzymes and three encoding ETC components are also repressed by Dcp2 (*Figure 7—source data 1A, B*). Indeed, a group of ~50 such mitochondrial proteins directly involved in oxidative phosphorylation (OXPHOS) shows increased median ribosome occupancies (*Figure 7B*) and steady-state protein abundance (*Figure 7—figure supplement 1K*) in *dcp2Δ* vs. WT cells, which we confirmed for several proteins by western blotting (*Figure 7C*). These findings suggest that respiration is up-regulated in the *dcp2Δ* mutant on YPD medium, which was supported by demonstrating increased mitochondrial membrane potential in *dcp2Δ* vs. WT cells (*Figure 7D*).

*S. cerevisiae* also preferentially utilizes glucose and represses catabolism of alternative carbon sources on glucose-replete medium, in part by deactivating transcriptional activators Adr1 and Cat8 (*Zaman et al., 2008*). We observed increased mRNA abundance and translation of *ADR1, CAT8* and a group of ~100 glucose-repressed or Adr1/Cat8 target genes (*Figure 7—source data 1C, D* and *Figure 7E*) in *dcp2Δ* cells. Similarly, a group of ~36 genes that are transcriptionally repressed in the presence of preferred nitrogen sources (NCR genes) showed elevated median mRNA levels and translation in *dcp2Δ* cells on YPD medium (*Figure 7F*). These findings suggest that Dcp2 post-transcriptionally enhances fermentation of glucose and catabolism of preferred nitrogen sources on rich medium. Consistent with this, Dcp2 helps to suppress the generation of amino acids from proteins via autophagy on rich medium by repressing levels of certain *ATG* mRNAs encoding proteins required specifically for this process (*Hu et al., 2015*), which we confirmed here (*Figure 7—figure supplement 1F, G*). Interestingly, *dcp2Δ* also increased the abundance of multiple cell adhesion genes and conferred increased cell filamentation and invasive growth on rich medium (*Figure 7—figure supplement 1H* and *Figure 7G–I*), processes that are normally suppressed when nutrients are plentiful

(*Roberts and Fink, 1994*). The same phenotypes are also displayed to a lesser degree by elimination of Pat1 (*Figure 7—figure supplement 1I, J*). If filamentation and invasive growth are regarded as strategies for nutrient foraging, then the repression of these processes by Dcp2 and Pat1 on rich medium supports the model that mRNA decapping and attendant translational repression by Dcp1/Dcp2 participates in repressing multiple pathways that are dispensable in cells growing on nutrient-rich medium.

Previously, it was shown that the yeast Pumilio family RNA-binding protein Puf3 coordinately represses the expression of multiple mRNAs encoding mitochondrial proteins that support the synthesis or assembly of the complexes of OXPHOS during fermentative growth on glucose. Puf3-targeted mRNAs encode components of mitochondrial ribosomes and factors involved in import, folding, or maturation of nuclear-encoded mitochondrial proteins, but only two proteins (Atp1 and Atp18) directly involved in OXPHOS (*Lapointe et al., 2018*). Puf3 reduces the abundance of these mRNAs by enhancing their deadenylation, decapping and degradation in glucose-grown cells (*Olivas and Parker, 2000*; *Miller et al., 2014*). We found that Dcp2 represses a sizeable group of 52 OXPHOS transcripts, primarily at the level of translation (*Figure 7B*) in a manner consistent with the indirect ribosome competition model for translational reprograming (*Figure 5E*), as they exhibit greater than average median TE in WT cells. About 10 of these transcripts are also repressed by Dcp2 at the level of mRNA abundance, but the two Puf3 targets *ATP1* and *ATP18* are not among them. It seems unlikely therefore that Puf3 plays a key role in the Dcp2-mediated repression of OXPHOS mRNAs.

Interrogating a group of 91 mRNAs encoding mitochondrial proteins that both bind Puf3 and exhibit increased protein expression in *puf3Δ* cells (Puf3 'cis' targets of Lapointe et al.), we found that most displayed derepression of TE but not mRNA in *dcp2Δ* cells (*Figure 8—figure supplement 3*), with only five exhibiting comparable >1.5-fold increases in RPF and mRNA abundance: *IMG2*, *SDH5*, *MPM1*, *CIR1*, and *MSC6*. Thus, it seems likely that Puf3 could be involved in Dcp2-mediated mRNA degradation for only these five Puf3 'cis' targets. While Puf3 might play a role in repressing the translation of the other Puf3 'cis' targets whose TE is increased by *dcp2Δ*, these mRNAs are well translated in WT cells and may conform to the ribosome competition mechanism instead. It seems plausible that other RNA-binding proteins besides Puf3 mediate the repression of mRNA abundance by Dcp2 for many of the ~1330 other transcripts in the mRNA_up_*dcp2Δ* group or members of the 100 mRNAs translationally repressed by Dcp2 in a manner requiring Dhh1, which were identified in our study.

## Materials and methods

### Key resources table

| Reagent type (species) or resource | Designation | Source or reference | Identifiers | Additional information |
|---|---|---|---|---|
| Chemical compound, drug | Cycloheximide | Sigma | Cat # C7698 | |
| Chemical compound, drug | 5-Fluoro-orotic acid | USBiological | Cat # F5050 | |
| Commercial assay or kit | NEBuilder HiFi DNA assembly | New England Biolabs | Cat # E2621S | |
| Commercial assay or kit | RNase I | Ambion | Cat # AM2294 | |
| Commercial assay or kit | RNA Clean and Concentrator kit | Zymo | Cat # R1018 | |
| Commercial assay or kit | T4 Polynucleotide kinase | New England Biolabs | Cat # M0201L | |
| Commercial assay or kit | T4 Rnl2(tr) K227Q | New England Biolabs | Cat # M0351S | |
| Commercial assay or kit | 5′ deadenylase/RecJ exonuclease | Epicentre | Cat # RJ411250 | |
| Commercial assay or kit | Oligo Clean and Concentrator column | Zymo Research | Cat # D4060 | |

*Continued on next page*

*Continued*

| Reagent type (species) or resource | Designation | Source or reference | Identifiers | Additional information |
|---|---|---|---|---|
| Commercial assay or kit | Protoscript II | New England Biolabs | Cat # M0368L | |
| Commercial assay or kit | CircLigase ssDNA Ligase | Epicenter | Cat # CL4111K | |
| Commercial assay or kit | Phusion polymerase | New England Biolabs | Cat # M0530S | |
| Commercial assay or kit | High Sensitivity DNA Kit | Agilent | Cat # 5067-4626 | |
| Commercial assay or kit | Fragmentation Reagent | Ambion | Cat # AM8740 | |
| Commercial assay or kit | Stop Solution | Ambion | Cat # AM8740 | |
| Commercial assay or kit | Ribo-Zero Gold rRNA Removal Kit | Illumina | Cat # MRZ11124C | |
| Commercial assay or kit | RNA 6000 Nano kit | Agilent | Cat # 5067-1511 | |
| Commercial assay or kit | ERCC ExFold RNA spike-In Mixes | Ambion | Cat # 4456739 | |
| Commercial assay or kit | DNA Library Prep Kit | New England Biolabs | Cat # E7370L | |
| Commercial assay or kit | DNase I | Roche | Cat # 4716728001 | |
| Commercial assay or kit | Luciferase Control RNA | Promega | Cat # L4561 | |
| Commercial assay or kit | Superscript III First-Strand synthesis kit | Invitrogen | Cat # 18080051 | |
| Commercial assay or kit | Brilliant II SYBR Green qPCR Master Mix | Agilent | Cat # 600828 | |
| Commercial assay or kit | 25 mM triethylammonium-bicarbonate | Thermo Scientific | Cat # 90114 | |
| Commercial assay or kit | GelCode Blue Stain | Thermo Scientific | Cat # 24592 | |
| Commercial assay or kit | Pierce BCA Protein Assay Kit | Thermo Scientific | Cat # 23225 | |
| Commercial assay or kit | Protease inhibitor cocktail | Roche | Cat # 5056489001 | |
| Commercial assay or kit | Enhanced chemiluminescence (ECL) | Cytiva | Cat # RPN2109 | |
| Commercial assay or kit | Nano-Glo substrate | Promega | Cat # N1120 | |
| Commercial assay or kit | Bradford reagent | Bio-Rad | Cat # 5000006 | |
| Antibody | anti-Idh1 (Goat polyclonal) | Abnova | Cat # PAB19472 | Dilution: 1:500 (WB) |
| Antibody | anti-GAPDH (Mouse monoclonal) | Proteintech | Cat # 60004 | Dilution: 1:30,000 (WB) |
| Antibody | anti-rabbit IgG (HRP-conjugated, donkey polyclonal) | GE | Cat # NA9340V | Dilution: 1:10,000 (WB) |
| Antibody | anti-mouse IgG (HRP-conjugated, sheep polyclonal) | GE | Cat # NA931V | Dilution: 1:10,000 (WB) |
| Antibody | anti-goat IgG (HRP-conjugated, chicken polyclonal) | Abnova | Cat # PAB29101 | Dilution: 1:10,000 (WB) |

*Continued*

| Reagent type (species) or resource | Designation | Source or reference | Identifiers | Additional information |
|---|---|---|---|---|
| Antibody | anti-Rpb1 (Mouse monoclonal) | 8WG16, Biolegend | Cat # 664906 | 4 μl used in Rpb1-ChIP |
| Strain, strain background (*Saccharomyces cerevisiae*) | *Supplementary file 1a* | This paper | | Strains used |
| Genetic reagent (plasmid) | *Supplementary file 1b* | This paper | | Plasmids used |
| Sequence-based reagent | *Supplementary file 1c* | This paper | | Primers used |
| Sequence-based reagent | *Supplementary file 1d* | This paper | | *nLUC* reporter genes |
| Sequence-based reagent | *Supplementary file 1e* | This paper | | *nLUC* CDS |
| Software, algorithm | Notched box-plots | http://shiny.chemgrid.org/boxplotr/ | | |
| Software, algorithm | Venn diagrams | https://www.biovenn.nl/ | | |
| Software, algorithm | Hypergeometric distribution | https://systems.crump.ucla.edu/hypergeometric/index.php | | |
| Software, algorithm | Volcano plots | https://huygens.science.uva.nl/VolcaNoseR/ | | |
| Software, algorithm | Gene ontology (GO) | http://funspec.med.utoronto.ca/ | | |
| Software, algorithm | DESeq2 | https://github.com/hzhanghenry/RiboProR | | |
| Software, algorithm | Integrative Genomics Viewer | IGV 2.4.14, http://software.broadinstitute.org/software/igv/ | | |
| Software, algorithm | Genome-wide occupancy profiles for Rpb1 (ChIP) | https://github.com/hzhanghenry/OccProR | | |
| Software, algorithm | Image Lab 6.0.1 program | https://www.bio-rad.com/en-us/product/image-lab-software?ID=KRE6P5E8Z | | |
| Software, algorithm | SwissProt Yeast database | https://www.uniprot.org/proteomes/UP000002311 | | |
| Software, algorithm | Proteome Discoverer 2.4 | Thermo Scientific | | |

## Strains, plasmids, and culture conditions

Yeast strains, plasmids, and primers used in the study are listed in *Supplementary file 1a-1c*, respectively. Yeast strain VAK022 harboring *dcp2-E149Q,E153Q* (*dcp2-EE*) was constructed in two steps. First, mutations G to C at positions 445 and 457 were introduced into *DCP2* in plasmid pQZ145 (*Zeidan et al., 2018*) using primers AKV005/AKV006 and the Quick-change Site-Directed mutagenesis kit (Agilent, 200519), generating plasmid pAV008 containing *dcp2-E149Q,E153Q*. Next, a 3.3-kb fragment containing *dcp2-E149Q,E153Q* with 400-bp upstream of the start codon (containing a Pf1N1 restriction site) was PCR amplified using primers AKV088/AKV090, and the resultant fragment was inserted between the BamHI and SacI sites of integrative vector YIplac211 to produce plasmid pAKV013. The DNA fragment generated by digestion of pAKV013 with Pf1N1 was used to transform WT strain W303 to Ura[+]. Finally, strain VAK022 was obtained from one such Ura[+] transformant by selecting for loss of the *URA3* marker via homologous recombination by counter-selection on

medium containing 5-fluoro-orotic acid. The replacement of *DCP2* with *dcp2-E149Q, E153Q* was verified by sequencing the PCR product obtained from chromosomal DNA amplified with the primer pairs AKV010/AKV015.

The 13 reporter plasmids containing *nLUC* reporters listed in ***Supplementary file 1d*** were constructed to fuse the nanoLUC CDSs (codon optimized for *S. cerevisiae*, ***Supplementary file 1e***), preceded by the GGG glycine codon, to the final codon of the complete CDS of each gene, preserving the native stop codon, 3′UTR sequences, and segment of 3′-noncoding sequences of the gene, as well as a segment of 5′-noncoding sequences including the native promoter and 5′UTR sequences, of the gene of interest (fragment lengths of 5′ and 3′ region listed in ***Supplementary file 1d***), by the following three-step procedure. First, DNA fragments were synthesized containing a SmaI site, the nLUC CDS (including the ATG and stop codon), the native stop codon and 3′ noncoding sequences for each gene of interest, and an EcoRI site, and inserted between the SmaI and EcoRI sites of pRS316 to produce an intermediate plasmid for each gene of interest. Second, PCR amplification from WT yeast genomic DNA was conducted to generate a fragment for each gene of interest containing 20 nt of pRS316 adjacent to the SmaI site, the CCC nucleotides of the SmaI site, the 5′-noncoding region, 5′UTR, and CDS of the relevant gene (excluding the stop codon), the CCC complement of the GGG glycine codon, and the first 20nt of the *nLUC* CDS. Third, Gibson assembly was used to insert the PCR-amplified fragments from the second step at the SmaI sites of the intermediate plasmids, using 2x ExSembly Cloning Master mix (LifeSct LLC, M0005) and following the vender's instructions except that a 15-min incubation at 25°C was included prior to the 37°C incubation to allow for optimal SmaI digestion.

Plasmid pNG158 contains the *PAT1* CDS, with 500 bp upstream and 192 bp downstream, on a 3083-bp fragment amplified by PCR from yeast genomic DNA, and inserted between the SalI and XmaI sites of pRS315. The plasmids were constructed by NEBuilder HiFi DNA assembly (New England Biolabs) according to the manufacturer's protocol.

Unless mentioned otherwise, strains were cultured in YPD medium to mid-exponential growth phase (OD$_{600}$ ~ 0.6) before harvesting.

## Ribosome profiling and parallel RNA sequencing

Ribosome profiling and RNA-Seq analysis were conducted in parallel, essentially as described previously (***McGlincy and Ingolia, 2017***), using isogenic strains W303 (WT), CFY1016 (*dcp2Δ*), and VAK023 (*dcp2-EE*) with two biological replicates performed for each genotype. Cells were harvested by vacuum filtration and flash-frozen in liquid nitrogen. Cells were lysed in a freezer mill in the presence of lysis buffer (20 mM Tris (pH 8), 140 mM KCl, 1.5 mM MgCl$_2$, 1% Triton X-100) supplemented with 500 mg/ml cycloheximide. Lysates were cleared by centrifugation at 3000 × *g* for 5 min at 4°C, and the resulting supernatant was subjected to centrifugation at 15,000 × *g* for 10 min at 4°C, flash-frozen in liquid nitrogen, and stored at −80°C.

For preparation of libraries of RPFs, 50 $A_{260}$ units of cell lysates were digested with 450 U of RNase I (Ambion, AM2294) for 1 hr at room temperature (RT, ~25°C) on a Thermomixer at 700 rpm, and extracts were resolved on 10–50% sucrose gradients by centrifugation for 160 min at 39,000 rpm, 4°C in a Beckman SW41Ti rotor. Gradients were fractionated at 0.75 ml/min with continuous monitoring of $A_{260}$ values using a Biocomp Instruments Gradient Station. RNA was purified from the 80S monosome fractions using an RNA Clean and Concentrator kit (Zymo, R1018), resolved by electrophoresis on a 15% TBE-Urea gel, and 25–34 nt fragments isolated from the gel were dephosphorylated using T4 Polynucleotide kinase (New England Biolabs, M0201L). For each library sample, barcoded 5′-pre-adenylated linkers were added to the 3′ ends of footprints using T4 Rnl2(tr) K227Q (New England Biolabs, M0351S), and excess unligated linker was removed using 10 U/μl 5′ deadenylase/RecJ exonuclease (Epicentre, RJ411250), followed by pooling and purification of ligated footprints using an Oligo Clean and Concentrator column (Zymo Research, D4060). Ribosomal RNA contamination was removed by biotinylated primers (***McGlincy and Ingolia, 2017***), followed by reverse transcription using Protoscript II (New England Biolabs, M0368L) and circularization of cDNA using CircLigase ssDNA Ligase (Epicenter, CL4111K). Each pooled library was PCR amplified using Phusion polymerase (F-530) (New England Biolabs, M0530S). Quality of the libraries was assessed with a Bioanalyzer using the High Sensitivity DNA Kit (Agilent, 5067-4626) and quantified by Qubit. Single-end 50 bp sequencing was done on an Illumina HiSeq system at the NHLBI DNA Sequencing and Genomics Core at NIH (Bethesda, MD).

For RNA-Seq library preparation, total RNA was extracted and purified from aliquots of the same snapped-frozen cells described above using the Zymo RNA Clean and Concentrator kit. Five mg of total RNA was randomly fragmented by incubating with Ambion Fragmentation Reagent (Ambion, AM8740) at 70°C for 12 min. The reaction was stopped by adding Stop Solution (Ambion, AM8740) and precipitated by adding one part of isopropanol followed by precipitation with 70% ethanol. Fragment size selection, library generation, and sequencing were carried out using the same protocol described above for RPF library preparation, except that Ribo-Zero Gold rRNA Removal Kit (Illumina, MRZ11124C) was employed to remove rRNA after linker ligation.

As described earlier (*Martin-Marcos et al., 2017*), Illumina sequencing reads were trimmed to remove the constant adapter sequence, mixed sample sequences were separated by the sample barcodes followed by removal of PCR duplicates using a custom Python (3.7) script. The sequences aligned to yeast noncoding RNAs were removed using bowtie (*Langmead et al., 2009*) and non-rRNA reads (unaligned reads) were then mapped to the *S. cerevisiae* genome (R64-1-1 S288C SacCer3 Genome Assembly) using TopHat (*Langmead et al., 2009*). Only uniquely mapped reads from the final genomic alignment were used for subsequent analyses. Statistical analysis of changes in mRNA, RPFs, or TE values between two replicates each of any two strains being compared was conducted using DESeq2 (*Love et al., 2014*) excluding any genes with less than 10 total mRNA reads in the four samples (of two replicates each) combined. DESeq2 is well suited to identifying changes in mRNA or RPF expression, or TEs, with very low incidence of false positives using results from only two highly correlated biological replicates for each of the strains/conditions being compared (*Zhang et al., 2014*; *Lamarre et al., 2018*). The R script employed for DESeq2 analysis of TE changes can be found on Github (https://github.com/hzhanghenry/RiboProR; *hzhanghenry, 2023*; *Kim et al., 2019*). Wiggle files were generated as described previously (*Zeidan et al., 2018*) and visualized using Integrative Genomics Viewer (IGV 2.4.14, http://software.broadinstitute.org/software/igv/) (*Robinson et al., 2011*). Wiggle tracks shown are normalized according to the total number of mapped reads.

## Parallel CAGE and RNA sequencing

Total RNA was prepared by hot-phenol extraction (*Schmitt et al., 1990*) from two biological replicates of each strain. RNA integrity was determined using the Agilent RNA 6000 Nano kit (5067-1511) and the concentrations were measured by Nanodrop spectroscopy. CAGE libraries were constructed and sequenced following the nAnT-iCAGE protocol (*Murata et al., 2014*) by K.K. DNAFORM of Japan. Briefly, cDNAs were transcribed to the 5′ ends of capped RNAs, ligated at the 5′ and 3′ ends with barcoded linkers, followed by second strand synthesis, and the resulting DNA libraries were sequenced using the Illumina NextSeq500 (Single-end, 75 bp reads) platform. Between 20 and 25 million mapped CAGE tags were obtained for each sample (*Figure 3—figure supplement 1—source data 1*).

The sequenced CAGE reads of each sample were aligned to the reference genome of *S. cerevisiae* S288C (Assembly version: sacCer3) using HISAT2 (*Kim et al., 2019*). For read alignment, we disabled the soft clipping option in HISAT2 by using '--no-softclip' to avoid false-positive transcription start sites (TSSs). CAGE reads mapped to rRNA genes were identified using rRNAdust (http://fantom.gsc.riken.jp/5/sstar/Protocols:rRNAdust), and were excluded from subsequent TSS analyses (*Figure 3—figure supplement 1—source data 1*).

TSS identification, inference of TSS clusters (TC, representing putative core promoters), and assigning TCs to their downstream genes were carried out by using TSSr (*Lu et al., 2021*). CAGE reads with a mapping quality score (MAPQ) >20 were considered uniquely mapped reads, which were used for subsequent analyses. CAGE signals of biological replicates were then merged as a single sample. The transcription abundance of each TSS was quantified as the numbers of CAGE tags/reads supporting the TSS per million mapped reads (TPM). Only TSSs with TPM ≥0.1 were used to infer TCs, representing putative core promoters.

The 'peakclu' method (*Lu et al., 2021*) was used to infer TCs for each sample, with the following options 'peakDistance = 50, extensionDistance = 25, localThreshold = 0.01'. A set of Consensus TCs of all samples were generated by using the 'consensusCluster' function in TSSr with an option of 'dis = 100'. Consensus TCs were then assigned to their downstream genes if they are within 1000 bp upstream and 50 bp downstream of the start codon of annotated ORFs. The TPM value of a consensus

TC in a sample is the sum of TPM values of all TSSs within its range. The TPM value of a gene was calculated as the sum of all consensus TCs assigned to the gene.

RNA sequencing libraries were produced in parallel from the same RNA samples subjected to CAGE sequencing by the NHLBI DNA sequencing Core at NIH (Bethesda, MD) using the TruSeq Stranded mRNA Library Prep Kit (Illumina, Paired-end 50 bp reads) and sequenced using the NovaSeq6000 Illumina platform. Prior to library preparation, rRNA was depleted using the QIAGEN FastSelect yeast rRNA depletion kit. Sequencing reads were mapped to the S288C genome (R64-1-1 S288C SacCer3) using STAR aligner (*Dobin et al., 2013*) and PCR duplicates were removed by Samtools (*Figure 3—figure supplement 1—source data 1*).

Identification of differentially expressed genes in total RNA (ΔmRNA_T) or capped RNA (ΔmRNA_C) in *dcp2Δ* vs. WT cells was conducted by DESeq2 analysis (*Love et al., 2014*) using raw read counts from the RNA-Seq or CAGE sequencing experiments conducted in parallel on the same RNA samples. To calculate C/T ratios, RNA-Seq reads assigned to each gene were normalized as TPM reads by dividing the read counts by the length of each gene in kilobases (reads per kilobase, RPK), summing all the RPK values in a sample and normalizing, to generate 'per million' scaling factors. Further, RPK values were normalized by the 'per million' scaling factor to give TPM. Because a single read/tag is generated for each transcript in CAGE, its TPM (tags per million mapped tags) is equivalent to the TPM value obtained from RNA-Seq, allowing comparisons of the two types of TPM values. The C/T ratio of each gene was calculated by dividing CAGE TPMs by RNA-Seq TPMs in each strain after first removing all genes with zero CAGE reads in either the *dcp2Δ* or WT samples being compared. In interrogating C/T ratios for the groups of mRNAs increased or decreased by *dcp2Δ*, the same two groups defined using the RNA-Seq data obtained in parallel with ribosome profiling were examined to allow changes in mRNA abundance, C/T ratios, ribosome occupancies, and TEs to be compared for the same two sets of mRNAs. The conclusions reached regarding changes in C/T ratios were not altered if the groups were defined using the RNA-Seq data obtained in parallel with CAGE sequencing instead.

## RNA sequencing with spike-in normalization

ERCC ExFold RNA spike-In Mixes (Ambion, part no. 4456739) consisting of spike-In Mix I and spike-In Mix II were equally added to WT and *dcp2Δ* total RNA (2.4 µl of 1:100-fold diluted spike-In to 1.2 µg of total RNA), respectively, according to the manufacturer's instructions. STAR software was used to align mapped sequencing reads, including reads from the spike-In RNAs, to the S288C genome or ERCC RNA sequences. Reads obtained from the 23 spike-in transcripts belonging to subgroup B (present in equal concentrations in Mix I and Mix II) were used to calculate size factors for each library and reads corresponding to yeast genes were normalized by the size factors. DESeq2 was further employed to calculate the differential expression between strains by setting the size factor to unity.

## ChIP-Seq and data analysis

WT and *dcp2Δ* strains were cultured in triplicate in YPD medium to $A_{600}$ of 0.6–0.8 and treated with formaldehyde as previously described (*Qiu et al., 2016*). ChIP-Seq was conducted as described (*Qiu et al., 2016*) using monoclonal antibody against Rpb1 (8WG16, Biolegend, 664906). DNA libraries for Illumina paired-end sequencing were prepared using the DNA Library Prep Kit for Illumina from New England Biolabs (E7370L). Paired-end sequencing (50 nt from each end) was conducted by the DNA Sequencing and Genomics core facility of the NHLBI, NIH. Sequence data were aligned to the SacCer3 version of the genome sequence using Bowtie2 (*Langmead et al., 2009*) with parameters -X 1000 -very-sensitive, to map sequences up to 1 kb with maximum accuracy. PCR duplicates from ChIP-Seq data were removed using the samtools rmdup package. Numbers of aligned paired reads from each ChIP-Seq experiment are summarized in *Figure 3—source data 3*. Raw genome-wide occupancy profiles for Rpb1 were computed using the coverage function in R and relative occupancies were obtained by normalizing each profile to the average occupancy obtained for the relevant chromosome (https://github.com/rchereji/bamR; *Chereji, 2019*). To visualize specific loci, BigWig files of samples were loaded in the Integrative Genomics Viewer (IGV) (*Robinson et al., 2011*).

For spike-in normalization of Rpb1 ChIP-Seq data, identical aliquots of *S. pombe* chromatin were added to each *S. cerevisiae* chromatin sample being analyzed in parallel, corresponding to 10% of the DNA in the *S. cerevisiae* chromatin samples, prior to immunoprecipitating with Rpb1. As described fully in *Figure 3—source data 3*, a normalization factor for each sample was calculated by dividing

the average number of total *S. pombe* reads obtained across all samples by the total *S. pombe* reads obtained for that sample. The observed reads mapping to the *S. cerevisiae* genome were multiplied by the normalization factor to yield the spike-in normalized reads for that sample. Raw genome-wide occupancy profiles for Rpb1 were computed using the coverage function in R, wherein each profile was set with the same total 'OCC' to allow the comparison between WT and *dcp2Δ*, using the custom R script (https://github.com/hzhanghenry/OccProR; *hzhanghenry, 2022*).

## qRT-PCR analysis of mRNA abundance

Total RNA was isolated by hot-phenol extraction as previously described (*Schmitt et al., 1990*) and the concentration was determined using the Nanodrop ND-1000 spectrophotometer (Thermo Fisher). Ten µg of total RNA was treated with DNase I (Roche, 4716728001) and 40 pg of Luciferase Control RNA (Promega L4561) was added to 1 µg of DNase I-treated total RNA and subjected to cDNA synthesis using a Superscript III First-Strand synthesis kit (Invitrogen 18080051). qRT-PCR was carried out using 10-fold diluted cDNA and Brilliant II SYBR Green qPCR Master Mix (Agilent, 600828) with the appropriate primer pairs (listed in *Supplementary file 1c*) at 200 nM. Expression of each transcript was normalized to that of the luciferase spike-in RNA from at least two biological replicates.

## TMT-MS/MS analysis

Three biological replicates of WT and *dcp2Δ* were cultured in YPD medium and harvested by centrifugation for 5 min at 3000 × *g*. Whole-cell extracts (WCEs) were prepared using 8 M urea in 25 mM triethylammonium-bicarbonate (TEAB; Thermo Scientific, 90114) by washing the pellets once with the same buffer and vortexing with glass beads in the cold room for 2 min with intermittent cooling on ice water. Lysates were clarified by centrifugation at 13,000 × *g* for 30 min and the protein quality was assessed following sodium dodecyl sulfate–polyacrylamide gel electrophoresis using GelCode Blue Stain (Thermo Scientific, 24592) and quantified using Pierce BCA Protein Assay Kit (Thermo Scientific, 23225). Sample preparation and TMT-MS/MS (*Zecha et al., 2019*) was performed by the NHLBI Proteomics Core at NIH (Bethesda, MD). Briefly, 100 µg of WCEs was incubated for 1 hr at 37°C with freshly prepared dithiothreitol (DTT, 20 mM final) to reduce disulfide bridges. Alkylation was performed at RT for 1 hr with freshly made 50 mM iodoacetamide (50 mM, final) in 25 mM ammonium bicarbonate and the reaction was quenched by adding DTT (50 mM, final). Lysates were diluted 10-fold with 25 mM ammonium bicarbonate and digested with 3 µg of trypsin (Promega, v5111) overnight at 37°C. Digests were acidified by adding formic acid (1%, final) and desalted with Waters Oasis HLB 1 cc columns. Peptides were eluted from desalted samples with 1 ml of buffer E (0.1% formic acid in 50% acetonitrile) and dried in a SpeedVac. Samples were labeled with TMT reagents for multiplexing (TMT10plex label reagent set, Thermo Scientific) according to the manufacturer's instructions. Briefly, resuspended TMT reagent was added to each sample, incubated for 1 hr at RT and the reaction quenched with 8 µl of 5% hydroxylamine for 15 min. Equal amounts of each sample were combined and the pooled sample was dried in a SpeedVac. To increase the protein coverage, each set of pooled TMT samples was separated into 24 fractions using basic reverse phase liquid chromatography (bRPLC). Quantification of TMT-labeled peptides was conducted on an LTQ Orbitrap Lumos-based nanoLCMS system (Thermo Scientific) with a 2-hr gradient at 120 k resolution for MS1 and 50 K for MS2 at 38% HCD energy.

Raw data were processed using Proteome Discoverer 2.4 (Thermo Scientific) and the MS2 spectra were searched in the SwissProt Yeast database (https://www.uniprot.org/proteomes/UP000002311) using the SEQUEST search engine (*Eng et al., 1994*). Peptide spectral matches (PSM) were validated using Percolator based on *q*-values at a 1% FDR (*Brosch et al., 2009*; http://www.sanger.ac.uk/Software/analysis/MascotPercolator/). Relative abundance of each peptide in a strain is measured by normalizing to the total abundance of that peptide coming from all the strains used in the study. We determined the protein-level fold-changes based on the median of peptide-level fold-changes from the Proteome Discoverer-produced abundances.

## Measuring *nLUC* reporter expression

Nano-luciferase was assayed in WCEs as previously described (*Masser et al., 2016*). Briefly, WT and *dcp2Δ* transformants harboring the appropriate *nLUC* reporter plasmids (*Supplementary file 1b*) or empty vector pRS316 were cultured in synthetic complete medium lacking uracil (SC-Ura) to OD$_{600}$ of

~1.2. Cells were collected by centrifugation at 3000 × *g* and resuspended in 1× phosphate-buffered saline lysis buffer containing 1 mM phenylmethylsulfonyl fluoride (PMSF) and protease inhibitor cocktail (Roche, 5056489001). WCEs were prepared by vortexing with glass beads, and clarified by centrifugation at 13,000 × *g* for 30 min at 4°C. Nano-Glo substrate (Promega, N1120) was diluted 1:50 with the supplied lysis buffer and mixed with 10 µl of WCE in a white 96-well plate. Bioluminescence was determined immediately using a Centro Microplate Luminometer (Berthold), and light units were normalized by the total protein concentrations of the corresponding WCEs determined using the Bradford reagent (BioRad, 5000006).

## Polysome profiling to measure ribosome content

Three biological replicates of WT and *dcp2Δ* were cultured in 300 ml of YPD medium at 30°C to $OD_{600}$ of 1.2–1.5 and quick-chilled by pouring into centrifuge tubes filled with ice. After collecting the cells by centrifugation for 10 min at 7000 × *g*, the cell pellets were resuspended in an equal volume of Buffer A (20 mM Tris–HCl [pH 7.5], 50 mM NaCl, 1 mM DTT, 200 µM PMSF, 0.15 µM Aprotinin, 1 µM Leupeptin, 0.1 µM Pepstatin A) and WCEs were prepared by vortexing with glass beads in the cold room, followed by two cycles of centrifugation for 10 min at 3000 rpm and 15,000 rpm at 4°C, respectively. (Cycloheximide and $MgCl_2$ were omitted from the lysis buffer in order to separate 80S ribosomes into 40S and 60S subunits.) Equal volumes of cleared lysate from WT and *dcp2Δ* cultures were resolved on 5–47% (wt/wt) sucrose gradients by centrifugation at 39,000 rpm for 3 hr at 4°C in a Beckman SW41Ti rotor. Gradient fractions were scanned at 260 nm using a gradient fractionator (Biocomp Instruments, Triax), and the area under the 40S and 60S peaks were quantified using ImageJ software. To estimate the ribosomal content per cell volume, the combined areas under the 40S and 60S peaks were normalized by the $OD_{600}$ values of the starting cultures.

## Western blot analysis

For western analysis of respiratory proteins, WCEs were prepared by trichloroacetic acid extraction as previously described (*Reid and Schatz, 1982*) and immunoblot analysis was conducted as described previously (*Nanda et al., 2009*). After electroblotting to polyvinylidene fluoride (PVDF) membranes (Millipore, IPVH00010), membranes were probed with antibodies against Aco1, Atp20, Cox14, Pet10 (a kind gift from Dr. Nikolaus Pfanner), Idh1 (Abnova, PAB19472), and GAPDH (Proteintech, 60004). Secondary antibodies employed were HRP-conjugated anti-rabbit (GE, NA9340V), anti-mouse IgG (GE, NA931V), and anti-goat IgG (Abnova, PAB29101). Detection was performed using enhanced chemiluminescence (ECL, Cytiva, RPN2109) and the Azure 200 gel imaging biosystem.

## Measuring mitochondrial membrane potential

Precultures were grown in SC-Ura (to select for the *URA3* plasmids) to $OD_{600}$ of ~3.0 and used to inoculate YPD medium at $OD_{600}$ of 0.2. Cells were grown to $OD_{600}$ of ~0.6–0.8 and incubated with 500 nM TMRM for 1 hr. Cells were washed once with distilled $H_2O$ and fluorescence was measured using flow cytometry (BD LSR II) at the Microscopy, Imaging & Cytometry Resources (MICR) Core at Wayne State University. Dye fluorescence is proportional to mitochondrial membrane potential. Median fluorescence intensity of single cells was analyzed with the FlowJo software, and normalized to the $OD_{600}$ of the cultures. Data presented are in arbitrary fluorescence units normalized to $OD_{600}$ of the cultures. In control samples, 50 µM FCCP was added to cells to dissipate the membrane potential and provide a measure of non-specific background fluorescence.

## Plate-washing assay of invasive cell growth

The plate-washing assay was performed as described (*Roberts and Fink, 1994*). Briefly, plates were incubated for 4 days at 30°C and photographed before washing using the ChemiDoc XRS + molecular imager (Bio-Rad) under the blot/chemicoloric setting with no filter. Small amounts of cells were excised from colonies using a toothpick and resuspended in water for microscopic examination. Plates were also photographed after being washed in a stream of water. Invasive growth was measured by the Image Lab 6.0.1 program (Bio-Rad) using the round volume tool. Invasive growth levels were normalized for colony size and reported as the average of three independent replicates (*Vandermeulen and Cullen, 2020*; *Vandermeulen et al., 2022*). Error represents the SD. Significance was determined by Student's *t*-test, p-value <0.05.

## Data visualization and statistical analysis

Notched box-plots were constructed using a web-based tool at http://shiny.chemgrid.org/boxplotr/. In all such plots, the upper and lower boxes contain the second and third quartiles and the band gives the median. If the notches in two plots do not overlap, there is roughly 95% confidence that their medians are different. For some plots, the significance of differences in medians was assessed independently using the Mann–Whitney *U* test computed using the R Stats package in R. Scatter-plots displaying correlations between sequencing read counts from biological replicates were created using the scatterplot function in Microsoft Excel. Spearman's correlation analysis and the Student's *t*-test were conducted using Microsoft Excel. Venn diagrams were generated using the web-based tool (https://www.biovenn.nl/) and the significance of gene set overlaps in Venn diagrams was evaluated with the hypergeometric distribution using the web-based tool (https://systems.crump.ucla.edu/hypergeometric/index.php). Hierarchical clustering analysis of mRNA or TE changes in mutant vs. WT strains was conducted with the R heatmap.2 function from the R 'gplots' library, using the default hclust hierarchical clustering algorithm. Volcano plots were created using the web-based tool (https://huygens.science.uva.nl/VolcaNoseR/). GO analysis was conducted using the web-based tool at http://funspec.med.utoronto.ca/.

## Data resources for mRNA features

Analyses of mRNA features for different gene sets were conducted using the following published compilations: 5′UTR and CDS lengths (*Pelechano et al., 2013*), WT mRNA steady-state amounts (in molecules per dry cellular weight – pgDW) (*Lahtvee et al., 2017*), WT mRNA half-lives (*Chan et al., 2018*), and stAI values (*Radhakrishnan et al., 2016*).

*Zeidan et al., 2018Celik et al., 2017Radhakrishnan et al., 2016Jungfleisch et al., 2017He et al., 2018Radhakrishnan et al., 2016*

# Acknowledgements

We thank Feng He, Allan Jacobson, and Bertrand Seraphin for generous gifts of yeast strains, and Nikolaus Pfanner for gifts of antibodies. We thank Yong Chen and Marjan Gucek of the NHLBI Proteomics Core for guidance on performing TMT-MS. We are grateful to Thomas Dever, Nicholas Guydosh, and Jon Lorsch for many helpful discussions about data analysis and interpretation of results, and all members of the Hinnebusch, Lorsch, Dever, and Guydosh labs, for useful comments. This work was supported in part by the Intramural Program of the NIH. C.O. and M.L.G. were supported by NIH grant R01 HL117880; M.D.V. and P.J.C. by NIGMS grant GM098629; and X.N. and Z.L. by NSF grant 1951332 and the Saint Louis University 2022 President's Research Fund.

# Additional information

### Competing interests

Alan G Hinnebusch: Reviewing editor, *eLife*. The other authors declare that no competing interests exist.

### Funding

| Funder | Grant reference number | Author |
| --- | --- | --- |
| National Heart, Lung, and Blood Institute | HL117880 | Miriam L Greenberg<br>Chisom Onu |
| National Institute of General Medical Sciences | GM098629 | Matthew D Vandermeulen<br>Paul J Cullen |
| National Science Foundation | 1951332 | Xiao Niu<br>Zhenguo Lin |

The funders had no role in study design, data collection, and interpretation, or the decision to submit the work for publication.

## Author contributions
Anil Kumar Vijjamarri, Conceptualization, Data curation, Formal analysis, Validation, Investigation, Visualization, Methodology, Writing - original draft, Writing - review and editing; Xiao Niu, Formal analysis, Investigation; Matthew D Vandermeulen, Chisom Onu, Formal analysis, Investigation, Visualization, Writing - original draft; Fan Zhang, Neha Gupta, Formal analysis, Investigation, Writing - original draft; Hongfang Qiu, Formal analysis, Investigation, Methodology; Swati Gaikwad, Methodology; Miriam L Greenberg, Conceptualization, Formal analysis, Supervision, Writing - original draft, Project administration; Paul J Cullen, Conceptualization, Supervision, Investigation, Writing - original draft, Project administration; Zhenguo Lin, Conceptualization, Software, Formal analysis, Supervision, Visualization, Methodology, Writing - original draft, Project administration; Alan G Hinnebusch, Conceptualization, Formal analysis, Supervision, Funding acquisition, Writing - original draft, Project administration, Writing - review and editing

## Author ORCIDs
Anil Kumar Vijjamarri (iD) http://orcid.org/0009-0006-6721-3109
Chisom Onu (iD) http://orcid.org/0000-0002-3338-5141
Swati Gaikwad (iD) http://orcid.org/0000-0002-1438-9497
Alan G Hinnebusch (iD) http://orcid.org/0000-0002-1627-8395

## Decision letter and Author response
Decision letter https://doi.org/10.7554/eLife.85545.sa1
Author response https://doi.org/10.7554/eLife.85545.sa2

# Additional files

## Supplementary files
• Supplementary file 1. The yeast strains, plasmids, and primers employed in this study are listed in files 1a, 1b, and 1c, respectively. File 1d contains details of the *nLUC* reporter genes. File 1d contains the *nLUC* CDS codon optimized for *S. cerevisiae*.

• MDAR checklist

## Data availability
Sequencing data have been deposited in GEO under accession codes GSE220578 and GSE216831. All other data generated or analyzed during this study are included in the manuscript and supporting files; Source Data files have been provided for all figures.

The following datasets were generated:

| Author(s) | Year | Dataset title | Dataset URL | Database and Identifier |
|---|---|---|---|---|
| Vijjamarri AK, Niu X, Vandermeulen MD, Onu C, Zhang F, Qiu H, Gupta N, Gaikwad S, Greenberg ML, Cullen PJ, Lin Z, Hinnebusch AG | 2023 | Decapping factor Dcp2 controls mRNA abundance and translation to adjust metabolism and filamentation to nutrient availability | https://www.ncbi.nlm.nih.gov/geo/query/acc.cgi?acc=GSE220578 | NCBI Gene Expression Omnibus, GSE220578 |
| Vijjamarri AK, Niu X, Vandermeulen MD, Onu C, Zhang F, Qiu H, Gupta N, Gaikwad S, Greenberg ML, Cullen PJ, Lin Z, Hinnebusch AG | 2023 | Decapping factor Dcp2 controls mRNA abundance and translation to adjust metabolism and filamentation to nutrient availability | https://www.ncbi.nlm.nih.gov/geo/query/acc.cgi?acc=GSE216831 | NCBI Gene Expression Omnibus, GSE216831 |

The following previously published datasets were used:

| Author(s) | Year | Dataset title | Dataset URL | Database and Identifier |
|-----------|------|---------------|-------------|-------------------------|
| Jungfleisch J, Nedialkova DD, Dotu I, Sloan KE | 2016 | Ribosome profiling of dhh1Δ yeast | https://www.ncbi.nlm.nih.gov/geo/query/acc.cgi?acc=GSE87892 | NCBI Gene Expression Omnibus, GSE87892 |
| Radhakrishnan A, Chen YH, Martin S, Alhusaini N | 2016 | Ribosome profiling study of Dhh1p overexpression and the dhh1 knockout strain using monosome-protected footprints | https://www.ncbi.nlm.nih.gov/geo/query/acc.cgi?acc=GSE81269 | NCBI Gene Expression Omnibus, GSE81269 |
| He F, Celik A, Wu C, Jacobson A | 2017 | Genome-wilde identification of decapping substrates in the yeast Saccharomyces cervisiae | https://www.ncbi.nlm.nih.gov/geo/query/acc.cgi?acc=GSE107841 | NCBI Gene Expression Omnibus, GSE107841 |
| Celik A, Baker R, He F, Jacobson A | 2016 | A High Resolution Profile of NMD Substrates in Yeast | https://www.ncbi.nlm.nih.gov/geo/query/acc.cgi?acc=GSE86428 | NCBI Gene Expression Omnibus, GSE86428 |
| Zeidan Q, He F, Zhang F, Zhang H | 2019 | Conserved mRNA-granule component Scd6 targets Dhh1 to repress translation initiation and activates Dcp2-mediated mRNA decay in vivo | https://www.ncbi.nlm.nih.gov/geo/query/acc.cgi?acc=GSE114892 | NCBI Gene Expression Omnibus, GSE114892 |

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
