## [Editor Report]

This fundamental study represents a real tour de force, demonstrating the impact of mutation on the mRNA decapping machinery. Accumulation of mRNAs in dcp2 mutants, is dependent both on the classical 5' to 3' pathway of mRNA decay and on the NMD pathway- highlighting the 'non-nonsense' roles of the NMD pathway and how little we really know about the complete set of pathways of mRNA degradation.

---

## [Decision Letter]

**Decision letter after peer review:**

Thank you for submitting your article "Decapping factor Dcp2 controls mRNA abundance and translation to adjust metabolism and filamentation to nutrient availability" for consideration by *eLife*. Your article has been reviewed by 3 peer reviewers, and the evaluation has been overseen by Timothy Nilsen as the Reviewing Editor and James Manley as the Senior Editor. The following individuals involved in review of your submission have agreed to reveal their identity: Mark Peter Ashe (Reviewer #2).

Essential revisions:

As you will see, all the reviewers were quite positive about the work as was the reviewing editor. Specifically, they thought that the demonstration of Dcp2's extensive roles in gene expression was quite interesting. No additional experiments are required, but each reviewer has made a number of comments regarding the text. Please address these issues as thoroughly as possible before submitting a revised version.

*Reviewer #1 (Recommendations for the authors):*

Vijjamarri et al. describe a multi-omics approach to determining the gene expression consequences of deleting the DCP2 gene in the yeast *Saccharomyces cerevisiae*. This gene encodes the catalytic subunit of yeast's mRNA decapping enzyme. Throughout the literature it has generally been understood that dcp2∆ strains fail to decap mRNAs and that this defect promotes the stabilization of mRNAs ordinarily subjected to the major 5' to 3' mRNA decay pathway. Previous evidence supporting this conclusion includes data demonstrating that, in dcp2∆ cells, several specific mRNAs maintain their 5' caps, increase their steady-state levels, and extend their half-lives. In this manuscript, the authors seek to beat a dead horse with more sophisticated transcriptome-wide methods, ultimately proving the original point, but also uncovering substantially interesting new results.

The initial experiments (Figure 1) first examined the authors' (and others') prior RNA-Seq datasets to determine the number of transcripts that either increase or decrease in abundance in dcp2∆ cells, ultimately identifying 1376 mRNAs that increase in abundance (so-called "derepressed mRNAs") and 1281 mRNAs that decrease in abundance (so-called "repressed mRNAs"). The respective changes were then validated by qRT-PCR for 10 out of 11 mRNAs tested. To be certain that the RNA-Seq results reflected a loss of mRNA decapping activity, they also evaluated RNA-Seq data from cells expressing catalytically dead Dcp2, an experiment that yielded results almost identical to those obtained with dcp2∆ cells.

Next, the authors sought to determine the role of Dhh1, a decapping activator, in Dcp2-mediated regulation by examining RNA-Seq results for dhh1∆ and dcp2∆dhh1∆ cells. These experiments showed that 752 of the 1376 mRNAs that increased in abundance in dcp2∆ cells could do so when only DHH1 was deleted (albeit to a lesser extent), with 607 mRNAs relatively unaffected by that deletion. These experiments, too, were consistent with previously published results indicating that only a subset of yeast mRNAs were targeted for decapping by Dhh1. Comparable analyses of strains deleted for UPF1, PAT1, PAT1/DHH1, or EDC3/SCD6 yielded results indicating that all of these strains, except for upf1∆ cells, stabilized significant fractions of the Dhh1-dependent mRNAs. upf1∆ cells, however, only appeared to stabilize mRNAs that were Dhh1-independent, results that were also consistent with published data on NMD substrates.

Subsequent experiments employed CAGE analysis and Rpb1 ChIP-seq to further evaluate the basis for increased or decreased levels of the populations of mRNAs identified in the RNA-Seq studies. CAGE analysis was employed to determine whether inferred cap status (capped vs decapped) was correct, and whether there existed subtle differences in cap status of distinct populations of mRNAs. The ChIP-seq experiments were used as an alternative means to testing whether increases or decreases in specific mRNAs reflected changes in transcription of their genes vs. mRNA stabilization. These experiments largely confirmed that mRNAs which accumulated in dcp2∆ cells (and other mutants affecting mRNA decapping) were capped (albeit to different extents) and that RNA polymerase loading differences could not, for the most part, account for the observed changes in mRNA levels.

An additional set of experiments evaluated ribosomal profiling data to assess translational efficiency (TE) of the mRNAs in the expanding subsets of mRNAs that changed in abundance as a consequence of mutations in genes regulating or catalyzing mRNA decapping. These analyses identified 541 mRNAs with TEs that increased in dcp2∆ cells (median increase: 1.65-fold), as well as 659 mRNAs whose TEs decreased in the same cells (median decrease: 0.59-fold). Much of the rest of the paper sought to explain these changes in TE. Agreement between the ribosomal profiling data and TMT/MS-MS analyses indicated that the TE differences reflected actual changes in the synthesis rates of the respective proteins. Subsequent analyses indicated that the translationally-repressed mRNAs generally had lower than average TE in WT cells whereas those that were translationally activated tended to have higher than average TE in WT. The authors' mass spec analyses also indicated that ribosomal protein levels decreased ~25% in dcp2∆ cells, suggesting that the TE differences in dcp2∆ cells might reflect competition caused by limiting ribosomes (or other translational factors) between mRNAs that normally were efficiently vs. inefficiently translated inherently. Numerous other experiments indicated that this indirect model for translational regulation in dcp2∆ cells was likely correct and also identified fine tuning of the expression of genes regulating respiration, alternative carbon and nitrogen sources, mitochondrial function, and filamentation.

During my first reading I was overwhelmed by the length of this manuscript (almost 140 pages!). It was by far the longest paper I had ever reviewed and I was feeling pity for its future readers. That feeling vanished as I reread the paper and absorbed the extent to which the authors had chased down so many details for the phenomena they were tracking, ultimately coming up with a thorough and convincing story, and making it clear to me that cutting this "whole" story up into distinct chapters would be a waste of both the authors' and future readers' time. Hence, I'm supportive of publishing this substantive manuscript in *eLife*, but here are a few revisions that might make it even better:

1. I'm not fond of the use of the terms "repressed" and "derepressed" to describe mRNAs that respectively decrease or increase in abundance in dcp2∆ cells. I understand that the terms comprise technically correct genetic nomenclature, but we know from prior work that there's no need to opt for vagueness here, i.e., these are mRNAs that go down or up in abundance. Please modify the text to make the observation more precise and describe the repressed class as mRNAs whose abundance decreases and the derepressed class as mRNAs whose abundance increases.

2. The CAGE analysis was quite informative, but the reliability of its measurements of "percent capped" is suspect. This comment follows from the data of Figure 3D, where steady-state levels of capped mRNAs appear to be ~40% of all mRNAs. This figure is at least half of what has been seen in other publications so the authors have an obligation to validate their CAGE analyses with measurements of the extent of capping in individual mRNAs. I recommend the use of cap IPs, but other methods could also work. Also, it's necessary to provide the capped/total data for the dcp2∆ cell samples.

3. On p. 13, the authors note that an "in-depth analysis" of pat1∆, pat1∆dhh1∆, and edc3∆/scd6∆ strains is to be published elsewhere. Is that really necessary? In light of the enormous amount of useful information already in this manuscript why the hesitancy to add a little more?

4. The authors' "old school" models in Figure 8 —figure supplement 1 appear to ignore the recent results and the model for multiple decapping complexes published recently by He et al. (*eLife* 2022 and FEBS Letters 2022). Do the authors think that their data conflicts with that of He et al? If so, the significant disagreements should be noted.

5. Is it possible that the reduced TE observed for a class of mRNAs in dhh1∆ cells that the authors explain as samples having enhanced decapping may also be targeted specifically by other decapping activators such as Edc3, Scd6, or Pat1? This data may already exist in the set designated for publication elsewhere.

6. Please explain the likely reason for the differences in absolute vs. relative differences in RNA polymerase binding to mRNAs. It's understood that the two datasets differ by the use or not of spike-in controls. However, since there are such large differences in the two datasets it would be helpful to have validation for some of the transcripts in the spike-in datasets that show large changes in abundance.

7. Some figure legends note that outliers have been omitted. Please provide an n for each case of outlier omission.

8. In Figure 3D columns 4 and 5 the authors compare Dhh1 dependent and independent decapping levels for the up_dcp2 mRNAs. Since iESR substrates are 24% of the Dhh1 dependent group they should be removed from the respective datasets to get a clearer picture of what's actually happening.

9. The authors continually refer to one of their datasets (GSE220578) as "unpublished." One cannot examine it in GEO (it's private) so we need to know whether additional authors should be added to this paper. Further, if that dataset is included in this manuscript can we please stop using the term "unpublished."

10. Abstract, line 30: nothing about Lsm2 was addressed in this manuscript.

11. P. 4, line 81: there's nothing about an autoinhibitory region in Fromm et al., 2012. The second reference that's needed is Paquette et al., 2018, ie, the same reference listed on line 78.

12. P. 4, line 63: calling Dcp2 a "bi-lobed enzyme" is probably much too simplistic. The rest of that sentence does a fine job of summarizing Dcp2's structure so I recommend deleting the "bi-lobed" thought.

13. P. 13, line 265: please correct spelling of "trasnscripts;" p. 15, line 299 ("versusvs."); p. 23, line 474, Figure 4F should be 3F; and p. 27, line 556, "do not" should be "does not".

*Reviewer #2 (Recommendations for the authors):*

This manuscript details a whole range of proteomic, transcriptomic, and ribosome profiling studies comparing mutants in the decapping enzyme, Dcp2 with a parental yeast strain. Furthermore, a careful, logical and scholarly assessment of the implications of the 'omics data are presented. The manuscript gives unprecedented detail at the transcription, mRNA stability and mRNA translation levels concerning the implications of a deficiency in mRNA decapping. The data show widespread mRNA accumulation due to reduced mRNA degradation and consequent alterations in the fine balance of translation regulation. Two areas that are not covered in great detail are the implications of higher level of P-bodies in the mRNA decay mutants and the potential role of the cytosolic exosome in mRNA degradation. However, no paper can cover everything and this manuscript already considers so many angles!

The authors don't mention that mRNA decay mutants such as dcp2delta have constitutive very high levels of P-bodies (Teixeira et al. 2007. PMID: 17429074). This means much of the material that is measured could be present in different contexts within the cell between the mutant and wild type. For instance, do the authors generally spin their whole cell lysates before they conduct their 'omics experiments – if so at least some of this material could have been lost. In my view the connections with P-bodies and biological condensates should be discussed, and any limitations in the study given the presence of these condensates covered.

The idea that increased expression of mRNA and ribosomes in yeast could alter the translational profile in mutants of the mRNA decay pathway has been put forward before in the pre-'omics era. In the paper, the mRNA decay mutants were resistant to the effects of glucose and amino acid starvation. This paper (Holmes et al. 2004. PMID: 15024087) is cited in the manuscript, but not in this context. In my view the authors should cite and discuss this mass action model given the similarities to their hypothesis.

Have the authors cross-compared their datasets with datasets from the Tollervey group looking at exosome substrates?

More specific points

Line 46- eIF4F does not really exist in this context in yeast – i.e. eIF4A is only quite weakly associated and does not generally accompany eIF4G and eIF4E at anywhere near a stoichiometric level- it might be better to get away from mentioning eIf4F

Line 55- the authors state that the exosome pathway is a minor pathway of mRNA decay – while this narrative exists in the literature – I am not clear what the evidence is and, if it exists, it should be cited here. Otherwise, while Delan-Forino et al., 2017. PMID 28355211, don't directly assess the relative contribution of different mRNA decay pathways, they do show that substantial numbers of cytoplasmic mRNAs interact with the exosome.

Line 148- 'suggesting that either Dhh1 or Pat1 is sufficient for repression of translation initiation in glucose starvation'. While Dhh1 and Pat1 may be sufficient to allow repression of translation- the mechanism of repression more likely involves targeting of eIF4A and/or Ded1 (PMID: 33053322, PMID: 21795399, PMID: 34946015). So the impact of the mRNA decay mutants is more likely indirect -see comments above.

Line 214- 'all but one mRNA' – I may have missed it but what is this mRNA?

Line 245- this conclusion implies that the increases in mRNA levels observed are due to a direct impact of DCP2 deletion on mRNA degradation. At this stage in the manuscript, the authors haven't addressed potential transcriptional effects and so the conclusion needs to be more accurately worded.

Line 250- maybe change 'Edcs' to 'Edc1-3'.

Line 483, the authors state most studies have focused on mRNA decay – they need some citations here if they word it in this manner.

Line 495 'These results suggest that Dcp2 broadly controls gene expression at the translational level in addition to regulating mRNA stability'. Given that the authors will later conclude that the impact of the mutant on translation is due to indirect effects on mRNA and ribosome levels – this conclusion is too strongly worded for me.

Line 496 'in an effort to establish' sounds like the result was already known before the experiment. 'In order to evaluate whether' – or something like this would sound better.

Line 519 – relating to Figure 4D- do the authors have controls they could add to this – an mRNA where RPF levels are reduced or are maintained in the mutant relative to the parent strain?

Line 536- YLR361C-A?

Line 569 – is this sulfometuron or sulfometuron methyl?

On Figure 6E, there appears to be an error on the y-axis of the plo.t

Overall though an excellent paper.

*Reviewer #3 (Recommendations for the authors):*

The manuscript by Vijjamarri et al., is focused on the role of the Dcp2 decapping enzyme in *S. cerevisiae*. Using a Dcp2 disrupted strain (dcps∆) they address the downstream consequences of the absence of the primary fungal decapping enzyme Dcp2. Although the lack of Dcp2 decapping is expected to result in the accumulation of mRNAs that would have otherwise been degraded and lead to indirect downstream consequences on gene expression and cellular physiology, these outcomes have not been addressed. Here the authors undertake a very thorough analysis. One important and surprising contribution is the association of Dcp2 to decreased translation efficiency. Rather than an expected mechanism of reduced stability contributing to reduced translation, the authors show that this is an indirect effect of Dcp2 by modulating ribosomal protein mRNA stability that in turn leads to a reduction of ribosomal protein levels and competition for limiting ribosomes for translation. A network of mRNAs that are regulated by Dcp2 that are involved in modulating the repression of aerobic growth in glucose-rich media were also uncovered. These discoveries further unravel the indirect contribution of Dcp2 decapping during respiratory growth under non-glucose carbon sources. This is an extremely thorough evaluation of Dcp2 function in yeast and will be an important contribution to the field.

1. Can the authors further elaborate on why they think they do not see (or minimally see) the contribution of codon optimality to Dhh1 directed mRNA decay contrary to current publications.

2. In Figure 5E, the terms "strong" and "weak" mRNA nomenclature should be further clarified or changed.

3. The authors should consider altering Figure 8 to present the functional contribution of Dcp2 rather than the outcome of dcp2 disruption. It seems more informative (to me at least) to state what Dcp2 does rather than what it's not doing when it is absent (as currently depicted in Figure 8).

4. Figure 5C: A brief explanation of C/T should be provided in the text or figure legend to avoid having to search through the M&M section to figure it out.

5. Define TE in the abstract.

6. Lastly, I found the manuscript to be extremely long (more than double the recommended manuscript length for *eLife*). Although I don't recommend it for this manuscript, I would strongly suggest the authors consider splitting such a manuscript into at least two papers in the future. It is hard to envision most readers retaining everything that is presented in the manuscript by the time they are finished reading it and it simply dilutes the impact.

---

## [Author Response]

Essential revisions:As you will see, all the reviewers were quite positive about the work as was the reviewing editor. Specifically, they thought that the demonstration of Dcp2's extensive roles in gene expression was quite interesting. No additional experiments are required, but each reviewer has made a number of comments regarding the text. Please address these issues as thoroughly as possible before submitting a revised version.Reviewer #1 (Recommendations for the authors):Vijjamarri et al. describe a multi-omics approach to determining the gene expression consequences of deleting the DCP2 gene in the yeast *Saccharomyces cerevisiae*. This gene encodes the catalytic subunit of yeast's mRNA decapping enzyme. Throughout the literature it has generally been understood that dcp2∆ strains fail to decap mRNAs and that this defect promotes the stabilization of mRNAs ordinarily subjected to the major 5' to 3' mRNA decay pathway. Previous evidence supporting this conclusion includes data demonstrating that, in dcp2∆ cells, several specific mRNAs maintain their 5' caps, increase their steady-state levels, and extend their half-lives. In this manuscript, the authors seek to beat a dead horse with more sophisticated transcriptome-wide methods, ultimately proving the original point, but also uncovering substantially interesting new results.[…]During my first reading I was overwhelmed by the length of this manuscript (almost 140 pages!). It was by far the longest paper I had ever reviewed and I was feeling pity for its future readers. That feeling vanished as I reread the paper and absorbed the extent to which the authors had chased down so many details for the phenomena they were tracking, ultimately coming up with a thorough and convincing story, and making it clear to me that cutting this "whole" story up into distinct chapters would be a waste of both the authors' and future readers' time. Hence, I'm supportive of publishing this substantive manuscript in eLife, but here are a few revisions that might make it even better:

We are gratified that the reviewer appreciated the depth of our analysis, but also appreciate that the length of the manuscript would be daunting to many readers. We have shortened the main text by ~8% to make the presentation more concise despite the addition of new text to address the various reviewers’ comments.

1. I'm not fond of the use of the terms "repressed" and "derepressed" to describe mRNAs that respectively decrease or increase in abundance in dcp2∆ cells. I understand that the terms comprise technically correct genetic nomenclature, but we know from prior work that there's no need to opt for vagueness here, i.e., these are mRNAs that go down or up in abundance. Please modify the text to make the observation more precise and describe the repressed class as mRNAs whose abundance decreases and the derepressed class as mRNAs whose abundance increases.

We replaced almost all occurrences of the term “derepressed” with “increased“, “elevated” or “up-regulated”, and many of the instances of “repressed” with “reduced” or “down-regulated”.

2. The CAGE analysis was quite informative, but the reliability of its measurements of "percent capped" is suspect. This comment follows from the data of Figure 3D, where steady-state levels of capped mRNAs appear to be ~40% of all mRNAs. This figure is at least half of what has been seen in other publications so the authors have an obligation to validate their CAGE analyses with measurements of the extent of capping in individual mRNAs. I recommend the use of cap IPs, but other methods could also work. Also, it's necessary to provide the capped/total data for the dcp2∆ cell samples.

In this study we calculated the ratio of normalized CAGE reads (in TPMs) to normalized RNA-seq reads (in TPMs), or (C/T), for every gene in each strain as a proxy of the % capped transcripts. This is analogous to the inference of translation efficiency (TE) obtained by dividing normalized ribosome footprint reads by independently normalized RNA sequencing reads. Thus, C/T is a relative, not actual, measure of the proportion of capped molecules, and we revised the manuscript to make this clear. Similar to TE values, the change in C/T ratios for a given gene can be used to infer relative changes in % capped transcripts between different strains. If increased mRNA levels in *dcp2∆* cells are driven by impaired decapping and decay vs. transcription, then we should observe an increased proportion of capped transcripts for the mRNAs up-regulated by *dcp2∆*. To test this prediction, the true proportion of capped molecules is not needed and can be evaluated using the C/T ratios we obtained for each gene. Our results in Figure 3E showing a ~1.6-fold higher median C/T ratio in *dcp2∆* vs WT cells for the mRNAs up-regulated by *dcp2∆* vs. all mRNAs supports the decapping mechanism.

To provide independent evidence that impaired decapping drives transcript abundance changes conferred by *dcp2∆*, we have now calculated the ratios of CAGE TPMs between mutant and WT and the corresponding ratios of total RNA TPMs. If changes in mRNA abundance result primarily from altered transcription, we expect similar changes in both ratios for most up-regulated transcripts, ie. TPM_CAGE_*dcp2∆*/WT_ ≈ TPM_RNA_*dcp2∆*/WT_. If instead impaired decapping is responsible, then we expect to find TPM_CAGE_*dcp2∆*/WT_ > TPM_RNA_*dcp2∆*/WT_ for the mRNAs elevated by *dcp2∆.* Consistent with the latter, the majority of mRNA_up_*dcp2∆* transcripts show TPM_CAGE_*dcp2∆*/WT_ ratios exceeding the corresponding TPM_RNA_*dcp2∆*/WT_ ratios, as shown in the new Figure 3—figure supplement 3D.

3. On p. 13, the authors note that an "in-depth analysis" of pat1∆, pat1∆dhh1∆, and edc3∆/scd6∆ strains is to be published elsewhere. Is that really necessary? In light of the enormous amount of useful information already in this manuscript why the hesitancy to add a little more?

Unfortunately, adding a complete analysis of the RNA-seq and Ribo-seq data on the *pat1∆, pat1∆dhh1∆,* and *edc3∆/scd6∆* mutants would increase the length of this already hefty manuscript prohibitively.

4. The authors' "old school" models in Figure 8 —figure supplement 1 appear to ignore the recent results and the model for multiple decapping complexes published recently by He et al. (eLife 2022 and FEBS Letters 2022). Do the authors think that their data conflicts with that of He et al? If so, the significant disagreements should be noted.

We thank the reviewer for this excellent suggestion. We have modified Figure 8 —figure supplement 1 to reflect recent advances from the Jacobson lab on the network of interactions of decapping activators with distinct segments in the C-terminal tail of Dcp2 and formation of different decapping complexes that recognize distinct mRNAs targeted by the activators. Our genome-wide RNA-seq data from mutants lacking different activators suggests that many of the group of Dhh1-dependent targets of Dcp2 are frequently recognized by a Dcp1-Dcp2 decapping complexes containing all four decapping activators Dhh1/Edc3/Scd6/Pat1, which is reflected in panel B of our revised model. However, as now mentioned in the DISCUSSION, close inspection of the cluster analysis of mRNA changes shown in Figure 2B suggests that certain Dhh1-dependent transcripts are targeted differentially by Pat1 versus Dhh1, Edc3, and Scd6, which is consistent with evidence for distinct complexes of Dcp2 bound to particular subsets of decapping activators (He et al. 2022). Our data further suggest that the Dhh1-independent set of mRNAs appear to be targeted by the Upf proteins independently of Dhh1/Edc3/Scd6/Pat1. This might represent a departure from the model proposed by He et al. (2022) wherein the complex containing Upf1/Xrn1 also contains Edc3. It is possible however that the Upf1-containing complex contains Edc3, as they proposed, even though Edc3 is not required for Upf1-mediated targeting of the majority of Dhh1-independent mRNAs, as we found. We have tried to depict this subtlety in the new version of the figure and added new text to the DISCUSSION on this issue.

5. Is it possible that the reduced TE observed for a class of mRNAs in dhh1∆ cells that the authors explain as samples having enhanced decapping may also be targeted specifically by other decapping activators such as Edc3, Scd6, or Pat1? This data may already exist in the set designated for publication elsewhere.

We do observe that transcripts showing TE derepression in *dcp2∆* cells dependent on Dhh1 (Dhh1-dep. TE_up_dcp2) also tend to be targeted for decapping and degradation by Pat1 in addition to Dhh1, which will be explained in our subsequent manuscript on functional interplay between Pat1 and Dhh1 in mRNA turnover and translational repression.

6. Please explain the likely reason for the differences in absolute vs. relative differences in RNA polymerase binding to mRNAs. It's understood that the two datasets differ by the use or not of spike-in controls. However, since there are such large differences in the two datasets it would be helpful to have validation for some of the transcripts in the spike-in datasets that show large changes in abundance.

It’s important to note that applying spike-in normalization with *S. pombe* chromatin does not change the ranking of genes according to their changes in Rpb1 occupancies in mutant vs. WT compared to the ranking established using relative Rpb1 occupancies normalized only for library depth, as spike-in normalized occupancies are obtained simply by multiplying the observed occupancies for each gene by a single size factor calculated from the sum of Rpb1 occupancies for all *S. pombe* genes in that sample. By revealing absolute changes in Pol II occupancies, the spike-in normalized data showed that transcription is reduced overall in the decapping mutants, such that genes showing moderate increases in relative Rpb1 occupancies (ie. normalized only for library depth) actually undergo moderate reductions in absolute occupancies (spike-in normalized data). Regardless of whether relative or absolute Rpb1 occupancies were considered however, we arrived at the same conclusion that nearly all transcripts up-regulated by *dcp2Δ* do not exhibit increased Pol II occupancies for the corresponding genes. Thus, to simplify and condense the presentation, we have revised the RESULTS to describe only the spike-in normalized Rpb1 data because it reveals the absolute changes in transcription rate that can be compared directly with the absolute changes in mRNA abundance determined using the ERCC-normalized RNA-seq data.

7. Some figure legends note that outliers have been omitted. Please provide an n for each case of outlier omission.

We have now indicated the number of outliers that were omitted in the legends of each figure.

8. In Figure 3D columns 4 and 5 the authors compare Dhh1 dependent and independent decapping levels for the up_dcp2 mRNAs. Since iESR substrates are 24% of the Dhh1 dependent group they should be removed from the respective datasets to get a clearer picture of what's actually happening.

This is a valid point. As now indicated in RESULTS, we observed nearly identical results as those shown in Figures 3D-E after excluding all ESR transcripts from consideration.

9. The authors continually refer to one of their datasets (GSE220578) as "unpublished." One cannot examine it in GEO (it's private) so we need to know whether additional authors should be added to this paper. Further, if that dataset is included in this manuscript can we please stop using the term "unpublished."

We omitted “unpublished” from the manuscript and all datasets are now available on GEO.

10. Abstract, line 30: nothing about Lsm2 was addressed in this manuscript.

We have omitted “Lsm2” from the abstract.

11. P. 4, line 81: there's nothing about an autoinhibitory region in Fromm et al., 2012. The second reference that's needed is Paquette et al., 2018, ie, the same reference listed on line 78.

Thank you for pointing out the error. We have replaced Fromm et al., 2012 with Paquette et al., 2018 in the revised version.

12. P. 4, line 63: calling Dcp2 a "bi-lobed enzyme" is probably much too simplistic. The rest of that sentence does a fine job of summarizing Dcp2's structure so I recommend deleting the "bi-lobed" thought.

We have deleted “bi-lobed” from the sentence.

13. P. 13, line 265: please correct spelling of "trasnscripts;" p. 15, line 299 ("versusvs."); p. 23, line 474, Figure 4F should be 3F; and p. 27, line 556, "do not" should be "does not".

We regret these spelling mistakes and figure mislabeling, which were corrected in the revised version of the manuscript.

Reviewer #2 (Recommendations for the authors):This manuscript details a whole range of proteomic, transcriptomic, and ribosome profiling studies comparing mutants in the decapping enzyme, Dcp2 with a parental yeast strain. Furthermore, a careful, logical and scholarly assessment of the implications of the 'omics data are presented. The manuscript gives unprecedented detail at the transcription, mRNA stability and mRNA translation levels concerning the implications of a deficiency in mRNA decapping. The data show widespread mRNA accumulation due to reduced mRNA degradation and consequent alterations in the fine balance of translation regulation. Two areas that are not covered in great detail are the implications of higher level of P-bodies in the mRNA decay mutants and the potential role of the cytosolic exosome in mRNA degradation. However, no paper can cover everything and this manuscript already considers so many angles!

We are gratified by the reviewer’s positive assessment of the manuscript, and his/her appreciation that the manuscript covers multiple aspects of post-transcriptional control exerted by Dcp2, even though it did not address the implications for P-bodies and role of the cytoplasmic exosome. We have added new text relating to these last topics however, as summarized below.

The authors don't mention that mRNA decay mutants such as dcp2delta have constitutive very high levels of P-bodies (Teixeira et al. 2007. PMID: 17429074). This means much of the material that is measured could be present in different contexts within the cell between the mutant and wild type. For instance, do the authors generally spin their whole cell lysates before they conduct their 'omics experiments – if so at least some of this material could have been lost. In my view the connections with P-bodies and biological condensates should be discussed, and any limitations in the study given the presence of these condensates covered.

Thanks for raising this important point. All of the -omics experiments described here involved clarifying cell lysates by high-speed centrifugation, which might have resulted in depletion of P-bodies to an extent that differs between *dcp2* mutant and WT cells. We have added text to the Discussion section to acknowledge this limitation of our study and the likelihood of its importance to our final conclusions: “Accumulation of mRNA degradation intermediates in *dcp1*Δ or *xrn1*∆ cells greatly elevates PB assembly during exponential growth in glucose medium (Sheth and Parker 2003; Teixeira and Parker 2007). Our RNA-Seq and Ribo-Seq experiments involved centrifugation of cell extracts to remove cellular debris, which might have depleted mRNAs associated with PBs preferentially in *dcp2*∆ vs. WT cells (Kershaw et al. 2021). This may not be an important limitation of our study however, because *dcp2*∆ leads to much smaller PBs compared to *dcp1*Δ or *xrn1*Δ in glucose medium (Teixeira and Parker 2007). Moreover, *dhh1*∆ does not increase PBs in glucose-grown cells (Teixeira and Parker 2007), yet we observed highly similar up-regulation of Dhh1-dependent mRNAs between *dcp2*∆ and *dhh1*∆ cells (Figure 2B).”

The idea that increased expression of mRNA and ribosomes in yeast could alter the translational profile in mutants of the mRNA decay pathway has been put forward before in the pre-'omics era. In the paper, the mRNA decay mutants were resistant to the effects of glucose and amino acid starvation. This paper (Holmes et al. 2004. PMID: 15024087) is cited in the manuscript, but not in this context. In my view the authors should cite and discuss this mass action model given the similarities to their hypothesis.

We thank the reviewer for reminding us of this important study and have added the following text to the Discussion. “There is precedent for our proposal that elevated mRNA levels in *dcp2*∆ cells promote translational reprogramming. Holmes et al. (2004) showed that suppression of bulk translation initiation under different stress conditions is diminished in various mutants that either lack decapping factors, accumulate free 40S subunits, or overexpress eIF4G, and concluded that increased amounts of mRNA, free 40S subunits or eIF4G can drive assembly of 48S PICs and counteract regulatory responses (eg. eIF2α phosphorylation) that diminish PIC assembly during stress. It is possible therefore that the increased mRNA levels conferred by *dcp2*∆ mitigates the effects of decreased ribosome levels conferred by the ESR on 48S PIC formation; however, our results suggest that the increased mRNA:40S ratio still favors translation of the mRNAs best able to compete for limiting PICs in *dcp2∆* cells.”

Have the authors cross-compared their datasets with datasets from the Tollervey group looking at exosome substrates?

We did not make this comparison because we assumed that mRNAs targeted by Dcp2 would be degraded 5’-to-3’ by Xrn1 rather than 3’-to-5’ by the exosome. At the referee’s suggestion we have determined that the mRNAs we identified as targeted by Dcp2 for degradation (mRNA_up_*dcp2∆*) are significantly depleted for exosome substrates defined by Tollervey et al. based on their association with exosome subunits Rrp44, Rrp41 and Rrp6 by Schneider et al., (PMID: 23000172). This finding suggests that most mRNAs that undergo decapping-mediated 5’ to 3’ decay are not preferentially targeted for 3’ to 5’ degradation by the exosome. This new insight is now mentioned in the Discussion.

More specific pointsLine 46- eIF4F does not really exist in this context in yeast – i.e. eIF4A is only quite weakly associated and does not generally accompany eIF4G and eIF4E at anywhere near a stoichiometric level- it might be better to get away from mentioning eIf4F

We modified the sentence to replace eIF4F with eIF4E and eIF4G.

Line 55- the authors state that the exosome pathway is a minor pathway of mRNA decay – while this narrative exists in the literature – I am not clear what the evidence is and, if it exists, it should be cited here. Otherwise, while Delan-Forino et al., 2017. PMID 28355211, don't directly assess the relative contribution of different mRNA decay pathways, they do show that substantial numbers of cytoplasmic mRNAs interact with the exosome.

We modified the sentence in the Introduction to read: “Deadenylation is followed by either 3’ to 5’ exonucleolytic degradation by the cytoplasmic exosome complex decapped by the conserved Dcp1/Dcp2 holoenzyme, which hydrolyzes the cap to release m^7^GDP and 5’ monophosphate RNA, which is then degraded 5’ to 3’ by the exoribonuclease Xrn1”

Line 148- 'suggesting that either Dhh1 or Pat1 is sufficient for repression of translation initiation in glucose starvation'. While Dhh1 and Pat1 may be sufficient to allow repression of translation- the mechanism of repression more likely involves targeting of eIF4A and/or Ded1 (PMID: 33053322, PMID: 21795399, PMID: 34946015). So the impact of the mRNA decay mutants is more likely indirect -see comments above.

This point is well taken and we have modified the Introduction to include the alternative, indirect mechanism for recovery of translation in stressed *pat1∆* and *dhh1∆* cells proposed by Holmes et al., and have also mentioned the evidence that targeting eIF4A and Ded1 is involved in suppressing translation during stress: “Alternatively, the recovery of polysomes during stress in *dhh1*∆ and *pat1*∆ mutants could result indirectly from a broad stabilization of mRNAs that helps to restore 48S PIC assembly by mass action and overcome inhibition of PIC formation during stress produced by other means (Holmes et al. 2004). Indeed, impairing the functions of RNA helicases eIF4A and Ded1, which normally stimulate 48S PIC assembly, is involved in suppressing translation during stress (Castelli et al. 2011; Bresson et al. 2020; Iserman et al. 2020; Sen et al. 2021).” We thank the reviewer for reminding us of these important findings.

Line 214- 'all but one mRNA' – I may have missed it but what is this mRNA?

We now stipulate that *YOR178C* mRNA is the one mRNA showing a significant difference between the *dcp2*∆ and *dcp2-EE* mutants.

Line 245- this conclusion implies that the increases in mRNA levels observed are due to a direct impact of DCP2 deletion on mRNA degradation. At this stage in the manuscript, the authors haven't addressed potential transcriptional effects and so the conclusion needs to be more accurately worded.

We have modified the statement to read: “Together, the results suggest that only about one-half of the mRNAs repressed in abundance by Dcp2 require Dhh1 for this repression.”

Line 250- maybe change 'Edcs' to 'Edc1-3'.

Edcs has been changed to Edc1-3.

Line 483, the authors state most studies have focused on mRNA decay – they need some citations here if they word it in this manner.

We eliminated the sentence.

Line 495 'These results suggest that Dcp2 broadly controls gene expression at the translational level in addition to regulating mRNA stability'. Given that the authors will later conclude that the impact of the mutant on translation is due to indirect effects on mRNA and ribosome levels – this conclusion is too strongly worded for me.

We have modified the sentence to: “These results suggest that Dcp2 broadly controls gene expression directly or indirectly at the translational level in addition to regulating mRNA stability.”

Line 496 'in an effort to establish' sounds like the result was already known before the experiment. 'In order to evaluate whether' – or something like this would sound better.

We modified the sentence to: “To evaluate whether the changes in TE are generally associated with changes in the synthesis and abundance of the encoded proteins…”

Line 519 – relating to Figure 4D- do the authors have controls they could add to this – an mRNA where RPF levels are reduced or are maintained in the mutant relative to the parent strain?

Unfortunately, we did not construct *nLUC* reporters for any gene whose RPF levels are maintained or reduced in *dcp2∆* vs. WT cells.

Line 536- YLR361C-A?

The name of the gene was corrected to *YLR361C-A*.

Line 569 – is this sulfometuron or sulfometuron methyl?

We replaced sulfometuron with sulfometuron methyl.

On Figure 6E, there appears to be an error on the y-axis of the plo.t

Thanks for pointing out this error. We modified the y-axis of the figure 6E.

Overall though an excellent paper.

We greatly appreciate the positive assessment.

Reviewer #3 (Recommendations for the authors):The manuscript by Vijjamarri et al., is focused on the role of the Dcp2 decapping enzyme in *S. cerevisiae*. Using a Dcp2 disrupted strain (dcps∆) they address the downstream consequences of the absence of the primary fungal decapping enzyme Dcp2. Although the lack of Dcp2 decapping is expected to result in the accumulation of mRNAs that would have otherwise been degraded and lead to indirect downstream consequences on gene expression and cellular physiology, these outcomes have not been addressed. Here the authors undertake a very thorough analysis. One important and surprising contribution is the association of Dcp2 to decreased translation efficiency. Rather than an expected mechanism of reduced stability contributing to reduced translation, the authors show that this is an indirect effect of Dcp2 by modulating ribosomal protein mRNA stability that in turn leads to a reduction of ribosomal protein levels and competition for limiting ribosomes for translation. A network of mRNAs that are regulated by Dcp2 that are involved in modulating the repression of aerobic growth in glucose-rich media were also uncovered. These discoveries further unravel the indirect contribution of Dcp2 decapping during respiratory growth under non-glucose carbon sources. This is an extremely thorough evaluation of Dcp2 function in yeast and will be an important contribution to the field.1. Can the authors further elaborate on why they think they do not see (or minimally see) the contribution of codon optimality to Dhh1 directed mRNA decay contrary to current publications.

We believe there is no discrepancy with the previous work implicating codon non-optimality in Dhh1-mediated mRNA decay. Previously, in Zeidan et al. (2018), we observed an inverse correlation between sTAI values and changes in mRNA abundance in *dhh1∆* versus WT cells evident in our RNA-seq data and also that of Radhakrishnan et al., consistent with the model that Dhh1 targets mRNAs occupied by elongating ribosomes paused at slowly decoded codons. Our current findings that the group of mRNAs most highly repressed by Dcp2 dependent on Dhh1 exhibit average codon optimality could be explained if they contain concentrated runs of non-optimal codons that don’t register in the averaged values (as we suggested), or might simply indicate that other properties besides codon nonoptimality are more important in dictating their preferential degradation by Dcp2/Dhh1. We have added the latter explanation to the Results.

2. In Figure 5E, the terms "strong" and "weak" mRNA nomenclature should be further clarified or changed.

We have defined the terms "strong" and "weak" in the RESULTS and figure legends as well-translated or poorly-translated mRNAs, respectively, owing to their differing abilities to assemble 48S PIC in WT cells.

3. The authors should consider altering Figure 8 to present the functional contribution of Dcp2 rather than the outcome of dcp2 disruption. It seems more informative (to me at least) to state what Dcp2 does rather than what it's not doing when it is absent (as currently depicted in Figure 8).

Thanks for the suggestion. We modified Figure 8 and its legend to depict the role of Dcp2 protein in controlling different biological functions in WT cells.

4. Figure 5C: A brief explanation of C/T should be provided in the text or figure legend to avoid having to search through the M&M section to figure it out.

We defined C/T as “capped mRNA TPMs/total mRNA TPMs” both in RESULTS and in the Figure 5C legend.

5. Define TE in the abstract.

We added the definition for TE in the abstract.

6. Lastly, I found the manuscript to be extremely long (more than double the recommended manuscript length for eLife). Although I don't recommend it for this manuscript, I would strongly suggest the authors consider splitting such a manuscript into at least two papers in the future. It is hard to envision most readers retaining everything that is presented in the manuscript by the time they are finished reading it and it simply dilutes the impact.

We appreciate the suggestion and we acknowledge that the manuscript is lengthy. The results on translational changes (TE) are intertwined with the mRNA abundances changes, and dysregulation at both levels of gene expression contributes to altered protein expression and biological functions conferred by *dcp2∆*. While this precludes publishing the data separately, we shortened the main text by ~8% to make it more concise while also adding new text as requested by the referees.